



**Distribution, seasonality, optical characteristics, and fluxes of dissolved organic**
**matter (DOM) in the Pearl River (Zhujiang) estuary, China**
Yang Li[1], Guisheng Song[2], Philippe Massicotte[3], Fangming Yang[2], Ruihuan Li[4], Huixiang Xie[5,1]
[1] College of Marine and Environmental Sciences, Tianjin University of Science & Technology, Tianjin,
300457, China
[2] School of Marine Science and Technology, Tianjin University, Tianjin, 300072, China
[3] Takuvik Joint International Laboratory (UMI 3376) Université Laval (Canada) & Centre National de
la Recherche Scientifique (France), Université Laval, Québec, G1V 0A6, Canada
[4] State Key Laboratory of Tropical Oceanography, South China Sea Institute of Oceanology, Chinese
Academy of Science, Guangzhou, 510301, China
[5] Institut des sciences de la mer de Rimouski, Université du Québec à Rimouski, Rimouski (Québec),
G5L 3A1, Canada
**Correspondence to:** Guisheng Song (guisheng.song@tju.edu.cn); Huixiang Xie
(huixiang_xie@uqar.ca)





## Abstract

Dissolved organic carbon concentration in the Pearl River estuary (PRE) of China was measured in May, August, and October 2015 and January 2016. Chromophoric and fluorescent dissolved organic matter (CDOM and FDOM) in the latter three seasons were characterized by absorption and fluorescence spectroscopy. Parallel factor analysis of the fluorescence spectra identified two protein-like, two humic-like, and one oxidized quinone-like FDOM components. The seasonality of average DOM abundance varied as follows: DOC: May (156 μmol L$^{-1}$) > January (114 μmol L$^{-1}$) ≈ August (112 μmol L$^{-1}$) > November (86 μmol L$^{-1}$); CDOM absorption at 330 nm: August (1.76 m$^{-1}$) > November (1.39 m$^{-1}$) ≈ January (1.30 m$^{-1}$); FDOM expressed as the sum of the maximum fluorescence intensities of all FDOM components: November (1.77 R.U.) > August (1.54 R.U.) ≈ January (1.49 R.U.). Average DOM abundance in surface water was higher than in bottom water, their difference being marginal (0.1–10%) for DOC in all seasons and for CDOM and FDOM in November and January, and moderate (16–21%) for CDOM and FDOM in August. DOC showed little cross-estuary variations in all seasons while CDOM and FDOM in January were higher on the west side of the estuary than in the middle and on the east side. All three variables exhibited large variations and/or rapid drawdowns at the head of the estuary (salinity <5) due to multiple freshwater endmembers and/or biotic losses. In the saltier zone, they declined linearly with salinity except relatively constant DOC in May and November. The decrease in FDOM was 5–35% faster than that in CDOM, which in turn was 2–3 times faster than that in DOC. Salinity and CDOM absorption coefficients can serve as indicators of DOC in August and January. Absorbance- and fluorescence-based indices demonstrate that freshwater endmembers in all seasons mainly contained fresh, protein-rich DOM of microbial origin, though the proportion of humic-like components was somewhat higher in August. Protein-like materials were preferentially consumed in the low-salinity section but the dominance of the microbial signature was maintained throughout the saltier zone. Exports of DOC and CDOM (in terms of $a_{330}$)



into the South China Sea were estimated as $195 \times 10^9$ g and $266 \times 10^9$ m$^2$ for the PRE, and $362 \times 10^9$ g
and $493 \times 10^9$ m$^2$ for the entire Pearl River Delta. Compared to other world major estuaries, the PRE
presents the lowest concentrations and export fluxes of DOC and CDOM. Nonetheless, DOM delivered
by the PRE is protein-rich and thus may significantly impact the local ecosystem.
**1 Introduction**
**1.1 Overview of DOM**
Dissolved organic matter (DOM) in the ocean drives major biogeochemical cycles involving carbon,
nutrients, trace metals, and trace gases (Miller and Zepp, 1995; Cauwet, 2002; Wells, 2002; Cobble,
2007). The chromophoric component of DOM (CDOM), which absorbs solar ultraviolet (UV) and
visible radiation (Blough et al., 1993; Nelson et al., 1998; Siegel et al., 2002), affects ocean optics and
generates various photoreactions (Mopper and Kieber, 2002; Zafiriou, 2002; Zepp, 2003). The
biogeochemical and optical significance of DOM depends on both its abundance and quality (i.e.
chemical composition), with the latter strongly linked to its origin of formation (Repeta, 2015; Lønborg
et al., 2016, Massicotte et al., 2017).
Coastal waters, particularly those impacted by river plumes, contain higher contents of DOM
relative to the open ocean. DOM in river water originates from soil leaching (terrigenous DOM, or
tDOM) and in situ microbial production. tDOM, enriched with lignin phenols (Opsahl and Benner,
1997), differs substantially from microbial-derived DOM, enriched with proteins and aliphatic
hydrocarbons (Martínez-Pérez, et al., 2017; Brogi et al., 2018), in optical property and biological and
photochemical lability (Hansen et al., 2016; Sulzberger and Arey, 2016). Consequently, river runoff
can profoundly impact coastal ecosystem functioning by increasing the quantity and altering the quality
of DOM in coastal waters.



The loads of terrigenous and microbial DOM and their proportions in a river rely on many factors,
among which precipitation and the vegetation type of the catchment area are key players. Forest-
covered soils leach tDOM, while agricultural land boosts microbial DOM production by delivering to
rivers fertilizers that fuel biological activity. High precipitation during wet seasons flushes more tDOM
from soils into rivers compared to low precipitation during dry seasons (Fichot et al., 2014; Li et al.,
2015). On the other hand, the residence time of river water during wet seasons is shorter than that
during dry seasons, which may decrease autochthonous DOM production (Taylor et al., 2003).
Furthermore, DOM in rivers may be subject to physical (e.g. flocculation and coagulation, Asmala et
al., 2014), biological (e.g. microbial uptake, Benner and Kaiser, 2011), and chemical (e.g.
photobleaching, Vecchio and Blough, 2002) removals during estuarine mixing, thereby reducing its
abundance and modifying its chemical and optical properties before being exported to coastal seas.
Conversely, biological production can add organic matter to the DOM pool transported downstream
from the rivers (Bianchi et al., 2004; Fellman et al., 2010; Benner and Kaiser, 2011; Deutsch et al.,
2012). In highly urbanized areas, industrial and residential sewages can be a significant contributor of
DOM to river systems (Baker, 2001; Guo et al., 2014). Pollutions not only directly bring in
anthropogenic DOM but also carry nutrients that enhance biological DOM production.

**1.2 The Pear River estuary**
The Pearl River extends for 2214 km and has a catchment area of 450,000 km$^2$ (Lloyd et al., 2003;
Zhang et al., 2008), with its entire drainage basin located south of 27°N in the subtropical zone. After
entering the delta area, the Pearl River becomes a complex water network because of the continuous
bifurcation of three main tributaries (the West, North, and East Rivers) and other smaller rivers (Fig. 1).
The Pearl River system is connected to the South China Sea via three estuaries, Lingdingyang,

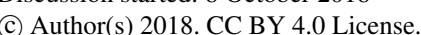



Modaomen, and Huangmaohai. The Lingdingyang estuary, the principal estuary of the Pearl River, is
commonly referred to as the Pearl River estuary (PRE).

The PRE stretches for ~70 km, covers an area of ~2000 km$^2$, and has an average depth of 4.8 m

(Dong et al., 2004). Its topography is featured by three shoals (the east, west, and middle shoals; depths
< 2 m) separated by two channels (the east and west channels; depths >5 m) which merge in the upper
reach of the estuary near Humen (Wai et al., 2004) (Fig. 1). Tides in the PRE are irregular and semi-
diurnal; the mean tidal range is 0.86–1.7 m, increasing landward and reaching >3 m at Humen (Zhao,

1990).

Ranked the second largest in China and the thirteenth largest in the world (Zhang et al., 2008), the

Pearl River discharges a freshwater volume of 285 ×10$^9$ m$^3$ year$^{-1}$ to the South China Sea. The West
River is the largest tributary, contributing 73% of the Pearl River's total freshwater discharge, followed
by the North River (14%), and the East River (8%) (Wei and Wu, 2014). The PRE receives 50–55% of
the Pearl River's total freshwater flow from four major water outlets, namely Humen, Jiaomen,
Hongqimen, and Hengmen (Mikhailov et al., 2006), with Humen providing 35% of the freshwater
input to the PRE, followed by Jiaomen (33%), Hengmen (20%), and Hongqimen (12%) (Kot and Hu,
1995). About 70% to 80% of the freshwater discharge in the Pearl River occurs in the wet season
(April–September) and only 20–30% in the dry season (October–March) (Wei and Wu, 2014).
Freshwater in the PRE tends to flow seaward along the west side, while coastal saline water intrudes
landward along the east channel, causing large cross-estuary salinity gradients (Dong et al., 2004).
Seawater intrusion can reach 20–25 km downstream of Humen in the wet season and beyond Humen in
the dry season. The water column is strongly stratified during the wet season due to the large
freshwater input but well-mixed or far less stratified during the dry season (Wai et al., 2004; Ou et al.,

2009).





The Pearl River delivers $30.64 \times 10^6$ tons of sediment per year into the PRE, with 92–96% of this
discharge taking place during the wet season (Xu et al., 1985; Wai et al., 2004). The suspended
sediment concentration ranges from 40–300 mg $L^{-1}$ in the wet season and 20–190 mg $L^{-1}$ in the dry
season and reaches >100 mg $L^{-1}$ in turbidity maxima occurring at several locations of the estuary (Xu,
1985; Zhao, 1990; Wai et al., 2004).
Phytoplankton biomass in the PRE is generally higher in the wet season than in the dry season but
lower than expected from the high concentrations of dissolved inorganic nitrogen (Yin et al., 2000;
Harrison et al., 2008; Lu and Gan, 2015; Li et al., 2017). Phytoplankton blooms develop only on local
scales, usually in the mid-estuary during the dry season and in the lower part of the estuary during the
wet season (Lu and Gan, 2015).
Mountainous and hilly landscapes dominate the drainage basin of the Pearl River with almost no
forest (Luo et al., 2002), leading to relatively low dissolved organic carbon concentrations ([DOC])
(117–132 μmol $L^{-1}$) upstream of the Pearl River Delta (Shi et al., 2016). On the other hand, the Pearl
River Delta, a highly urbanized and industrialized region, delivers $5.8\times10^9$ tons of industrial and
domestic sewage per year into the PRE (Lu et al., 2009), which is considered the principal source of
DOC in the upper reach of the PRE (Lin, 2007; He, 2010). A number of studies have determined
[DOC] and the abundance of CDOM ([CDOM]) (in terms of fluorescence or absorption coefficients) in
the PRE (e.g. Dai et al., 2000; Callahan et al., 2004; Chen et al., 2004; Hong et al., 2005; He, 2010; Lei
et al., 2018). These studies show no consistent seasonality and estuarine mixing behavior of [DOC] and
[CDOM] and no correlation between the variables except one occasion for the mid-salinity (5–20)
section of the estuary (Callahan et al., 2004).
The lack of seasonality and consistent estuarine mixing behavior of [DOC] and [CDOM] suggests
complex processes controlling their transport, production, and loss in the PRE; it could, however, also
result in part from the difference in spatiotemporal coverage of the stations sampled by different



studies. As previous DOC and CDOM data were collected over a span of 12 and 15 years, respectively,
the possibility of interannual variability cannot be ruled out. In addition, none of the past DOC studies
surveyed all four seasons and many of them chose two different months to represent the wet and dry
seasons, though [DOC] and its mixing behavior may change on smaller time scales. The more limited
number of CDOM absorption surveys only sampled a single season with no winter visits. Concerning
the spatial coverage, some studies differ substantially in the distribution of sampling stations (e.g. Hong
et al., 2005 vs. Lei et al., 2018) and many did not cover the upper reach of the estuary (e.g. Chen et al.,
2003; Chen et al., 2004; Wang et al., 2014; Lei et al., 2018).
Compared with the quantitative information on DOC and CDOM, much less is known about the
seasonality and mixing behavior of their qualitative aspects. He et al. (2010) examined the DOC
compositions (monosaccharides vs. polysaccharides and dissolved free amino acids vs. dissolved
combined amino acids) along a longitudinal salinity-gradient transect in the PRE. Hong et al. (2005)
determined the fluorescence excitation-emission matrices (EEMs) on samples collected in the dry
season and suspected that CDOM in the PRE bears a microbial signature derived from sewage
effluents. Besides, spectral slope coefficient (Hong et al., 2005; Lei et al., 2018) and [DOC]-normalized
fluorescence intensity (Callahan et al., 2004) have been sporadically used to assess the quality of
CDOM in the PRE. Finally, only a few studies have estimated the DOC export flux from the Pearl
River to the South China Sea (Lin, 2007; Ni et al., 2008; He et al., 2010), often with limited seasonal
coverage. The estimate made by Lin (2007) is almost two times that by Ni et al. (2008). No estimates
of CDOM export are available.

**1.3 Hypothesis and objectives**
Given the large volume and seasonality of the freshwater discharge of the Pearl River, we
hypothesize that DOM in the PRE presents substantial seasonal variability in terms of both abundance



and chemical composition and that the PRE is an important source of DOM to global oceans. To test
this working hypothesis, the present study sampled the same locations in different seasons within a 12-
month period, with the objectives of 1) evaluating the seasonality and estuarine mixing behavior of
DOC and CDOM in the PRE both quantitatively and qualitatively; 2) improving the estimate of DOC
export to the South China Sea; 3) providing the first assessment of seaward export of CDOM in the
PRE based on absorption coefficient measurements. Results from this study further increase our
understanding of DOM cycling in human-impacted estuarine waters and their contribution to the DOC
and CDOM budgets in coastal oceans.

**2 Methods**
**2.1 Sample collection**
The sampling area covered the entire PRE, stretching from ~30 km upstream of Humen to the outer
limit of the estuary (Fig. 1). Ten stations (M01–M10) were distributed across the main longitudinal axis
of the estuary, together with two shorter along-estuary transects, each having four stations on the east
(E01–E04) and west (W01–W04) sides. The coordinates of the stations alongside other sampling
information are shown in Table 1. Water samples were collected in duplicate from the surface (~1m)
and near the bottom (1–2 m above the seabed) using a 5-L plexiglass sampler between 8–12 May, 7–11
August, 16–19 November 2015 and 10–14 January 2016 for [DOC] measurement and in the last three
seasons for CDOM analysis. The samples were filtered through 0.2-µm polyethersulfone (PES) filters
(Pall Life Sciences) under low vacuum and the filtrates were transferred into 20 mL (DOC) and 100
mL (CDOM) clear-glass bottles with Teflon-lined screw caps. DOC samples were acidified to pH ~2
with 2 N HCl (Reagent grade, Merck). All samples were stored in the dark at 4°C until being analyzed
in a land-based laboratory within two weeks after water collection. Prior to use, the glass filtration
apparatus and the sample storage bottles were acid-cleaned and combusted at 450°C for 4 h, and the



PES filters were thoroughly rinsed with Milli-Q water and sample water. Water temperature and
salinity were determined with a SBE-25 conductivity-temperature-depth (CTD) profiler.

**2.2 Sample analysis**
[DOC] was determined in triplicate using a Shimadzu TOC-L$_{CPH}$ analyzer calibrated with potassium
hydrogen phthalate. The performance of the analyzer was checked, at intervals of 10 consecutive
sample analyses, against Hansell's low carbon ([DOC]: 1–2 µmol L$^{-1}$) and deep Florida Strait ([DOC]:
41–44 µmol L$^{-1}$) reference waters. The coefficient of variation on five replicate injections was < 2%.
CDOM absorbance spectra were scanned from 800 nm to 200 nm at 1-nm intervals with a Shimadzu
UV-2550 dual beam spectrophotometer fitted with 10-cm quartz cells and referenced to Nanopure
water. The samples were allowed to warm up to room temperature in darkness before analysis. A
baseline correction was made by subtracting the mean absorbance value over 683–687 nm from all
spectral values (Babin et al., 2003). The Napierian absorption coefficient, $a_{CDOM}$ (m$^{-1}$), was calculated
as 2.303 times the absorbance divided by the light pathlength of the cell in meters (0.1 m). The
analytical uncertainty of $a_{CDOM}$ measurement was assessed by analyzing six pairs of duplicate samples
collected from the August cruise. Average $a_{CDOM}$ at 330 nm ($a_{330}$) was 2.19 m$^{-1}$ (range: 1.19–4.37
m$^{-1}$); the average difference in each pair was 0.07 ± 0.05 m$^{-1}$, or 3.0% ± 1.4%.
Fluorescence excitation-emission-matrices (EEMs) were acquired using a Hitachi F-4600
fluorescence spectrophotometer fitted with a 1-cm quartz cuvette to characterize the FDOM
composition (Coble, 1996; Boehme et al., 2004). Again, samples were warmed up to room temperature
before analysis. Emission spectra were scanned from 230 nm to 600 nm at 2-nm intervals over
excitation wavelengths between 200 nm and 450 nm at 5-nm increments. Raman scattering was
removed by subtracting Nanopure water EEMs that were scanned on the same day as those for the
samples. The spectral fluorescence intensities were normalized to Raman Units (R.U.) following the





Raman Scatter Peak correction reported by Lawaetz and Stedmon (2009). Potential inner-filtering
effects were corrected using the obtained absorbance spectra (Ohno, 2002), even though self-shading
should be insignificant since the absorption coefficient at 254 nm ($a_{254}$) was less than 15 m$^{-1}$ for all
samples. Major peaks in the EEMs were identified according to the peak definitions proposed in Coble

(2007).

PARAFAC analysis was performed to decompose the EEMs into a set of underlying fluorescent
components (Bro, 1997; Stedmon et al., 2003; Stedmon and Bro, 2008). The analysis was fed with 117
EEMs from all three seasons sampled for CDOM (Sect. 2.1). To reduce the dominance of high
fluorescence intensity signals, the EEMs were first scaled to a unit of variance within the sample mode
to construct the calibration model (Bro, 1997). PARAFAC models from 2 to 7 components with
constraints of non-negativity in all modes were successively conducted with MATLAB (version 2008b;
MathWorks 2008) using DOM Fluorescence Toolbox (DOM Fluor version 1.6) and validated as
described by Stedmon and Bro (2008). The parameters obtained from the PARAFAC model were used
to calculate an approximate abundance of each component, expressed as $F_{max}$ in Raman's unit (R.U.),
which corresponds to the maximum fluorescence intensity for a particular sample. Based on analysis of
triplicate samples from Sta. M01, M08, and M10, the uncertainty of $F_{max}$ for each modeled component
was <2%.

**3 Results**
For brevity, seasons and/or water layers for a property are added as a superscript to the symbol or
abbreviation denoting that property. For example, [DOC]$^{\text{surf/Aug}}$ stands for [DOC] in surface water in
August and $a_{330}^{\text{btm/Jan}}$ for CDOM absorption coefficient at 330 nm in bottom water in January. The
slope of linear regression of a property against salinity will be denoted by SLP (Sect. 3.4), with a
superscript added to designate the specific variable and/or season. For instance, SLP$^{\text{[DOC]/May}}$ denotes



the slope of [DOC] vs. salinity in May. [DOM], [CDOM], and [FDOM] stand for the abundances of
DOM, CDOM, and FDOM, respectively, and names of PARAFAC components signify their $F_{max}$ as
well. Finally, symbols and abbreviations are used as both singular and plural forms.

### 3.1 Hydrological settings

The average freshwater discharges of the Pearl River for the sampling months were obtained from
the Ministry of Water Resources of the P. R. of China (http://www.mwr.gov.cn/zwzc/hygb/sqnb).
Assuming that 54% of the total discharge of the Pearl River went into the PRE (Mikhailov et al., 2006)
giving $8.9 \times 10^3$ m$^3$ s$^{-1}$ in May, $5.7 \times 10^3$ m$^3$ s$^{-1}$ in August, $6.7 \times 10^3$ m$^3$ s$^{-1}$ in November, and $5.0 \times 10^3$
m$^3$ s$^{-1}$ in January. The freshwater discharge was 15% lower in August than in November due to an
atypically dry weather in summer and a relatively higher precipitation in autumn. The precipitation in
January was also above average, leading to a higher-than-normal freshwater discharge in that month.
Surface water temperature averaged 27.2°C in May, 30.0°C in August, 25.2°C in November, 18.8°C
in January. Water temperature in August was higher in the inner than in the outer estuary, whereas a
reverse trend was seen in the other sampling seasons (Fig. S1). Cross-estuary gradients occurred in all
four seasons often with irregular patterns. Yet, the east transect showed the highest temperatures in
November and the west transect displayed the lowest temperatures in January. The difference between
the surface and bottom water was minor in January (0–1.5%) and minor to moderate in May (0–11.9%)
and November (0.08–2.5%) except a few stations near the mouth of the estuary. In August, the bottom
temperature was substantially lower (3–14%) than the surface temperature at many stations and the
difference increased towards the sea.
Surface water salinity (SWS) ranged from 0.2–30.3 (mean: 9.7) in May, 0.2–20.6 (mean: 8.0) in
August, 0.2–26.9 (mean: 8.3) in November, and 0.2–32.6 (mean: 17.0) in January. SWS was very low
(range: 0.15–0.66) and remained fairly constant upstream of Sta. M05 in May and November (Fig. S2).




Saltwater intruded farther upstream to Sta. M03 in January, in line with the lower tides (Fig. S3) and
lower freshwater discharge at that time. Despite August showing the lowest estuary-wide mean SWS
among the four seasons, its SWS values in the upper reach of the estuary (Sta. M01–M05) were
considerably higher than those in May and November, and the value at Sta. M02 even surpassed that in
January. This phenomenon could be partly attributed to most stations in the upper reach being sampled
at high tides in August (Fig. S3). Seaward of the upper reach, SWS increased rapidly, albeit with
fluctuations likely linked to tidal cycles and passage of salinity fronts (Dong et al., 2004). Consistent
with published results (Dong et al., 2004), SWS exhibited cross-estuary gradients, often increasing
from the shallow water on the west side to the deeper water on the east side, which was particularly
evident in January and May (Fig. S2). Bottom water salinity at most stations was nearly identical to
SWS in January, slightly greater in May, moderately elevated in November, and much higher in August
(Fig. S2). Based on the salinity difference between the two layers, the water column was mostly well
mixed in January, weakly stratified in May, modestly stratified in November, and strongly stratified in
August. Remarkable exceptions were certain shallow stations along the west (Sta. W01–W03) and east
(Sta. E01) transects at which the water column was well mixed in November and May and weakly
stratified in August. In addition, the water column in the low-salinity zone (Sta. W01–W05 in May and
November; Sta. W01–W02 in August and January) was essentially homogenous in all four seasons.

**3.2 Distributions of quantitative DOM variables**
The quantitative DOM variables reported here are [DOC], $a_{CDOM}$, and PARAFAC-derived FDOM
components. $a_{330}$ is chosen as an indicator of [CDOM] (Osburn et al., 2009; Xie et al., 2012; Song et
al., 2017). $a_{CDOM}$ at other commonly used wavelengths and the spectral slope coefficient between 300
nm and 500 nm are presented in Table S1. The residual and split-half PARAFAC analyses validated
five distinct FDOM components (Fig. 2), which explained 99.75% of the variance and thus adequately





modeled the different FDOM profiles in the dataset. Based on published spectral characteristics of
PARAFAC-modeled components (e.g. Stedmon et al., 2003; Cory and McKnight, 2005; Yamashita
and Jaffé, 2008; Murphy et al., 2008; Santín et al., 2009; Massicotte and Frenette, 2011), components 1
(C1) and 5 (C5) as tyrosine-like and tryptophan-like fluorophores, components 2 (C2) and 4 (C4) were
assigned as terrestrial or ubiquitous humic-like DOM fractions, and component 3 (C3) as oxidized
quinone-like moieties, respectively. As C1 and C2 are highly correlated with C5 and C4, respectively
(Table S2), the sum of C2 and C4 (C2+C4 hereafter) and of C1 and C5 (C1+C5 hereafter) will be used
to describe the quantitative distributions of the humic-like and protein-like fractions. C3 is better
correlated to C2+C4 ($R^2 = 0.953$) than to C1+C5 ($R^2 = 0.738$), suggestive of its humic character.
Table 2 summarizes the ranges and averages of all quantitative variables in different seasons and
water layers. The mean $[DOC]^{surf}$ was in descending order of May > January > August > November.
The mean $a_{330}^{surf}$, $(C2+C4)^{surf}$, and $C3^{surf}$ exhibited the same seasonality of August > November >
January, differing from that of [DOC]. The seasonal trend of the mean $(C1+C5)^{surf}$ followed November
> January > August, inconsistent with the two trends noted above. The seasonality in bottom water was
the same as that in surface water for the mean values of all these variables save $a_{330}$ which was equal
between November and January in the bottom. The mean values in bottom water were lower than in
surface water for all variables and all seasons (Table 2). The seasonal trend of the absolute percent
difference between the bottom and surface followed August > May > November > January for [DOC]
and August > November > January for the CDOM and FDOM variables, conforming to the
successively weakening water column stratification from summer to autumn to winter (Sect. 3.1). The
average vertical differences ranged from 6.5–21.0% in August, 1.0–11.9% in November, and 0.1–5.5%
in January depending on the variable in question, with [DOC] showing the smallest disparities (Table
2). Despite the overall small vertical gradients, certain stations, often with the deepest water depths
(Table 1), did exhibit larger differences (>20%, Fig. S4).



All variables displayed similar along-estuary distribution patterns characterized by overall declining
abundances with increasing seaward distance (Figs. S4,5). Two features are noted here. First, [DOC] in
May remained nearly constant from Sta. M01 to M03, consistent with the observation of He et al.
(2010) in April 2007. As Sta. M01–M03 all had near-zero salinities (0.18–0.27, Fig. S2) and were
distributed across the three entrances of the East River (Fig. 1), freshwater input from this river
appeared to have little influence on [DOC] in May. Second, the declines of $[DOC]^{surf}$ across the entire
main transect (Fig. 1), 40–42% in May, August, and November, and 54% in January, were
considerably lower than those of $a_{330}$ and C1+C5, 70–74% in August, 80–84% in November, and 92–
93% in January. The parallel declines of C3 and C2+C4 were somewhat inferior in August (53–57%)
but comparable in November (72–76%) and January (92%).
Unlike the substantial cross-estuary salinity gradients noted earlier, lateral variations in [DOC] were
generally small in all seasons, with one prominent exception at ~54 km downstream of Sta. M01,
where [DOC] in May on the east transect (Sta. E01: 192 µmol $L^{-1}$) was 47% higher than that on the
main transect (Sta. M05: 131 µmol $L^{-1}$; Fig. S4a). Systematic cross-estuary variations in $a_{330}$ (Fig.
S4e–g) and FDOM components (Fig. S5) were not evident in August and November, while values in
January were consistently higher along the west transect (Fig. S4g, Fig. S5c,f,i) echoing the
substantially lower salinities on the west side (Fig. S2d). Large lateral differences in C1+C5 and
C2+C4 were again observed between Sta. E01 on the east transect and Sta. M05 on the main transect in
November (Fig. S5b,e).

**3.3 Distributions of qualitative DOM metrics**
The qualitative metrics reported here are the $E_2/E_3$ quotient (hereafter $E_2/E_3$), biological index
(BIX), humification index (HIX), and percentages of C1+C5 (%(C1+C5)), C2+C4 (%(C2+C4)), and
C3 (%C3) relative to the sum of C1–C5. $E_2/E_3$, defined as the ratio of $a_{250}$ to $a_{365}$, serves as a proxy for





the average molecular weight (MW) and aromaticity of CDOM, with lower values indicating higher
MW and higher aromaticity (Peuravuori and Pihlaja, 1997; Lou and Xie, 2006; Li and Hur, 2017).
BIX, the ratio of fluorescence intensity at 380 nm to that at 430 nm with excitation at 310 nm, indicates
the relative contribution of fresh, autochthonous FDOM; higher BIX values signify higher
contributions of freshly produced FDOM of microbial origin (Huguet et al., 2009). HIX, the ratio of the
fluorescence intensity integrated over 435–480 nm to that over 300–345 nm with excitation at 254 nm,
is a surrogate of the extent of FDOM humification, with higher values denoting higher degrees of
humification (Ohno, 2002). %(C1+C5), %(C2+C4), and %C3 represent the relative contents of protein-
like, humic-like, and quinone-like components in the total FDOM pool.
Table 3 summarizes the ranges and averages of all qualitative metrics for each sampling season and
water layer. Bottom–surface differences were minor in all seasons and for all metrics. C1+C5 on
average accounted for 50.2–66.4% of the total FDOM components and thus exceeded C2+C4 and C3
in all seasons. The seasonal trends of HIX and %(C2+C4) were both August > November > January
and opposite to those of $E_2/E_3$, BIX, and %(C1+C5) (i.e., January > November > August). Note that the
difference in $E_2/E_3$ was marginal between August and November and in all other metrics between
November and January. %C3 was highest in August and equal between November and January.
$E_2/E_3$ in all three seasons increased gradually down-estuary (Fig. S6a–c) by up to 59% in August,
60% in November, and 76% in January. BIX in August and November dropped briefly within the first
10 km and then augmented slowly farther seaward; it remained, however, roughly constant in January
(Fig. S6d–f). Mild convex curves with maxima located at mid-estuary characterized the longitudinal
HIX distributions in all three seasons (Fig. S6g–i). %(C2+C4) presented an along-estuary distribution
pattern (Fig. S7d–f) similar to that of HIX and inverse to that of %(C1+C5) (Fig. S7a–c). With a few
exceptions, %C3 increased nearly monotonously from land to sea irrespective of seasons (Fig. S7g–i).
Cross-estuary gradients were generally minor (Figs. S6 and S7). An important exception was the
west transect giving lower $E_2/E_3$ in January, lower %(C1+C5) in August, and higher HIX and



%(C2+C4) in both August and January, with the gradients all diminishing seaward. Bottom–surface
differences at individual stations were mostly marginal (<10%). Certain stations, particularly those
with relatively deeper water depths (Table 1), showed considerably larger differences (>20%), as noted
in Figs. S6 and S7.

**3.4 Relationships between quantitative DOM variables and salinity**
Surface and bottom data for each variable in each season form a consistent property–salinity pattern
(data not shown) and are thus treated as a single dataset. All variables displayed large variations and/or
sharp decreases at salinity <5 but remained rather constant ([DOC] in May and November) or declined
linearly (all other cases) at salinity >5 (Figs. 3 and 4). Results of linear regressions for the saltier zone
are summarized in Table 4. At a 95% confidence level, both the slopes (SLP) and intercepts are
statistically no different between August and January for [DOC] and $a_{330}$ and between all three seasons
for C3 and C2+C4, indicating that the multi-season data on each of these occasions can be combined
into a single dataset. $SLP^{a330/Nov}$ is, however, ≥32% lower than those in August and January. $SLP^{C1+C5}$
presents significant seasonal variations, with the value in January 23% and 89% higher than those in
November and August, respectively.
The percent decrease of each variable per unit increase of salinity across the saltier zone was
calculated using the known regression equations shown in Table 4. $a_{330}$ decreased 2.1 and 2.7 times
faster than [DOC] in August and January, respectively (Table S3). [FDOM] declined faster than
[CDOM] but their difference was much smaller than that between [CDOM] and [DOC]. The percent
decreases in the FDOM components were 5–35% higher than those in $a_{330}$, with November showing
the largest difference (25–35%) followed by August (5–21%) and January (<10%) (Table S3).

**3.5 Relationships between qualitative DOM metrics and salinity**



As for the quantitative variables, the surface and bottom data of the qualitative metrics can also be
treated as a single dataset in relation to salinity (data not shown). In August and November, $E_2/E_3$
increased with salinity quickly in the restricted low-salinity section (salinity: <1.3) and slowly in the
saltier zone (Fig. 5). In January, the surge at low salinities was less obvious and a gradual rise of $E_2/E_3$
in the saltier zone was observed up to salinity 28.5 beyond which the trend curved up. All three seasons
gave consistent $E_2/E_3$ vs. salinity patterns from salinity 1.3 to 28.5. In the saltier zone, the data for each
season roughly followed the respective theoretical mixing line defined by the maximum- and
minimum-salinity $E_2/E_3$ values in the corresponding season (Fig. 5).
Between salinity 0 and 5, %(C1+C5) in August decreased by approximately 10% (Fig. 6a). In the
saltier zone, the west transect displayed an increasing %(C1+C5) with salinity but was constantly
below the main and east transects which formed a coherent %(C1+C5) vs. salinity pattern featured by a
rebound from salinity 3 to 13 and a continuous decline at salinity >13. In November, %(C1+C5)
between salinity 0 and 10 (63.9% ± 5.8%) is more scattered than that for salinity from 10 to 27 (65.1%
± 2.1%) but the average values for the two sections are very similar. %(C1+C5) for the two most
marine samples dropped by 5–10% compared with the average over salinity 10–27 (Fig. 6a). A pan
shape characterized the distribution of %(C1+C5) in January, revealing higher %(C1+C5) values at
both the lowest and highest salinities and relatively lower values across a wide range of salinities in
between (Fig. 6a). The distributions of %(C2+C4) vs. salinity approximately mirrored those of
%(C1+C5) (Fig. 6b). %C3 in August decreased with salinity along the west transect whereas it
increased linearly ($Y = 0.19*X + 15.61$, $R^2 = 0.867$, n = 28) along the main and east transects
combined (Fig. 6c). %C3 in January stayed rather constant (13.5% ± 0.8%) until an abrupt 14% decline
at salinity >32. The distribution of %C3 in November resembled that of %(C2+C4).
Except a few larger scatters at the lowest (November) and highest (November and January)
salinities, BIX displayed little dependence on salinity in all three seasons (Fig. 7a). The HIX vs. salinity
patterns (Fig. 7b) corresponded to those of %(C2+C4), leading to a strong linear relationship between



the two variables (Fig. S8a). HIX is also positively correlated to %C3 (Fig. S8b), despite a weaker
correlation than that of HIX to %(C2+C4), again suggestive of the humic character of C3 (Sect. 3.2).
No significant correlation was seen between BIX and %(C1+C5) (Fig. S8c).
In spite of the certain seasonal and spatial variations of the qualitative metrics noted above, the
overall changes of these variables in the saltier zone, after excluding several extreme values for $E_2/E_3$
and BIX, were fairly limited, ranging from 4.8–9.1, 0.94–1.36, 0.54–2.04, 43.4%–70.3%, 16.5%–
35.9%, and 10.4%–22.6% for $E_2/E_3$, BIX, HIX, %(C1+C5), %(C2+C4), and %C3, respectively.

**3.6 Relationships between [DOC] and $a_{CDOM}$ and FDOM fluorescence**
[DOC] was linearly correlated to $a_{330}$ for all three sampling seasons; the coefficient of determination
was, however, lower in November (Fig. 8a, Table 5). The fitted slope was in descending order of
January (32.0 ± 2.0 m µmol $L^{-1}$) > August (22.5 ± 1.4 m µmol $L^{-1}$) > November (18.8 ± 2.2 m µmol
$L^{-1}$). Similarly, [DOC] showed a strong, linear correlation with C1+C5 in August and January and a
relatively weaker one in November (Fig. 8b, Table 5). The fitted slopes in August and January were
comparable but ~2.8 (2.7–2.9) times that in November (Table 5). [DOC] was also significantly related
to C2+C4 and C3 (Fig. 8c,d) but the correlations were considerably weaker than that with C1+C5
(Table 5).

**4 Discussion**
**4.1 Sources of freshwater DOM endmembers**
Large variations in [DOC] and [CDOM] in the freshwater section of the PRE have been observed
previously (Chen et al., 2004; Lin, 2007; He, 2010; Wang et al., 2014; Lei et al., 2018). The present
study confirmed this phenomenon in August ([DOC] only) and November ([DOC], [CDOM], and
[FDOM]) when near zero-salinity (< 0.7) water was accessible down to Sta. M05 off Hongqimen (Fig.
1). This hefty fluctuation in DOM content is commonly ascribed to the presence of multiple freshwater



endmembers delivered by various water channels and outlets described in Sect. 1.2 (Cai et al., 2004;
Callahan et al., 2004; He et al., 2010). Because Humen holds most of the sewage discharge from
Guangdong Province (Pang and Li, 2001), which carries the highest DOM load, while the other
waterways on the west coast, less influenced by urbanization and industrialization, bear lower levels of
DOM (Callahan et al., 2004; Ni et al., 2008).

Although the existence of multiple "quantitative" endmembers in the PRE is well recognized, it

remains poorly understood if these endmembers differ qualitatively. Data published by Callahan et al.
(2004) shows that [DOC]-normalized fluorescences of the freshwater endmembers in Jiaomen,
Hongqimen, and Hengmen differed little (c.v. = 4%) while the Humen endmember was 17% higher
than the mean of the other three endmembers in November 2002. Besides, fluorescence EEMs
collected upstream of Humen reveal tryptophan-like fluorophores to be the dominant FDOM fraction in
the Humen endmember which was suspected to originate from sewage effluents (Hong et al., 2005).
The present study has analyzed by far the largest number of qualitative metrics (i.e. $E_2/E_3$, relative
abundances of FDOM components, BIX, and HIX) and thus offers a more comprehensive means to
assess the nature of the freshwater endmembers. $E_2/E_3$ in near zero-salinity samples fell in a rather
small range from 5.5 to 6.8 that corresponded to a MW range from 0.83 kDa to 1.18 kDa estimated
from the MW vs. $E_2/E_3$ relationship proposed by Lou and Xie (2006). The higher MW values were
observed in the Humen channel, while the lower ones in water from Jiaomen and Hongqimen, both
being close to the borderline separating the high- and low-MW CDOM (i.e. 1 kDa). %(C1+C5) varied
from 70% at Sta. M01 in the Humen channel to 56% off Hongqimen, consistent with a stronger
signature of anthropogenic DOC in the Humen channel (He et al., 2010). Yet %(C1+C5) for all
endmembers were >50%, demonstrating that protein-like components dominated all freshwater FDOM
endmembers. BIX was slightly higher while HIX lower at Sta. M01 than at Sta. M05 (BIX: 1.28 vs.
1.00; HIX: 0.53 vs. 1.34); all BIX and HIX values were, however, well above 0.8 and below 5,
respectively, implying the dominance of fresh, microbial-derived FDOM in all freshwater endmembers



(McKnight et al., 2001; Birdwell and Engel, 2010; Sazawa et al., 2011). Taking into account all these
qualitative metrics and the linear correlations between [DOC] and the FDOM components (Sect. 3.6),
we can conclude that all three freshwater DOM endmembers in November mainly comprised fresh,
relatively low-MW (~1 kDa) organic material of microbial origin, with the microbial nature in the
Humen endmember somewhat stronger. The sewage influence could be depressed due to a rapid
bacterial mineralization of the sewage-derived DOM between the point sources of pollution in the
Guangzhou area and the sampling stations downstream (He et al., 2010). Note that the three
endmembers also bore a perceptible terrigenous character, since the humic-like C2 and C4, albeit lower
in abundance than the protein-like C1 and C5, were still a significant fraction of the total FDOM pool
(Fig. 6). The values of the qualitative metrics at Sta. M01 in August and January were comparable to
those in November (Figs. S6,7), indicating that the Humen DOM endmembers in summer and winter
were also of microbial origin.
Based on an estimate of the relative contributions of land-, sewage-, and phytoplankton-derived
DOC, He (2010) and He et al. (2010) proposed that the land component is the dominant source of the
total DOC pool in the lower reach of the Humen channel. In this estimation, the authors assigned the
"natural background" [DOC] in the three major tributaries of the Pearl River (range: 114–125 μmol
$L^{-1}$; mean: 119 μmol $L^{-1}$) as "land-derived". Our result suggests that, apart from terrigenous DOC
leached from soil, this "land-derived" DOC contains an ample amount of river-born DOC of microbial
origin. This is consistent with the poorly forested watershed of the Pearl River (Luo et al., 2002) and
with the study of Ni et al. (2008) showing the molar carbon-to-nitrogen ratios of suspended particulate
organic matter in all major runoff outlets of the Pearl River Delta (7.2–9.3) to be close to those for
phytoplankton and bacterial biomass (5–8).

**4.2 Estuarine mixing and transformation of DOM**



Sharp decreases in [DOC], [CDOM], and [FDOM] in the low-salinity section of the PRE have been
previously observed and postulated as a result of adsorption, flocculation, biodegradation, and/or
incomplete mixing of multiple freshwater endmembers (Callahan et al., 2004; Chen et al., 2004; Lin,
2007; He et al., 2010). The present study confirmed the earlier observations but more importantly
provided additional qualitative metrics that are instrumental for constraining the principal processes
causing this swift drawdown of [DOM]. The increases in %(C2+C4) and HIX and decreases in
%(C1+C5) and BIX in the low-salinity section (Figs. 6 and 7) indicate a bacterial preferential uptake of
protein-rich materials and hence a key role of biodegradation in controlling the loss of DOM. Our
result corroborates the finding of He et al. (2010) showing higher fractions of biodegradable DOC and
higher DOC bio-uptake rates in the low-salinity section than in the saltier zone. Note that the more
scattering of the qualitative metrics data in November (Figs. 6 and 7) likely reflects an incomplete
mixing of the multiple freshwater endmembers stated earlier. This partial-mixing effect may smear or
even entirely overshadow the biodegradation signal.
In the saltier zone, the linear decreases in [DOC] (see exceptions below), [CDOM], and [FDOM]
with salinity point to the absence of net removal and input of these constituents and physical dilution
being the principal mechanism dictating their estuarine mixing behaviors. The two extreme cases of
near-constant [DOC] vs. salinity in May and November indicate that the loss of DOC in the low-
salinity section reduced its content to the level comparable to the marine endmember and again that the
removal of DOC in the saltier zone, if any, was roughly balanced by the input. Potentially important
DOM loss processes in the PRE are bacterial (He et al., 2010) and photochemical (Callahan et al.,
2004) degradation. The significance of these processes relies on both their rates and the residence time
of freshwater in the PRE. Using the volume of the estuary ($9.6 \times 10^9$ m$^3$) and the freshwater discharge
rate for each sampling season (Sect. 3.1), we estimated the residence time of freshwater in the top 1-m
layer to be 3.1 d in May, 4.9 d in August, 4.1 d in November, and 5.6 d in January. The value for May
is essentially identical to that previously reported for the wet season (Yin et al., 2000). Here the volume



of the estuary was obtained from the published average depth (4.8 m) and total area ($2 \times 10^9$ m$^2$) of the
estuary (Dong et al., 2004). The bacterial uptake rate of DOC in surface water of the saltier zone has
been reported to be 0.04 µmol L$^{-1}$ h$^{-1}$ in spring and 0.07 µmol L$^{-1}$ h$^{-1}$ in summer, giving a consumption
of 3.0 µmol L$^{-1}$ and 8.2 µmol L$^{-1}$, respectively, when multiplied by the corresponding residence time
for May and August. Our unpublished data indicates that photodegradation in August could at most
reduce [DOC] by 0.76 µmol L$^{-1}$ and $a_{330}$ by 0.11 m$^{-1}$, after considering the attenuation of solar
radiation and the competition for light absorption by particles in the water column (Wang et al., 2014).
The combined photochemical and bacterial DOC degradation in summer was thus ~9 µmol L$^{-1}$, ~8% of
the initial [DOC] in the saltier zone. The parallel photobleaching loss of $a_{330}$ was 7%. Such small losses
could be readily compensated for by DOM input from in situ primary production, sediment
resuspension, and/or freshwater discharge farther downstream. Notably, chlorophyll *a* concentration
maxima of up to 11.0 µg L$^{-1}$ and turbidity maxima of up to 154 mg L$^{-1}$ were spotted in the mid- and
lower estuary during our cruises (Xu et al., unpublished data). There was, however, no co-variation of
[DOM] (i.e. [DOC], [CDOM], and [FDOM]) with chlorophyll *a* or suspended particle concentration
(data not shown). This observation, in conjunction with the linear [DOM] vs. salinity relations,
demonstrates that autochthonous production was unlikely a major source of DOM and that adsorption
and flocculation were not a major sink of DOM in the saltier zone. The short residence time of
freshwater likely minimized the influences of these processes.
The completely different behaviors of [DOC] and [CDOM] with respect to salinity in the saltier
zone in November (Fig. 3c,f) led to a decoupling of the two variables. This phenomenon has also been
observed for summer by Chen et al. (2004). In fact, the disconnection of [DOC] and [CDOM] is an
extreme case of the higher salinity-based [CDOM] gradient relative to that of [DOC] seen in August
and January (Sect. 3.4). The difference in estuarine mixing behavior between [DOC] and [CDOM]
arose mainly from two factors. First, a large portion of the freshwater DOM endmember was non-
and/or weakly colored, as implied by its abundant fresh microbial constituents. Second, the marine



endmember's [DOC]-normalized $a_{CDOM}$ was lower than the freshwater endmember's: 0.60 vs. 2.18 L
$mg^{-1}$ $m^{-1}$ in August, 0.71 vs. 2.32 L $mg^{-1}$ $m^{-1}$ in November, and 0.26 vs. 1.71 L $mg^{-1}$ $m^{-1}$ in January at
330 nm.
The overall small variations of the qualitative metrics (Figs. 5–7 and Sect. 3.5) across the saltier
zone suggest that the chemical composition of CDOM and FDOM remained generally stable during
estuarine mixing, consistent with the marginal photochemical and microbial breakdown of DOM
elaborated above. The higher values of %(C2+C4), %C3, and HIX in August than in November and
January (Figs. 6 and 7) point to FDOM in summer containing a larger fraction of humic-like
fluorophores. The divergence in August of the west transect from the main and east transects with
respect to the FDOM metrics (save BIX) distributions vs. salinity (Figs. 6 and 7) suggests a different
freshwater mass on the west shoal enriched with humic-like FDOM and possibly originating from
Hengmen (Fig. 1).

**4.3 Indicators of $a_{CDOM}$ and [DOC] in the saltier zone**
Salinity is a useful proxy of $a_{CDOM}$ in light of their linear relationships in the saltier zone for all three
sampling seasons (Fig. 3). Furthermore, a common equation (Y = −0.048*X + 1.99) can serve as a
predictive tool of $a_{330}$ in August and January, given essentially the same statistics for each of these two
months (Table 4). For [DOC], salinity can be used as an indicator in August and January but not in
May and November (Fig. 3). Similar to the $a_{CDOM}$–salinity case, the August and January [DOC] data
can be combined to formulate a single [DOC]–$a_{CDOM}$ relationship (Y = 40.7*X + 75.6). Hence, [DOC]
in summer and winter can in principle be retrieved from remote sensing-based $a_{CDOM}$ data (Siegel et al.,
2002; Johannessen et al., 2003; Mannino et al., 2008). Absorption coefficients and fluorescence
intensities at the excitation and emission maximum wavelengths of C1 and C5 are also good indicators
of [DOC] in August and January (Fig. 8).



Caution should be exercised when applying the [DOC] and $a_{CDOM}$ predictive tools established here,
since interannual variability and other factors may limit their applicability on broader time and space
scales. For example, Hong et al. (2005) arrived at an $a_{CDOM}$–salinity relationship of $a_{355}$ =
−0.045*salinity + 1.81 for November 2002, which is different from ours in the saltier zone ($a_{355}$ =
−0.021*salinity + 0.98). Concurrent measurements of [DOC] and $a_{CDOM}$ in the PRE are rare but Chen
et al. (2004) reported no significant correlation between the two variables in July 1999.

**4.4 Fluxes of DOC and CDOM**
The fluxes of DOC and CDOM exported from the PRE to the South China Sea were estimated as
follows (Cai et al., 2004; Lin, 2007; He et al., 2010):
$F = Q \times C^*$                (1)
where F denotes the flux of DOC or CDOM, Q the freshwater discharge rate, $C^*$ the effective [DOC]
([DOC]$^*$) or $a_{330}$ ($a_{330}^*$). $C^*$ is the y-axis intercept of the regression line of [DOC] or $a_{330}$ vs. salinity in
the saltier zone (Table 4). For May and November when [DOC] remained roughly constant across the
saltier zone, $C^*$ signifies the average [DOC] over this zone. Monthly fluxes were computed using
freshwater discharge rates for the sampling year and those averaged over 2006–2016
(http://www.mwr.gov.cn/zwzc/hygb/sqnb), under the assumption that the [DOC] or $a_{330}$ obtained for
May, August, November, and January represents the entire spring (March, April, May), summer (June,
July, August), autumn (September, October, November), and winter (December, January, February),
respectively. As no CDOM data was collected in May, the $a_{330}^*$ for spring (1.99 ± 0.19 m$^{-1}$) was
derived from the mean of the [DOC]$^*$-normalized $a_{330}^*$ in January (1.31 L mg$^{-1}$ m$^{-1}$) and August (1.36
L mg$^{-1}$ m$^{-1}$) multiplied by the [DOC]$^*$ in May (124.5 μmol L$^{-1}$). This treatment, with unknown
uncertainties, was based on the relatively small variations of the [DOC]$^*$-normalized $a_{330}^*$ among the
three CDOM sampling seasons (range: 1.31–1.50 L mg$^{-1}$ m$^{-1}$).



Flux estimates for the sampling year are comparable to those for the 10-year period for spring and
summer, whereas the former is approximately twice the latter for autumn and winter due to above-
average freshwater discharge rates during the low-flow season of the sampling year (Table 6).
Aggregation of the fluxes for all four individual seasons arrives at an annual export of $240 \times 10^9$ g C
(sampling year) or $195 \times 10^9$ g C (10-year period) for DOC and of $329 \times 10^9$ m$^2$ (sampling year) or 266
$\times 10^9$ m$^2$ (10-year period) for CDOM in terms of $a_{330}$. As the PRE receives ~54% of the total Pearl
River freshwater discharge to the South China Sea (Mikhailov et al., 2006), including the rest 46%
gives a grand annual export of $362 \times 10^9$ g C of DOC and $493 \times 10^9$ m$^2$ CDOM, respectively, assuming
that the fluxes from the PRE are applicable to the entire Pearl River Delta.

**4.5 Comparison with previous studies and other major estuaries**
The [DOC]s obtained by this study in all four seasons are within the ranges previously reported for
the PRE (Table 7). DOC stock in the PRE thus did not seem to undergo large changes over the 7-year
span from the last survey in 2008 to our study in 2015, suggesting that the gross inputs and losses of
DOM remained stable during this period. Compared to DOC, previous $a_{CDOM}$ measurements are far
fewer and none of them was made in wintertime. The summer and autumn $a_{330}$ values from this study
are, however, comparable to those published (Table 7). Our DOC flux estimate for spring 2015 (5.8 ×
$10^8$ g C d$^{-1}$) is close to that reported by He et al. (2010) for spring 2007 ($5.3 \times 10^8$ g C d$^{-1}$). The
summer 2015 value ($9.0 \times 10^8$ g C d$^{-1}$) is, however, only 60% of the summer 2007's (He, 2010) due to
a much lower river runoff in 2015 (7174 m$^3$ s$^{-1}$ vs. 25060 m$^3$ s$^{-1}$). The DOC flux for the entire Pearl
River Delta estimated by this study ($362 \times 10^9$ g C year$^{-1}$) is comparable to that ($380 \times 10^9$ g C year$^{-1}$)
reported by Ni et al. (2008) but 44% lower than that ($650 \times 10^9$ g C year$^{-1}$) obtained by Lin (2007). The
estimate by Ni et al. (2008) was based on monthly [DOC] measurements at eight major runoff outlets
of the Pearl River Delta from March 2005 to February 2006. Lin (2007) derived the estimate from data





collected during three cruises carried out in winter (February 2004), early spring (March 2006), and
summer (August 2005). Part of the difference between our study and Lin's could result from the
different temporal coverages. The main difference, however, stems from the much greater $[DOC]^*$
obtained by Lin (2007) (147 µmol L$^{-1}$ for the wet season and 254 µmol L$^{-1}$ for the dry season).

[DOC] and [CDOM] in the PRE are the lowest among the major world rivers (Table 8). The low

DOM load in the PRE could be associated with a deficiency of organic matter in soil of the Pearl
River's watershed having almost no forest (Luo et al., 2002). Moreover, although sewage effluents may
bring in large amounts of DOM, a big portion of it can be rapidly bio-degraded before reaching the
head of the estuary (He et al., 2010). The lack of correspondence between $[DOC]^*$ and $a_{330}^*$ and the
freshwater discharge rate (Fig. S9) suggests that [DOM] in the PRE be controlled by both soil leaching
and pollution input. In contrast, DOM in the majority of large rivers is predominantly terrigenous
(Bianchi, 2011; Raymond and Spencer, 2015) and the abundance of DOM in many rivers increases
with the river flow rate (Cooper et al., 2005; Holmes et al., 2013). Note that the absence of a link
between [DOC] and the freshwater discharge rate in the PRE observed by this study differs from the
anti-covariation of the two variables reported by Lin (2007) and Ni et al. (2008). Based on this anti-
variation, Lin (2007) proposed that the PRE is a typical point source-regulated system in terms of DOC
concentration and distribution. It remains to be confirmed if our results imply a fundamental shift of the
PRE within ~10 years from a pollution-dominated system to a system jointly controlled by pollution
and soil flushing.

Owing mainly to the very low [DOC], our DOC export estimate for the Pearl River is the lowest

among the 30 largest rivers worldwide (Raymond and Spencer, 2015), though the Pearl River is ranked
the 13th largest river by discharge volume. The Pearl River value of $362 \times 10^9$ g C year$^{-1}$ only accounts
for 0.14% of the global riverine DOC flux estimate of $250 \times 10^{12}$ g C year$^{-1}$ (Raymond and Spencer,
2015). The estimate for CDOM export from the Pearl River is also the lowest among the limited





number of estimates available for the major world rivers (Table 9). It is worth noting that, despite its
small contribution on global scales, DOM delivered by the Pearl River is rich in labile, proteinaceous
constituents that can be readily utilized by microbes, thereby exerting a potentially important impact on
the local ecosystem.

**5 Conclusions**
The saltier zone of the PRE manifests smaller temporal variabilities and lower spatial gradients of
DOM than expected for a sizable estuary with a marked seasonality of river runoff. Several factors
functioning in concert lead to this phenomenon. First, a combination of the poorly forested watershed,
rapid degradation of pollution-derived DOM in the upper reach, and short residence time of freshwater
diminishes [DOM] and seasonal variations in both DOM quantity and quality. Second, the small
difference between the low-salinity and marine DOM endmembers tends to lessen the vertical and
lateral gradients in DOM again both qualitatively and quantitatively, notwithstanding the larger vertical
and cross-estuary salinity gradients. Both the concentrations and seaward exports of DOC and CDOM
in and from the PRE are the lowest among the major world rivers. However, as DOM undergoes
marginal processing during its transit through the estuary, the Pearl River delivers protein-rich, labile
organic matter to the continental shelf of the South China Sea where it may fuel heterotrophy.

*Author contributions.* GS and HX designed the study. HX and GS interpreted the results and prepared
the manuscript with input from PM. YL performed sample analysis and data processing. YL, GS, FY,
and RL carried out field sampling. PM conducted PARAFAC modeling.

*Competing interests.* The authors declare that they have no conflict of interest.



*Acknowledgments.* We are grateful to the captain and crews of the cruises for their corporation and to
Z. Shi, M. Chen, Q. Sun, and L. Han for their help during sampling. Editor's comments improved the
manuscript. This study was supported by grants from National Natural Science Foundation of China
(41606098 and 41376081) and Tianjin Natural Science Foundation (16JCQNJC08000). HX was
holding an adjunct professorship at Tianjin University of Science &Technology during this work.

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



**Figure captions**

**Figure 1.** Map of sampling stations in the Pearl River Estuary. Station names starting with letters M,
W, E designate the main, west, and east transects, respectively. See Table 1 for coordinates of the
stations. HM: Humen; JM: Jiaomen; HQM: Hongqimen; HeM: Hengmen; MDM: Maodaomen; HMH:
Huangmaohai.

**Figure 2.** Excitation-emission contours of five components identified by PARAFAC modeling (left
panels) and split-half validations of excitation and emission loadings (right panels). Excitation/emission
maximum wavelengths are: C1: 275/320 nm; C2: <240(335)/426 nm; C3: 245/378 nm; C4:
255(370)/464 nm; C5: <240(290)/348 nm.

**Figure 3.** DOC concentration and $a_{330}$ versus salinity in the PRE. Red circles denote samples collected
in the low-salinity section where DOC and $a_{330}$ showed rapid decreases or large variabilities with
salinity. Blue circles denote the samples collected in the saltier zone. Solid lines in panels a and c
represent means of the blue circles. Solid lines in the other panels denote linear fits of the blue circles.
Dashed lines signify the 95% confidence intervals. See Table 4 for fitted equations and statistics.

**Figure 4.** Same as in Figure 3b,d,e–g except for FDOM components C1+C5, C2+C4, and C3.

**Fiure 5.** $E_2/E_3$ versus salinity for each cruise. Lines denote conservative mixing lines defined by the
lowest- and highest-salinity points in the saltier zone.

**Figure 6.** Percentages of FDOM components versus salinity for each cruise. Red solid circles denote
samples collected along the west transect (see Figure 1) in August.

**Figure 7.** BIX (a) and HIX (b) versus salinity. Red solid circles denote samples collected along the
west transect (see Figure 1) in August.

**Figure 8.** DOC concentration versus $a_{330}$ (a), C1+C5 (b), C2+C4 (c), and C3 (d). Solid lines denote
linear fits of data for each cruise. See Table 5 for fitted equations and statistics.





**Table 1.** Coordinates of sampling stations, sampling times, and water depths at the sampling stations.

| Transect | Station | Lat. (°N) | Long. (°E) | Water depth (m) | Sampling date and time | | | |
|---|---|---|---|---|---|---|---|---|
| | | | | | May | August | November | January |
| Main transect | M01 | 23.033 | 113.517 | 12.0 | 12 (13:30) | 11 (11:04) | 19 (12:14) | 14 (11:57) |
| | M02 | 22.967 | 113.540 | 5.5 | 12 (12:10) | 11 (10:15) | 19 (11:20) | 14 (11:12) |
| | M03 | 22.790 | 113.623 | 18.6 | 12 (8:20) | 11 (8:13) | 19 (8:46) | 13 (14:27) |
| | M04 | 22.710 | 113.682 | 17.9 | 11 (15:50) | 7 (8:25) | 16 (9:00) | 10 (8:45) |
| | M05 | 22.585 | 113.691 | 4.9 | 11 (12:50) | 7 (10:00) | 16 (10:38) | 10 (10:20) |
| | M06 | 22.523 | 113.751 | 11.4 | 10 (12:40) | 10 (13:28) | 16 (11:28) | 10 (11:20) |
| | M07 | 22.426 | 113.752 | 8.8 | 10 (10:20) | 10 (10:30) | 16 (13:30) | 10 (13:15) |
| | M08 | 22.257 | 113.722 | 6.9 | 9 (9:52) | 9 (10:55) | 18 (10:38) | 12 (10:21) |
| | M09 | 22.122 | 113.715 | 7.7 | 8 (10:10) | 8 (9:15) | 17 (10:15) | 11 (10:23) |
| | M10 | 21.994 | 113.722 | 20.4 | 8 (13:00) | 8 (12:40) | 17 (14:35) | 11 (14:48) |
| West transect | W01 | 22.411 | 113.684 | 4.8 | 9 (14:50) | 7 (12:55) | 16 (14:30) | 13 (10:00) |
| | W02 | 22.318 | 113.632 | 6.0 | 10 (8:40) | 7 (14:15) | 16 (15:52) | 10 (15:13) |
| | W03 | 22.214 | 113.618 | 4.5 | 9 (8:50) | 9 (9:40) | 18 (9:00) | 12 (9:13) |
| | W04 | 22.065 | 113.590 | 6.3 | 8 (15:40) | 8 (14:40) | 17 (16:10) | 11 (9:23) |
| East transect | E01 | 22.600 | 113.777 | 8.2 | 11(14:40) | 10 (14:42) | 18 (15:28) | 13 (13:30) |
| | E02 | 22.358 | 113.864 | 6.7 | 9 (13:05) | 9 (14:35) | 18 (13:10) | 12 (13:26) |
| | E03 | 22.238 | 113.814 | 21.2 | 9 (11:25) | 9 (13:00) | 18 (11:35) | 12 (11:59) |
| | E04 | 22.040 | 113.825 | 16.2 | Not sampled | 8 (10:40) | 17 (11:47) | 11 (11:46) |








**Table 2.** Means (ranges) of DOC, $a_{330}$ and intensities of fluorescent components in surface and bottom waters for each cruise. Here surf denotes surface and btm bottom.

| | DOC ($\mu$mol L$^{-1}$) | $a_{330}$ (m$^{-1}$) | FDOM (R.U.) | | |
| --- | --- | --- | --- | --- | --- |
| | | | C1+C5 | C2+C4 | C3 |
| **May** | | | | | |
| Surface | 160 (110–243) | NA | NA | NA | NA |
| Bottom | 155 (114–234) | NA | NA | NA | NA |
| ((btm−surf)/surf)*100 | −3.2 | NA | NA | NA | NA |
| **August** | | | | | |
| Surface | 117 (96–167) | 1.92 (1.07–4.35) | 0.90 (0.43–2.02) | 0.51 (0.26–0.80) | 0.30 (0.17–0.46) |
| Bottom | 109 (78–166) | 1.60 (0.56–4.27) | 0.71 (0.16–1.97) | 0.41 (0.11–0.87) | 0.24 (0.08–0.51) |
| ((btm−surf)/surf)*100 | −6.5 | −16.4 | −21.0 | −19.2 | −18.6 |
| **November** | | | | | |
| Surface | 83 (77–133) | 1.42 (0.54–3.35) | 1.21 (0.47–2.30) | 0.40 (0.14–0.61) | 0.24 (0.10–0.37) |
| Bottom | 82 (70–100) | 1.29 (0.60–3.40) | 1.10 (0.20–2.28) | 0.36 (0.08–0.67) | 0.21 (0.06–0.38) |
| ((btm−surf)/surf)*100 | −1.0 | −8.7 | −8.7 | −9.8 | −11.9 |
| **January** | | | | | |
| Surface | 118 (71–194) | 1.34 (0.29–3.98) | 1.03 (0.24–3.09) | 0.30 (0.06–0.69) | 0.20 (0.04–0.51) |
| Bottom | 118 (66–207) | 1.29 (0.33–4.11) | 0.98 (0.23–2.95) | 0.29 (0.06–0.72) | 0.19 (0.04–0.53) |
| ((btm−surf)/surf)*100 | −0.1 | −1.2 | −5.5 | −4.1 | −3.6 |





**Table 3.** Means (ranges) of $E_2/E_3$, BIX, HIX, %(C1+C5), %(C2+C4), and %C3 in surface and bottom waters for each cruise. Here surf denotes surface and btm bottom.

| | $E_2/E_3$ | BIX | HIX | %(C1+C5) | %(C2+C4) | %C3 |
|---|---|---|---|---|---|---|
| **August** | | | | | | |
| Surface | 6.84 (4.76–8.29) | 1.04 (0.95–1.14) | 1.56 (1.01–2.36) | 51.6 (40.2–62.2) | 30.5 (23.6–39.1) | 18.0 (14.2–20.7) |
| Bottom | 6.98 (5.18–8.74) | 1.07 (0.94–1.25) | 1.60 (0.97–2.28) | 50.2 (41.2–62.8) | 31.0 (23.0–38.3) | 18.8 (14.2–22.6) |
| ((btm−surf)/surf)*100 | 2.0 | 3.4 | 2.4 | −2.6 | 1.8 | 4.4 |
| **November** | | | | | | |
| Surface | 6.89 (5.48–9.13) | 1.13 (1.02–1.30) | 1.01 (0.53–1.51) | 64.3 (54.2–71.1) | 22.2 (18.5–29.7) | 13.5 (10.7–16.1) |
| Bottom | 7.17 (6.02–8.44) | 1.09 (0.68–1.30) | 1.03 (0.86–1.50) | 64.0 (52.4–70.3) | 22.6 (18.9–31.9) | 13.5 (10.4–16.5) |
| ((btm−surf)/surf)*100 | 4.0 | −3.8 | 2.5 | −0.6 | 1.7 | 0.1 |
| **January** | | | | | | |
| Surface | 7.60 (6.08–11.16) | 1.15 (1.04–1.53) | 0.90 (0.54–1.26) | 66.4 (61.6–72.2) | 20.2 (15.9–23.5) | 13.5 (11.7–14.9) |
| Bottom | 7.61 (6.10–10.38) | 1.16 (1.03–1.36) | 0.88 (0.64–1.19) | 66.3 (61.8–71.7) | 20.3 (16.5–23.5) | 13.4 (11.9–14.8) |
| ((btm−surf)/surf)*100 | 0.2 | 0.5 | −2.7 | −0.1 | 0.5 | −0.4 |





**Table 4.** Results of linear regression (Y = a*X + b) of DOM quantitative variables
against salinity. SE denotes standard error.

|  | a±SE | b±SE | $R^2$ | $p$ |
|---|---|---|---|---|
| | | DOC | | |
| August | −1.31±0.16 | 121.3±2.4 | 0.72 | <0.0001 |
| January | −1.11±0.21 | 123.0±4.5 | 0.57 | <0.0001 |
| | | $a_{330}$ | | |
| August | −0.042±0.006 | 1.97±0.09 | 0.67 | <0.0001 |
| November | −0.029±0.003 | 1.47±0.05 | 0.80 | <0.0001 |
| January | −0.048±0.003 | 1.93±0.07 | 0.88 | <0.0001 |
| | | C1+C5 | | |
| August | −0.023±0.004 | 0.94±0.06 | 0.62 | <0.0001 |
| November | −0.035±0.005 | 1.43±0.10 | 0.67 | <0.0001 |
| January | −0.043±0.004 | 1.61±0.08 | 0.82 | <0.0001 |
| | | C2+C4 | | |
| August | −0.016±0.001 | 0.60±0.02 | 0.83 | <0.0001 |
| November | −0.014±0.001 | 0.52±0.01 | 0.94 | <0.0001 |
| January | −0.014±0.001 | 0.53±0.02 | 0.90 | <0.0001 |
| | | C3 | | |
| August | −0.008±0.001 | 0.33±0.01 | 0.74 | <0.0001 |
| November | −0.008±0.001 | 0.31±0.01 | 0.91 | <0.0001 |
| January | −0.008±0.001 | 0.32±0.01 | 0.86 | <0.0001 |







**Table 5.** Results of linear regression ($Y = a*X + b$) of [DOC] against $a_{330}$ and FDOM
components. SE denotes standard error.

| | a±SE | b±SE | $R^2$ | $p$ |
|---|---|---|---|---|
| | | $a_{330}$ | | |
| August | 22.5±1.4 | 72.5±2.9 | 0.89 | <0.0001 |
| November | 18.8±2.2 | 61.8±3.3 | 0.68 | <0.0001 |
| January | 32.0±2.0 | 71.4±3.3 | 0.90 | <0.0001 |
| | | C1+C5 | | |
| August | 43.5±2.1 | 76.2±2.0 | 0.93 | <0.0001 |
| November | 16.3±1.9 | 66.5±2.5 | 0.68 | <0.0001 |
| January | 40.7±2.3 | 72.6±3.0 | 0.91 | <0.0001 |
| | | C2+C4 | | |
| August | 100.7±11.4 | 64.3±5.9 | 0.72 | <0.0001 |
| November | 52.2±9.8 | 65.2±4.0 | 0.47 | <0.0001 |
| January | 159.2±13.7 | 66.7±4.9 | 0.82 | <0.0001 |
| | | C3 | | |
| August | 197.0±17.6 | 57.4±5.3 | 0.80 | <0.0001 |
| November | 106.8±16.3 | 61.1±3.9 | 0.56 | <0.0001 |
| January | 242.5±16.2 | 65.8±4.0 | 0.89 | <0.0001 |











**Table 6.** Estimates for DOC and CDOM ($a_{330}$-based) export from the Pear River to the South China Sea based on monthly freshwater discharge rates for the sampling year and those averaged over a 10-year period from 2006 to 2016. Standard errors of the fluxes for the sampling year were derived from the standard errors of the effective [DOC] and $a_{330}$ (Table 4), while those for the 10-year period also include the interannual variability of the freshwater discharge rate.

|  | Freshwater discharge ($\times 10^{10}$ m$^3$) | | Fluxes | | | |
|  |  |  | DOC ($\times 10^9$ g) | | CDOM ($\times 10^9$ m$^2$) | |
|  | Sampling year | 10-year average | Sampling year | 10-year average | Sampling year | 10-year average |
| Spring | 3.58 | 3.63±0.78 | 53.5±2.4 | 54.2±11.9 | 71.3±4.9 | 72.2±16.2 |
| Summer | 5.68 | 6.17±1.22 | 82.7±1.0 | 89.9±17.7 | 112±3 | 122±24 |
| Autumn | 5.06 | 2.75±0.74 | 49.6±2.1 | 27.0±7.3 | 74.1±1.4 | 40.3±10.8 |
| Winter | 3.71 | 1.65±0.45 | 54.3±1.2 | 24.3±6.7 | 71.0±1.5 | 31.8±8.7 |
| Annually | 18.0 | 14.2±1.7 | 240±4 | 195±24 | 329±6 | 266±32 |





**Table 7.** DOC concentrations and $a_{330}$ in surface water of the Pearl River estuary
reported in the literature and this study.

| Month | DOC ($\mu mol\ L^{-1}$) | Sampling Year | Reference |
|---|---|---|---|
| Jan. | 71–194 | 2016 | This study |
| Feb. | 100–247[a] | 2004 | Lin (2007) |
| Mar. | 109–266 | 1997 | Dai et al. (2000) |
| | 103–229[a] | 2006 | Lin (2007) |
| Apr. | 84–278[b] | 2007 | He et al. (2010) He (2010) |
| May | 110–243 | 2015 | This study |
| | 58–160[c] | 2001 | Callahan et al. (2004) |
| Jul. | 109–315 | 1996 | Dai et al. (2000) |
| | 68–250 | 1999 | Chen et al. (2004) |
| Aug. | 96–167 | 2015 | This study |
| | 107–164[a] | 2005 | Lin (2007) |
| | 94–124[b] | 2008 | He (2010) |
| Nov. | 77–133 | 2015 | This study |
| | 82–187[c] | 2002 | Callahan et al. (2004) |
| Month | $a_{330}$ ($m^{-1}$) | Sampling Year | Reference |
| Jan. | 0.29–3.98 | 2016 | This study |
| May | 0.37–7.48[d] | 2014 | Lei et al. (2018) |
| Jul. | 1.01–3.38[d] | 2013 | Wang et al. (2014) |
| | 0.54–1.98 | 1999 | Chen et al. (2004) |
| Aug. | 1.07–4.35 | 2015 | This study |
| Nov. | 0.54–3.35 | 2015 | This study |
| | 0.38–2.73 | 2002 | Hong et al. (2005) |

[a]Ranges were estimated using the fitted [DOC]-salinity equations in Lin (2007) over
salinity 0-30.
[b]DOC concentrations upstream of Sta. M01 in the present study are excluded.
[c]Values were retrieved from figures 5a and 8b in Callahan et al. (2004).
[d]Ranges were estimated using exponential decay equations established from data in table
1 in Lei et al. (2018).





**Table 8.** DOC concentrations and CDOM abundances ($a_{330}$) in major world rivers.

| River | DOM | References |
|---|---|---|
| DOC ($\mu$mol L$^{-1}$) | | |
| Amazon | 235 | Raymond and Bauer (2001) |
| | 277 | Cao et al. (2016) |
| | 307 (122–492) | Seidel et al. (2016) |
| Mississippi | 489 (231–672) | Bianchi et al. (2004) |
| | 417[a] | Spencer et al. (2013) |
| Atchafalaya | 331[a] | Spencer et al (2013) |
| St. Lawrence | 307 (25–1333) | Hudon et al. (2017) |
| | 231[a] | Spencer et al. (2013) |
| Mackenzie | 375±100 | Cooper et al. (2005) |
| | 347 (258–475) | Raymond et al. (2007) |
| | 402 (250–576)[b] | Osburn et al. (2009) |
| | 363 (250–475) | Stedmon et al. (2011) |
| Yukon | 533±242 | Cooper et al. (2005) |
| | 509 (217–1258) | Raymond et al. (2007) |
| | 574[a] | Spencer et al. (2013) |
| | 674 (200–1617) | Stedmon et al. (2011) |
| Kolyma | 500±167 | Cooper et al. (2005) |
| | 594 (250–1025) | Stedmon et al. (2011) |
| Lena | 724±283 | Cooper et al. (2005) |
| | 775 (542–1233) | Raymond et al. (2007 |
| | 948 (550–1600) | Stedmon et al. (2011) |
| Ob | 733±167 | Cooper et al. (2005) |
| | 780 (458–1000) | Raymond et al. (2007) |
| | 875 (375–1058) | Stedmon et al. (2011) |
| Yenisey | 733±316 | Cooper et al. (2005) |
| | 638 (242–1050) | Raymond et al. (2007) |
| | 754 (208–1250) | Stedmon et al. (2011) |
| Yellow | 202 (151–280) | Wang et al. (2012) |
| Yangtze | 169 (137–228) | Wang et al. (2012) |
| Pearl River | 149 (72–243)[c] | This study |
| $a_{330}$ (m$^{-1}$) | | |
| Amazon | 13.05[d] | Cao et al. (2016) |
| Mississippi | 9.60[a] | Spencer et al. (2013) |
| Atchafalaya | 11.55[a] | Spencer et al. (2013) |
| St. Lawrence | 9.65[e] | Xie et al. (2012) |
| | 2.16[a] | Spencer et al. (2013) |
| Mackenzie | 8.30 (5.19–13.30)[b] | Osburn et al. (2009) |
| | 6.04 (3.01–9.63) | Stedmon et al. (2011) |
| Yukon | 17.34[a] | Spencer et al. (2013) |
| | 14.50 (2.65–37.84) | Stedmon et al. (2011) |
| Kolyma | 13.63 (5.77–29.19) | Stedmon et al. (2011) |
| Lena | 26.51 (15.48–52.94) | Stedmon et al. (2011) |
| Ob | 22.43 (6.74–30.74) | Stedmon et al. (2011) |
| Yenisey | 22.14 (3.50–44.79) | Stedmon et al. (2011) |
| Yangtze (Changjiang) | 2.60 (2.29–3.02)[f] | Song et al. (2017) |
| Pearl River | 2.50 (1.04–4.35)[c] | This study |

[a]Retrieved from DOC and CDOM fluxes and freshwater discharge rates in Spencer et al. (2013).
[b]From data at salinities <5
[c]From data at salinities <5.
[d]Retrieved from the spectral slope and $a_{350}$ at Sta. 10 in Cao et al. (2016)
[e]Average value at Sta. SL1 and SL2 in Xie et al. (2012).
[f]Average value at salinities <5.





**Table 9.** CDOM fluxes ($a_{330}$-based) from major world rivers to the ocean reported in the
literature. The flux estimated for the Pearl River by this study is also included for
comparison.

| River | Flux ($\times 10^9$ m$^2$ year$^{-1}$) | Reference |
|---|---|---|
| Mississippi | 5070 | Spencer et al. (2013) |
| Atchafalaya | 2750 | Spencer et al. (2013) |
| St. Lawrence | 490 | Spencer et al. (2013) |
| Mackenzie | 1550 | Stedmon et al. (2011) |
| Yukon | 3520 | Spencer et al. (2013) |
|  | 3260 | Stedmon et al. (2011) |
| Kolyma | 1340 | Stedmon et al. (2011) |
| Lena | 17100 | Stedmon et al. (2011) |
| Ob | 7350 | Stedmon et al. (2011) |
| Yenisey | 12600 | Stedmon et al. (2011) |
| Pearl River | 266 | This study |






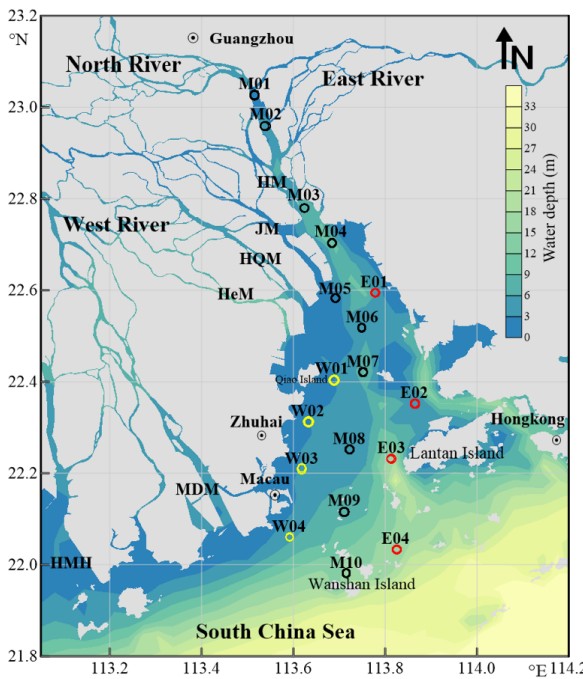


**Figure 1.** Map of sampling stations in the Pearl River Estuary. Station names starting
with letters M, W, E designate the main, west, and east transects, respectively. See Table
1 for coordinates of the stations. HM: Humen; JM: Jiaomen; HQM: Hongqimen; HeM:
Hengmen; MDM: Maodaomen; HMH: Huangmaohai.

981





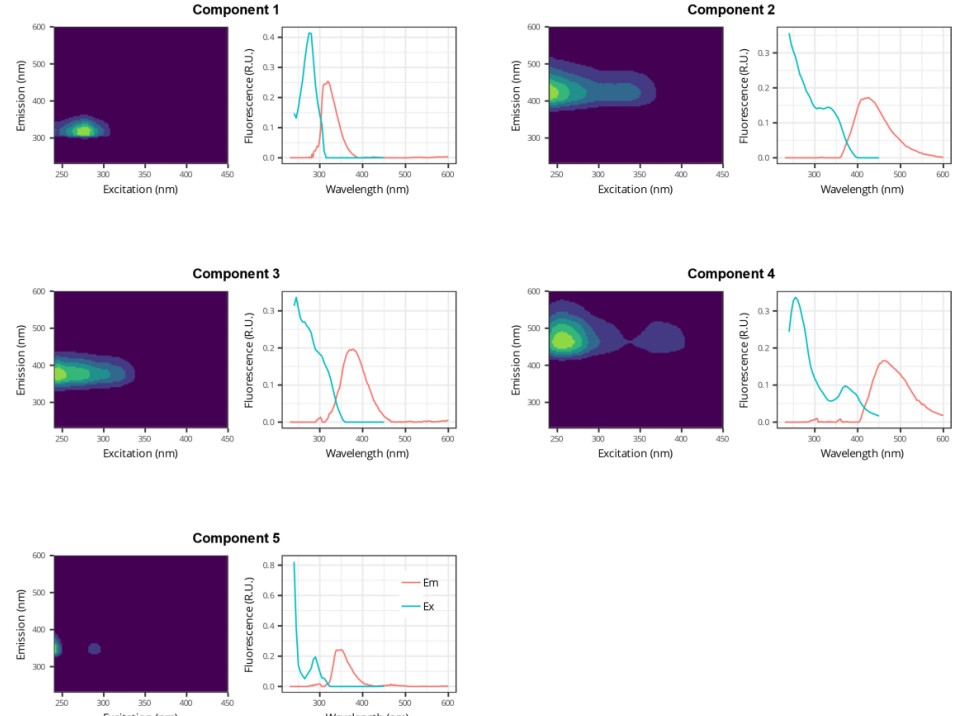

982

**Figure 2.** Excitation-emission contours of five components identified by PARAFAC modeling (left panels) and split-half validations of excitation and emission loadings (right panels). Excitation/emission maximum wavelengths are: C1: 275/320 nm; C2: <240(335)/426 nm; C3: 245/378 nm; C4: 255(370)/464 nm; C5: <240(290)/348 nm.

987



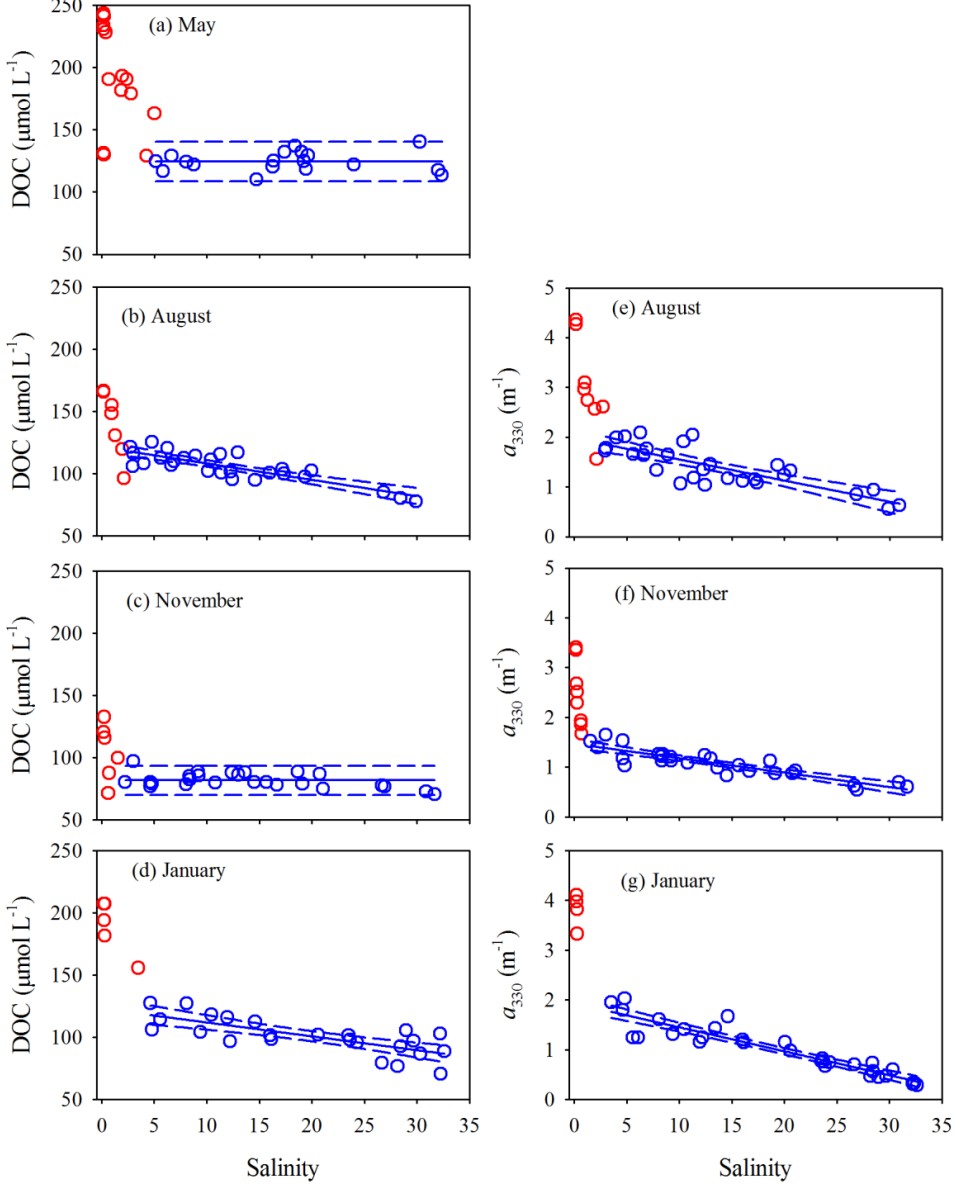

988

**Figure 3.** DOC concentration and $a_{330}$ versus salinity in the PRE. Red circles denote
samples collected in the low-salinity section where DOC and $a_{330}$ showed rapid decreases
or large variabilities with salinity. Blue circles denote the samples collected in the saltier
zone. Solid lines in panels a and c represent means of the blue circles. Solid lines in the
other panels denote linear fits of the blue circles. Dashed lines signify the 95%
confidence intervals. See Table 4 for fitted equations and statistics.





**Figure 4.** Same as in Figure 3b,d,e–g except for FDOM components C1+C5, C2+C4, and C3.






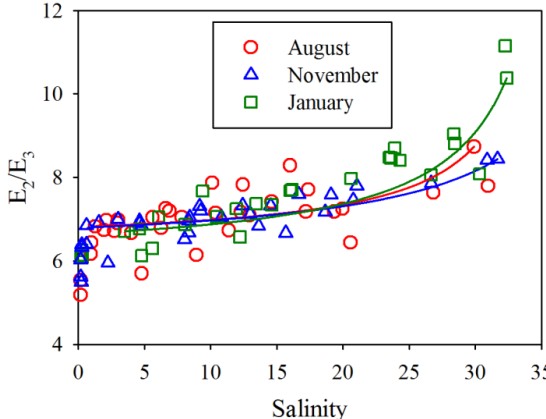


**Figure 5.** $E_2/E_3$ versus salinity for each cruise. Lines denote conservative mixing lines
defined by the lowest- and highest-salinity points in the saltier zone.



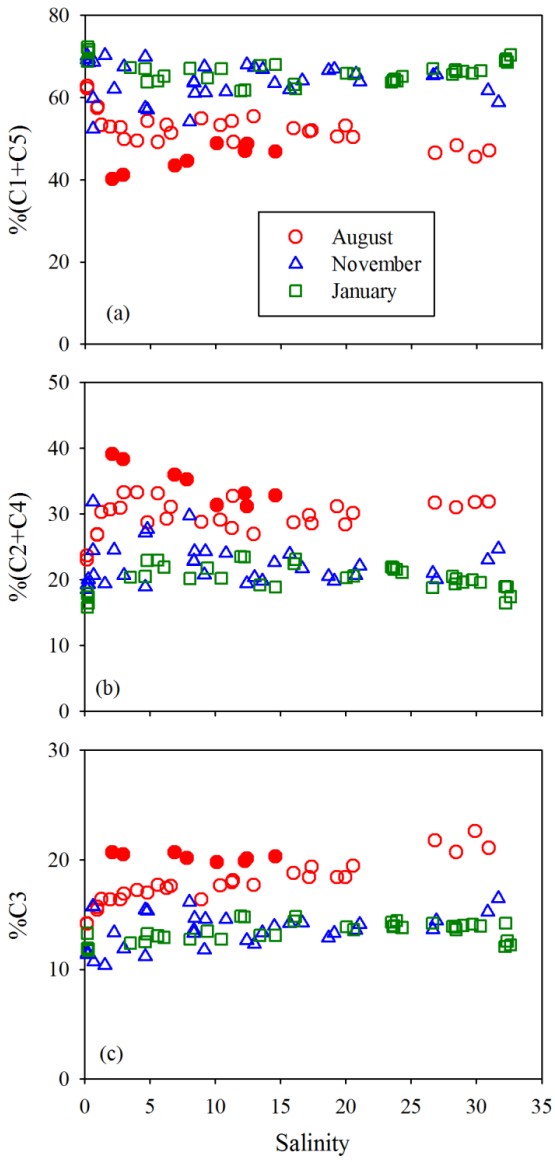


**Figure 6.** Percentages of FDOM components versus salinity for each cruise. Red solid circles
denote samples collected along the west transect (see Figure 1) in August.





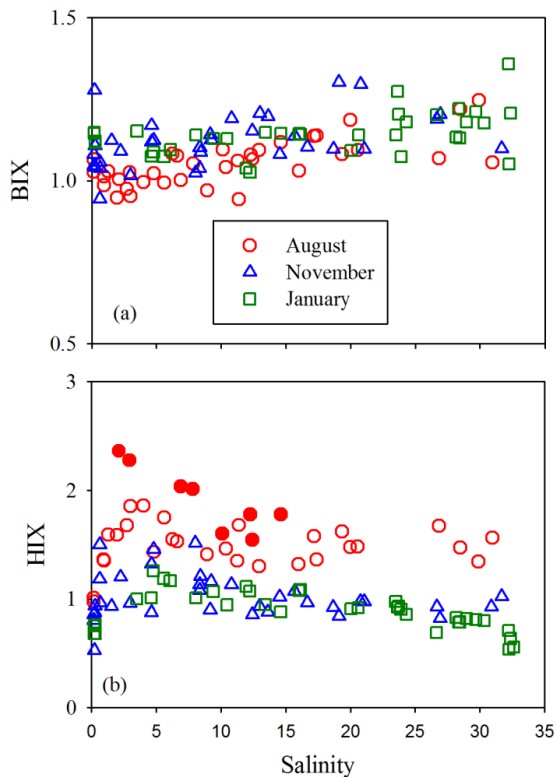


**Figure 7.** BIX (a) and HIX (b) versus salinity. Red solid circles denote samples collected
along the west transect (see Figure 1) in August.



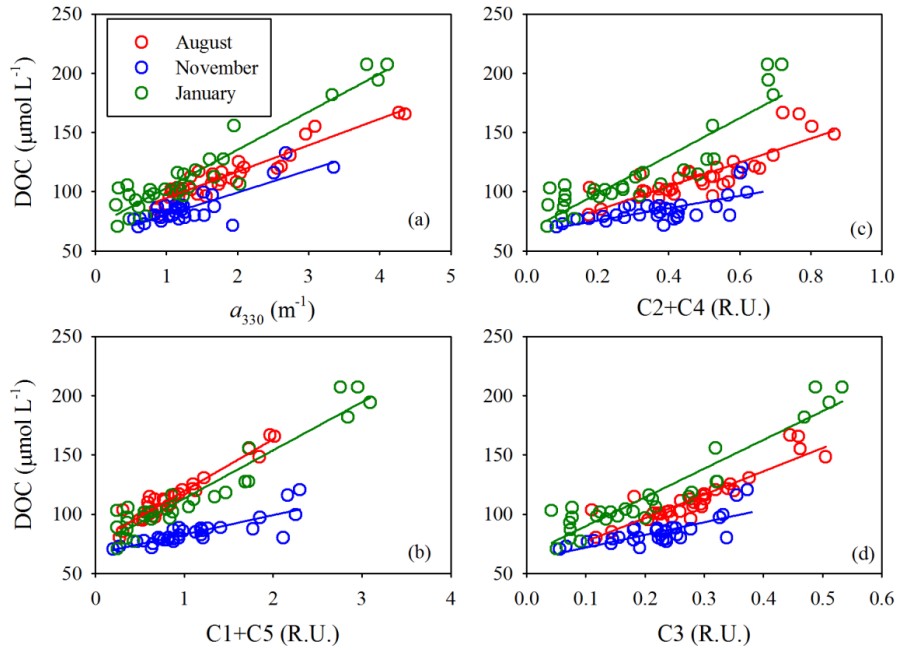


**Figure 8.** DOC concentration versus $a_{330}$ (a), C1+C5 (b), C2+C4 (c), and C3 (d). Solid lines
denote linear fits of data for each cruise. See Table 5 for fitted equations and statistics.