# Peer review of "Distribution, seasonality, and fluxes of dissolved organic matter (DOM) in the"

_Biogeosciences, 2018_

## Referee Comment (RC1) · He (Referee) · 19 Dec 2018

The paper entitled "Distribution, seasonality, optical characteristics, and fluxes of dissolved organic matter (DOM) in the Pearl River (Zhujiang) estuary, China" investigated seasonal and spatial variations of CDOM and FDOM characterized by absorption and fluorescence spectroscopy. Since I am an organic geochemist focusing on the organic carbon and nitrogen cycling mechanism in estuarine coastal zones and the role of microbes during the organic matter cycling, I am very familiar with the topic of this manuscript. This manuscript identified the compositional characteristics and sources of DOM. The main conclusion is that (i) microbial inputs and anthropogenic inputs are

important sources of DOM in the freshwater end; (ii) small seasonal variations with respect to DOC and CDOM; and (iii) PR exports the lowest quantality of DOC among 30 large world rivers, although the size of PR watershed ranked the thirteenth largest in the world by area. Considering the anthropogenic activities can influence the quality and quantity of DOM in aquatic ecosystems and urbanization trends continue in response to human population growth, anthropogenic influences on DOM composition will likely become more widespread. Such human effects on DOM quality could have strong impacts on carbon cycles and need to be better understood. Therefore, this study provides a typical case study to approach the scientific questions mentioned above. However, some points need to be addressed as follows. Nevertheless, this work did provide interesting findings, and the data is reasonably strong to make the conclusions, and there I suggest a moderate revision needs to perform before the acceptance of this manuscript.

General comments: 1. In terms of English, I suggest the writing should be improved further. 2. The description of "overview of DOM" is great. However, I realize that it is too general. I hope the authors could provide introduction related with their discussion or the questions that need to be solved (or knowledge gap). In addition, the transition from 1.1 to 1.2 seems not that smooth to me. 3. The chapter "1.2 The Pearl River estuary (PRE)" is too lengthy to describe the important focus and question, and some of descriptions can be moved to "Site description", otherwise part of the information seems duplicated. For instance, the authors spent 9 paragraphs to describe the PRE, and some of the information is not closely related with the results/discussions. This needs to be shortened and be questions oriented. 4. The authors mentioned precipitation is an important factor affecting soil flushing, which may affect both DOM quality and quantity. It would be great if the author could incorporate some monthly or seasonal precipitation data to support their claims. In particular, the article indicated the terrigenous DOM is the main source of investigated areas, but it did not describe the influences of land runoff and rainfall on seasonal variations of DOM. 5. In this manuscript the author suggested that the low DOC concentrations in PRE (especially the low salinity region) was affected by biological degradation (due to input of labile DOM) and low inputs due to the low forest cover. This is a good point! I suggest the author expand this description a little bit. For instance, (i) the addition of labile DOM may "prime" the degradation of terrestrial (relatively more recalcitrant) DOM; (ii) the author could specify the land use percentages of the PR watershed and compare it with the other large river-estuarine systems (such as the Amazon River). Some of the land use% data has been organized in Wagner et al. (2015), and I believe the land use% data is not that difficult to find for PR watershed; (iii) since the authors claim that the PRE is an super eutrophic system, it would be interesting at least present some nutrient data (from literatures) to further support their main findings. 6. I really like the main findings in the manuscript, but these findings are not well reflected in the abstract. I suggest the author re-organize their abstracts and focusing on the main findings. Reporting numbers are great, but there seem to be too many. Keep the important ones would be good enough. Wagner, S., Riedel, T., Niggemann, J., VaÌLhaÌLtalo, A. V., Dittmar, T., & JaffeÌA̧, R. (2015). Linking the molecular signature of heteroatomic dissolved organic matter to watershed characteristics in world rivers. Environmental science & technology, 49(23), 13798-13806. 7. Considering the author spent a huge effort collecting all these samples, it would be very interesting to perform some statistical analysis such as the principal component analysis (PCA) to further confirm the major controls to the DOM variability across the whole dataset.

Specific comments: 1. There was no explanation about the inverse changes of BIX and HIX in Fig.7 2. I suggest the author make it clear what is "the saltier zone" because this is a ambiguous description. 3. Considering there are way too many tables. I suggest move some of the tables (e.g., Table 1) to the supplementary information. The DOC ($\mu$mol L-1) needs to be moved to the second column. 4. Would be wonderful if the author could point out the major metropolitan areas (or even land use patterns) in Figure 1 since it closely related with the major discussions in this manuscript. 5. When the authors describe each PARAFAC component, I suggest the author use DOM Openfluor database to compare the components in this study with literature data. Murphy, K.

R., Stedmon, C. A., Wenig, P., & Bro, R. (2014). OpenFluor–an online spectral library of auto-fluorescence by organic compounds in the environment. Analytical Methods, 6(3), 658-661. 7. R.U. should be defined in the abstract.

---

## Short Comment (SC1) · 20 Dec 2018

Dissolved organic matter is an important component of the carbon cycle in aquatic systems and it exerts direct impact on the overall biogeochemical process in the ocean. DOM spectroscopy has emerged as a cost-effective and easy-to-measure technique for quantifying and, more recently, qualify the DOM content in the environment. The manuscript by Li and colleagues brings results on DOM amount (expressed by means of DOC and spectroscopic measurements), characterization (through EEM-PARAFAC), fluxes and seasonal variability for the Pearl River Estuary, China. The data set is robust and the methods applied align with current literature. Although the

sampling grid remains the same for the different seasons, the seasonal averages presented in the MS might be biased by the spatial variability presented within the water masses spatial distribution within the region. Therefore, I suggest the authors to have lead the MS through a more "oceanographic point of view", i.e., by investigating the seasonal changes within the water masses presented within the region. Although the manuscript is well written and reads easily, the way that sections are structure makes the manuscript repetitive when presenting and discussing results. I think it would become more concise and interesting if the authors focus on making a rearrangement of sections (by merging/condensing some of them) and on making a review through the text to avoid such repetitions. Additionally, the introduction is a bit too long and could be shortened by providing only information needed for interpretation of results from this study. Thus, to my judgment, the manuscript may be publishable after major reviews.

GENERAL COMMENTS:

- The abstract does not clearly illustrates the main findings obtained in the study.

- The hypothesis presented in section 1.3 seem weak and vague, and could be sharper. Seasonal variability in DOM flux is already expected from an estuary with marked seasonal variability in freshwater export, as documented by the authors.

- Sampling strategy: why was decided to collect the "deep water" sample near the bottom and not below the pycnocline? It can be affected by sediment resuspension, if there is any.

- Have the authors looked at the CDOM absorption spectral slope and slope ratio? It could provide more insights into the photochemical reactions along the estuarine mixing.

- The authors could also try to use multivariate analysis (e.g., PCA) to analyze the variability between the campaigns (i.e., over time) and to elucidate what are the main drivers on DOM variability within the region.

- I suggest the authors to compare their PARAFAC-derived components spectra with the OpenFluor database (https://openfluor.lablicate.com/). This would benefit the comparison established with other studies along the MS.

- With respect to the sources of DOM to region, especially the pollution-derived DOM, they could be more stressed along the MS. It is not totally clear how the findings of this study support that.

- Section 4.5 establishes comparisons among global DOM studies but I expected the discussion to bring some conclusions on the reason for such differences rather than just comparing them.

SPECIFIC COMMENTS:

L75-79: authors could give more background on anthropogenic/pollution-derived DOM, given that it is a DOM source for the region, as pointed out in this study.

L115-119: Please present values (ranges) for the variables. How much does the phytoplankton biomass vary within the seasons?

L124-125: Are there only those two studies supporting this affirmation? No study published in English?

L306-307: what do the authors mean by "freshwater input from this river appeared to have little influence on [DOC]" ?

L500-503: Missing references.

L522-526: I found the explanation for different mixing behavior weak and should be discussed more in deep.

L527-535: this paragraph/discussion could be deepened in the sense to explain the reasons for such variations.

L538-547: Why does it only have good correlations for summer and winter? What

happens with the correlations during the other seasons? Additionally, was the DOC-aCDOM correlation significant and strong? I ask that, because that correlation does not hold true for several environments.

L556-580: authors could deepen the discussion regarding the fluxes.

L615-623: what could the authors point out as the reason for such differences?

Figure 1: It would be interesting to have two panel composing this figure: one with the sampling sites and another with the city names and also the main circulation patterns.

Figs 3, 4, 5 and 8: please present the curve fits and stats.

―――――――――――――――――――

---

## Referee Comment (RC2) · Anonymous Referee #2 · 24 Jan 2019

This paper deals with the seasonal variability, spatial distribution, transformation processes and fluxes of dissolved organic matter (DOM) in the Pearl River estuary (PRE) in China. DOM is investigated through dissolved organic carbon (DOC), chromophoric (CDOM) and fluorescent (FDOM) dissolved organic matter. Overall, this work provides relevant results and good quality data concerning the dynamics and fluxes of DOM in the PER. The manuscript is well structured, quite well written, and is obviously within the scope of Biogeosciences.

Therefore, I recommend the paper to be published in Biogeosciences after "moderate" revisions. Below my comments:

[Figure]

- Title. The part "optical characteristics" could be removed from the title.

- Although English is not bad, the manuscript could benefit from corrections of an English native speaker.

- The abstract has to be substantially improved. It does not reflect at all the relevance of the study. For instance, the following part: "The seasonality of average DOM abundance varied as follows: DOC: May (156 $\mu$mol L$-1$) > January (114 $\mu$mol L$-1$) $\sim$ August (112 $\mu$mol L$-1$) > November (86 $\mu$mol L$-1$); CDOM absorption at 330 nm: August (1.76 m$-1$) > November (1.39 m$-1$) $\sim$ January (1.30 m$-1$); FDOM expressed as the sum of the maximum fluorescence intensities of all FDOM components: November (1.77 R.U.) > August (1.54 R.U.) $\sim$ January (1.49 27 R.U.). Average DOM abundance in surface water was higher than in bottom water, their difference being marginal (0.1–10%) for DOC in all seasons and for CDOM and FDOM in November and January, and moderate (16–21%) for CDOM and FDOM in August" did not deserve to be included in the abstract.

- Introduction. Subtitles ("1.1 Overview of DOM", "1.2 The Pear River estuary", "1.3 Hypothesis and objectives") should be removed. Usually there is no subtitle in the introduction. The first part concerning DOM is OK but the second one (PRE) is too long and too detailed. Most of these details should go in the "2 Methods" part, in a "2.1 Study area" section, which currently does not exist by the way. Only information about PRE that is useful for highlighting the problematic and hypothesis is necessary in the Introduction.

- Introduction. The sentence: "The biogeochemical and optical significance of DOM depends on both its abundance and quality (i.e.chemical composition), with the latter strongly linked to its origin of formation" is not clear. Please re-phrase.

- Sample collection. I guess the number of samples collected at each season for DOM analyses is not mentioned. This should be mentioned here.
- The subtitle "2.2 Sample analysis" should be replaced by "2.2. DOM "analysis"

- DOM analyses. "The analytical uncertainty of aCDOM measurement was assessed by analyzing six pairs of duplicate samples collected from the August cruise. Average aCDOM at 330 nm (a330) was 2.19 m−1 (range: 1.19–4.37 m−1); the average difference in each pair was 0.07 ± 0.05 m−1, or 3.0% ± 1.4%." This method for assessing the analytical uncertainty (precision?) is not clear to me. Why using six pairs of duplicates? I would have used six replicates (of the same sample). The values "0.07 ± 0.05 m−1, or 3.0% ± 1.4%" is not pertinent.

- DOM analyses. CDOM spectral slope in the range 300-500 nm (S300-500 in nm-1) is reported in the supplementary material (Table S1) but is not really discussed in the manuscript. Also, in addition to S300-500 I would recommend the determination and examination of S275-295, proposed by Helms et al. (2008) and largely used yet. It could bring significant information about CDOM molecular weight and transformation processes.

- DOM analyses. HIX, BIX and E2/E3 should be defined in this section and not in the results section.

- Results. The number of Tables is quite high. I recommend adding some in the supplementary material: Tables 1, 2, 4, 5.

- Results. Besides salinity, are ancillary parameters available for this sampling (i.e., dissolved oxygen, nutrients, chlorophyll,...) that could help the help the interpretation of the DOM dynamics?

- Results. I find there is a lack of use of statistical analyses. For example, ANOVA, t test, Mann Whithney test,... (depending on the normal distribution or not of samples) could be applied to determine statistical differences in the DOM concentrations between seasons, surface/bottom,.... - Moreover, instead of separate a priori the samples by seasons and looking at differences between these seasons (that do not necessarily

represent/reflect different hydrological or meteorological events which have occurred during the sampling period), it could be also interesting to apply multi-way statistical methods (principal component analysis, hierarchical ascendant classification,...) on all samples regardless of their sampling period. This could lead to different clustering of samples and underline particular processes affecting DOM dynamics, such as the impact of the mixing between marine and river waters, the impact of precipitation/runoff/river flow rate (ex: discrimination between samples collected in dry period and samples collected wet period), which could be obviously independent from seasons.

- Discussion. Lines 600-614: "[DOC] and [CDOM] in the PRE are the lowest among the major world rivers..." This is indeed intriguing. Why DOC and CDOM contents are so low in the PRE. In this part, the authors should also include the assumption of a DOM loss by bacterial degradation and photochemistry.

- Discussion. Line 604: "The lack of correspondence between [DOC]* and a330* and the freshwater discharge rate (Fig. S9) suggests that [DOM] in the PRE be controlled by both soil leaching and pollution input". Here could be also added the hypothesis of in situ autochthonous DOM production from phytoplankton activities, which are generally not negligible in rivers.

---

## Referee Comment (RC3) · Anonymous Referee #3 · 31 Jan 2019

Distribution, seasonality, optical characteristics, and fluxes of dissolved organic matter (DOM) in the Pearl River (Zhujiang) estuary, China Yang Li et al.

This work presents the seasonal distribution (May, Aug, Nov, and Jan 2015) of DOM (DOC concentrations, CDOM absorption and CDOM fluorescent components (from PARAFAC analysis) in Pearl River estuary (PRE), China. DOC concentrations and CDOM absorption and fluorescence properties (and their qualitative metrics) were examined in relation to salinity as well as to each other. In addition, fluxes of DOC and CDOM from the PRE to South China Sea were also estimated. Overall, results of this study provides new insights into the seasonal DOC and optical properties of CDOM

in PRE. In comparison, most previous studies have mainly reported one or two field campaigns, while this study comprised a more seasonal study (four field campaigns). However, the analysis of the data throughout involves simple correlation analysis and is descriptive with no rigorous analysis of field data (spatial analysis, precipitation, chlorophyll and turbidity measurements that were indicated in the text to have been measured). The additional analysis would support a better understanding of the sources and sinks related to the DOM in PRE. I find that the manuscript needs further improvements and the authors should address some major concerns/suggestions before the paper can be accepted for publication.

Major comments/suggestions: 1) There are various major sources of freshwater to the PRE. Previous studies have also indicated spatial differences in the surface and bottom properties in CDOM optical properties (absorption coefficients and spectral slope; e.g., Lei et al. 2018). Furthermore, seasonal analysis of DOC (Ye et al. 2018) indicated strong seasonality in DOC with substantial removal of DOC in the salinity range 5-22. I think a more comprehensive analysis using all the available data (e.g., chlorophyll, turbidity, etc) including spatial distribution plots (surface and bottom) would greatly help in supporting the conclusions of this study.

2) Throughout this study the authors describe the data collected in the main estuary as the saltier zone as opposed to fresh water zone. I think a more traditional separation of the zones (e.g., Cai et al. 2004; upstream region, estuary, outer estuary) would be more appropriate and could better support the results of this study.

3) The absorption coefficient at 330 nm used in this study has not generally been used and therefore not easily comparable to other studies. Although Table S1 includes some of these wavelengths, it would help if the authors replace the absorption at 330 nm with another commonly used wavelength. Also the spectral slope between 275-295 nm is now generally used to assess CDOM properties and should be included in the analysis.

4) CDOM generally is a good optical proxy for DOC, especially in estuaries. Also,

CDOM undergoes rapid photobleaching in the estuaries or the coastal waters. It may not be useful include estimates of CDOM fluxes at 330 mn from the estuary to the SCS, especially since the wavelength used is so unique to this study.

5) It may be useful to look at meteorological data (e.g., wind field) to see if mixing played a role in reducing the variability in DOM surface and bottom properties.

Minor comments: -No indication of how salinity was measured -Methods section could describe the study site rather than in the Introduction.

References: X. Lei, J. pan, A. T. Devlin. 2018. Mixing behavior of chromophoric dissolved organic matter in the Pearl River Estuary in sprig. Continental Shelf Research, 154, 46-54.

F. Ye, W. Guo, G. Wei, and G. Jia. 2018. The sources and transformations of dissolved organic matter in the Pearl River Estuary, China, as revealed by stable isotopes. J. Geophys. Res.: Oceans, 123, 6893-6908.

---

## Author Comment (AC1) · 17 Feb 2019

**Response to Reviewer#1**

**We appreciate the reviewer's constructive comments. Our responses are *italicized*.**

**AR stands for authors' response**

The paper entitled "Distribution, seasonality, optical characteristics, and fluxes of dis- solved organic matter (DOM) in the Pearl River (Zhujiang) estuary, China" investigated seasonal and spatial variations of CDOM and FDOM characterized by absorption and fluorescence spectroscopy. Since I am an organic geochemist focusing on the organic carbon and nitrogen cycling mechanism in estuarine coastal zones and the role of microbes during the organic matter cycling, I am very familiar with the topic of this manuscript. This manuscript identified the compositional characteristics and sources of DOM. The main conclusion is that (i) microbial inputs and anthropogenic inputs are important sources of DOM in the freshwater end; (ii) small seasonal variations with respect to DOC and CDOM; and (iii) PR exports the lowest quantity of DOC among 30 large world rivers, although the size of PR watershed ranked the thirteenth largest in the world by area. Considering the anthropogenic activities can influence the quality and quantity of DOM in aquatic ecosystems and urbanization trends continue in response to human population growth, anthropogenic influences on DOM composition will likely become more widespread. Such human effects on DOM quality could have strong impacts on carbon cycles and need to be better understood. Therefore, this study provides a typical case study to approach the scientific questions mentioned above. However, some points need to be addressed as follows. Nevertheless, this work did provide interesting findings, and the data is reasonably strong to make the conclusions, and there I suggest a moderate revision needs to perform before the acceptance of this manuscript.

General comments:

1. In terms of English, I suggest the writing should be improved further.

*AR: We did further language polishing. In addition, please see the positive comment in SC1 on the readability of the manuscript ("…the manuscript is well written and reads easily…").*

2. The description of "overview of DOM" is great. However, I realize that it is too general. I

hope the authors could provide introduction related with their discussion or the questions that need to be solved (or knowledge gap). In addition, the transition from 1.1 to 1.2 seems not that smooth to me.

3. The chapter "1.2 The Pearl River estuary (PRE)" is too lengthy to describe the important focus and question, and some of descriptions can be moved to "Site description", otherwise part of the information seems duplicated. For instance, the authors spent 9 paragraphs to describe the PRE, and some of the information is not closely related with the results/discussions. This needs to be shortened and be questions oriented.

*AR: As comments 2&3 both concern the Introduction, we respond to them together.*

*The Introduction was condensed and the details of the PRE were moved to a new section "2.1 Study area" in the Methods (as suggested by RC2). The new Introduction is not divided into subsections.*

*It should be noted that, for a paper of multidisciplinary nature like this, it is essential to place the targeted research question into a broad and multifaceted context. **This is particularly helpful for the readers who are not familiar with the PRE**. The Introduction follows a typical logic line proceeding from the general to the specific to the knowledge gap and eventually to the objectives. Section 1.1 unfolds with describing the importance of DOM in the marine ecosystem, followed by the key processes affecting the quantity and quality of DOM in coastal and estuarine environments. The latter serves not only as the basis for discussing the results and interpreting the data later on but also as an overture for presenting the relevant information on the PRE. Section 1.2 summarizes the geography, topography, hydrography, and biology of the PRE (now moved to a new section "2.1 Study area" in the Methods) preceding a brief review of previous DOM studies in this environment and a statement of the knowledge gap. We carefully checked the entire manuscript and can confirm that **every element presented in the Introduction is indispensable to and echoed in the later sections**. For example, the geographical information is required for describing the sampling scheme, while the topographic (water depths, shoals, channels), hydrographic (freshwater discharge rates, freshwater flow paths, water column stratification, turbidity), biological (phytoplankton biomass), and pollution-related information all provides fundamental context for data presentation and interpretation. Besides, the separate*

*treatment of quantitative and qualitative DOM variables in the Introduction foreshadows a similar structural arrangement in data presentation in the Results section. **The manuscript is such structured that removing any element in the Introduction (and section "2.1 Study area" in the revised version) would compromise the integrity of the entire article.***

*The manuscript currently does not have a "Site description" section and we could not find information in the Introduction is repeated somewhere else (as the reviewer mentioned in the comment 3).*

4. The authors mentioned precipitation is an important factor affecting soil flushing, which may affect both DOM quality and quantity. It would be great if the author could incorporate some monthly or seasonal precipitation data to support their claims. In particular, the article indicated the terrigenous DOM is the main source of investigated areas, but it did not describe the influences of land runoff and rainfall on seasonal variations of DOM.

***AR:** The freshwater discharge to the PRE, which has already been described in the paper, is directly correlated to precipitation over its watershed and is a more direct indicator of the impact of precipitation (than precipitation itself) on the study area.*

*Note that the article does not conclude that terrigenous DOM is the main source of DOM in the PRE. Instead, it underscores the microbial nature of this DOM pool and a potentially important contribution from river-borne DOM (line 462-471 in the original version).*

5. In this manuscript the author suggested that the low DOC concentrations in PRE (especially the low salinity region) was affected by biological degradation (due to input of labile DOM) and low inputs due to the low forest cover. This is a good point! I suggest the author expand this description a little bit. For instance, (i) the addition of labile DOM may "prime" the degradation of terrestrial (relatively more recalcitrant) DOM; (ii) the author could specify the land use percentages of the PR watershed and compare it with the other large river-estuarine systems (such as the Amazon River). Some of the land use% data has been organized in Wagner et al. (2015), and I believe the land use% data is not that difficult to find for PR watershed; (iii) since the authors claim that the PRE is a super eutrophic system, it would be interesting at least present some nutrient data (from literatures) to further support their main findings.

*AR: (i) The "priming" concept is a good suggestion. Nonetheless, our results indicate that this effect, if any, was minor, at least in May, August, and January. In the low-salinity section, the [DOC] after the rapid removal of the labile constituents (Fig. 3), except November, was in the same range as that of the background [DOC] reported for the Pearl River upstream of the Pearl River Delta (114-137 uM, line 122 and line 465-466 in the original version), demonstrating little "priming". Downstream of the upper reach, [DOC] either decreased (August and January) or remained roughly constant (May and November) with increasing salinity, again disproving a major DOC loss process caused by priming. We believe that the land-derived DOC in the Pearl River is either priming-resistant or the short residence times of freshwater in the PRE (a few days, line 496-498 in the original version) prevented a significant priming effect from occurring.*

*In the revised manuscript, we have briefly discussed the potential role of the priming effect, particularly for November when the [DOC] at the downstream side of the low-salinity section was substantially lower than the land-derived background [DOC].*

*(ii) Sorry, we exhausted our resources but could not find the land use% data for the Pearl River region. The landscape information reported by Luo et al. (2002), which we cited, though in a more general nature, provides a similar support for the relevant discussion.*

*(iii) We thoroughly checked the manuscript and found that **nowhere** does the article claim the PRE to be a super eutrophic system. The word "eutrophic" does not exist in this article. However, we have added the dissolved inorganic nitrogen values in the revised manuscript.*

6. I really like the main findings in the manuscript, but these findings are not well reflected in the abstract. I suggest the author re-organize their abstracts and focusing on the main findings. Reporting numbers are great, but there seem to be too many. Keep the important ones would be good enough.

*AR: We reorganized the abstract by emphasizing the major findings and reducing numbers.*

7. Considering the author spent a huge effort collecting all these samples, it would be very interesting to perform some statistical analysis such as the principal component analysis (PCA) to further confirm the major controls to the DOM variability across the whole dataset.

*AR: Our results have clearly demonstrated that physical mixing (i.e. salinity) is the predominant factor controlling the variability of DOM in the PRE (Figs. 3 and 4). Here we performed a principal component analysis (PCA) on the all-season dataset that includes variables in addition to salinity, such as water temperature, chl-a, nutrients, suspended particulate matter, and freshwater discharge rate. The DOM dynamics is represented by CDOM absorption at 330 nm ($a_{330}$) and DOC concentration. The first two axes of the PCA explained >74% of the variability in the dataset. Using the first axis on the following graph, one can see that DOC and $a_{330}$, along with a bunch of other variables (e.g. nitrate, nitrite, silicate, chl-a), are strongly negatively correlated to salinity, which is a typical indication of a conservative mixing behavior. In contrast, DOC and $a_{330}$ are only weakly (negatively) linked to the freshwater discharge rate, again consistent with our result (line 604-606 & Fig. S9 in the original version).*

*As the PCA does not bring much new information on the DOM dynamics, we have added the plot to the Supplemental Material (instead of the main text) and briefly discussed it (i.e. reinforcing the conclusion already reached) in the revised manuscript.*

[Figure]

*Figure: PCA analysis based on the all-season dataset. SPM: suspended particulate matter; $PO_4^{3-}$: phosphate; $NO_2^-$: nitrite; DOC: dissolved organic carbon; $a_{CDOM}$(330): CDOM absorption coefficient at 330 nm; $NO_3^-$: nitrate; Chla: chlorophyll a; $SiO_4^{4-}$: silicate; discharge: freshwater discharge rate.*

Specific comments:

1. There was no explanation about the inverse changes of BIX and HIX in Fig.7

*AR: This is self-evident according to the definitions of BIX and HIX (section 3.3): BIX denotes the relative contribution of fresh, microbial-derived FDOM, while HIX signifies the degree of humification, with old, humified FDOM having higher HIX values.*

*Now a statement as follows has been added in the second last paragraph of section 3.5:*

*"BIX and HIX displayed roughly inverse distributional patterns against salinity, as can be inferred from their definitions (Sect. 3.3)."*

2. I suggest the author make it clear what is "the saltier zone" because this is a ambiguous description.

*AR: The saltier zone is indirectly defined between line 358 and 361 in the original version. It refers to the zone with salinity generally >5, where the reported DOM variables showed much slower changes with increasing salinity as compared to the rapid changes near the head of the estuary (i.e. the low-salinity zone). However, the salinity separating these two areas was at times slightly season- and/or variable-specific. We have now explicitly defined the low-salinity and saltier zones in the first paragraph of section 3.4 as follows:*

*"Hereafter, the head region of the estuary showing fast change or high variability of DOM is termed "the low-salinity zone", while the downstream area exhibiting much gentler variations in DOM is referred to as "the saltier zone." The salinity demarcating the low-salinity and saltier zones was generally ~5 but could change slightly with season and the DOM variable of interest (Figs. 3 and 4)."*

3. Considering there are way too many tables. I suggest move some of the tables (e.g., Table 1) to the supplementary information. The DOC (μmol L-1) needs to be moved to the second column.

*AR: Tables 1, 4, and 5 were moved to Supplemental Material. DOC was moved to the second column in Table 8.*

4. Would be wonderful if the author could point out the major metropolitan areas (or even land use patterns) in Figure 1 since it closely related with the major discussions in this manuscript.

*AR: As stated in our response to comment#5, we could not find the land use data for this region. The major cities are already labeled. The discussion does not require information on the metropolitan borderlines. In fact, adding the metropolitan areas reduces the legibility of the map.*

5. When the authors describe each PARAFAC component, I suggest the author use DOM Open-fluor database to compare the components in this study with literature data. Murphy, K. R., Stedmon, C. A., Wenig, P., & Bro, R. (2014). OpenFluor–an online spectral library of auto-fluorescence by organic compounds in the environment. Analytical Methods, 6(3), 658-661.

*AR: This has now been done and a table showing the results of comparison is provided in the Supplemental Material.*

6. R.U. should be defined in the abstract.

*AR: Thanks. Done.*

---

## Author Comment (AC3) · 17 Feb 2019

**Response to Reviewer#2**

**We appreciate the reviewer's constructive comments. Our responses are *italicized*.**

**AR stands for authors' response**

This paper deals with the seasonal variability, spatial distribution, transformation processes and fluxes of dissolved organic matter (DOM) in the Pearl River estuary (PRE) in China. DOM is investigated through dissolved organic carbon (DOC), chromophoric (CDOM) and fluorescent (FDOM) dissolved organic matter. Overall, this work provides relevant results and good quality data concerning the dynamics and fluxes of DOM in the PRE. The manuscript is well structured, quite well written, and is obviously within the scope of Biogeosciences. Therefore, I recommend the paper to be published in Biogeosciences after "moderate" revisions. Below my comments:

1. Title. The part "optical characteristics" could be removed from the title.

*AR: "optical characteristics" was removed.*

2. Although English is not bad, the manuscript could benefit from corrections of an English native speaker.

*AR: The language has been further polished.*

3. The abstract has to be substantially improved. It does not reflect at all the relevance of the study. For instance, the following part: "The seasonality of average DOM abun- dance varied as follows: DOC: May (156 µmol L−1) > January (114 µmol L−1) ~ August (112 µmol L−1) > November (86 µmol L−1); CDOM absorption at 330 nm: Au- gust (1.76 m−1) > November (1.39 m−1) ~ January (1.30 m−1); FDOM expressed as the sum of the maximum fluorescence intensities of all FDOM components: November (1.77 R.U.) > August (1.54 R.U.) ~ January (1.49 27 R.U.). Average DOM abundance in surface water was higher than in bottom water, their difference being marginal (0.1– 10%) for DOC in all seasons and for CDOM and FDOM in November and January, and moderate (16–21%) for CDOM and FDOM in August" did not deserve to be included in the abstract.

*AR: We reorganized the abstract by emphasizing the major findings and reducing numbers.*

4. Introduction. Subtitles ("1.1 Overview of DOM", "1.2 The Pear River estuary", "1.3 Hypothesis and objectives") should be removed. Usually there is no subtitle in the introduction. The first part concerning DOM is OK but the second one (PRE) is too long and too detailed. Most of these details should go in the "2 Methods" part, in a "2.1 Study area" section, which currently does not exist by the way. Only information about PRE that is useful for highlighting the problematic and hypothesis is necessary in the Introduction.

*AR: The description of the PRE in the Introduction section was condensed and the details were moved to a new section "2.1 Study area" in the Methods. Please see our detailed response to Reviewer#1's General Comments 2&3.*

5. Introduction. The sentence: "The biogeochemical and optical significance of DOM depends on both its abundance and quality (i.e. chemical composition), with the latter strongly linked to its origin of formation" is not clear. Please re-phrase.

*AR: Now rephrased to "The significance of DOM-driven biogeochemical and optical processes depends on DOM's abundance and quality (i.e. chemical composition), with the latter strongly linked to the source of DOM"*

6. Sample collection. I guess the number of samples collected at each season for DOM analyses is not mentioned. This should be mentioned here.

*AR: Stating the number of samples does not provide extra essential information, since the numbers of sampling stations and depths are already reported.*

7. The subtitle "2.2 Sample analysis" should be replaced by "2.2. DOM "analysis"

*AR: Changed to "DOM analysis".*

8. DOM analyses. "The analytical uncertainty of aCDOM measurement was assessed by analyzing six pairs of duplicate samples collected from the August cruise. Average aCDOM at 330 nm (a330) was 2.19 m−1 (range: 1.19–4.37 m−1); the average difference in each pair was $0.07 \pm 0.05$ m−1, or $3.0\% \pm 1.4\%$." This method for assessing the analytical uncertainty (precision?) is not clear to me. Why using six pairs of duplicates? I would have used six replicates (of the same sample). The values "$0.07 \pm 0.05$ m−1, or $3.0\% \pm 1.4\%$" is not pertinent.

*AR:* *Analyzing replicates of the same sample excludes the uncertainty associated with the fact that the uncertainty, particularly the relative uncertainty, changes with $a_{CDOM}$. As shown in the article (Fig. 3), $a_{CDOM}$ in the PRE decreased substantially from the head to the mouth of the estuary. An uncertainty determined from "the same sample" thus cannot represent the entire PRE. In this respect, the uncertainty obtained from our approach is more realistic, since the samples for this assessment essentially covered the entire estuary. In fact, we also determined the uncertainty using 6 replicates of one sample ($a_{330}$: 4.37 $m^{-1}$), arriving at a standard deviation of 0.06 $m^{-1}$ or 1.3%, which, not surprisingly, is lower than that determined with the multi-sample approach.*

*Note that our approach is not new. It has been adopted by previous studies for measuring other chemical variables (e.g. Zafiriou et al., 2008; Xie et al., 2009).*

9. DOM analyses. CDOM spectral slope in the range 300-500 nm (S300-500 in nm-1) is reported in the supplementary material (Table S1) but is not really discussed in the manuscript. Also, in addition to S300-500 I would recommend the determination and examination of S275-295, proposed by Helms et al. (2008) and largely used yet. It could bring significant information about CDOM molecular weight and transformation processes.

*AR:* *The purpose of providing the $S_{300-500}$ in the Supplemental Material, as stated in the manuscript, is to facilitate the reader to compare results from different studies.*

*The spectral slope and slope ratio ($S_{275-295}$, $S_{350-400}$ and $S_R$) were also investigated and they showed similar patterns to those of $E_2/E_3$. $E_2/E_3$ was chosen, because 1) it exhibited larger variations than the spectral slopes and slope ratio; 2) it has been used as a valid proxy of molecular weight for a much longer history (De Haan, 1983; Peuravuori and Pihlaja, 1997) than the spectral slope and slope ratio, particularly for fresh and brackish waters (including estuarine waters); 3) it is very sensitive to and quantitatively responds to photobleaching (Lou and Xie, 2006; Qi et al., 2018); 4) a quantitative and validated relationship between $E_2/E_3$ and the molecular weight (MW) of CDOM is available (Lou and Xie, 2006; Qi et al., 2018), so that this relationship can be used to estimate the MW of CDOM for the present study (line 439-443 in the original manuscript). Note that such a broadly applicable relationship has not been established between $S_{275-295}$ and MW.*

*We have explicitly stated in the revised manuscript that $E_2/E_3$ serves similar functions to those of $S_{275-295}$.*

10. DOM analyses. HIX, BIX and E2/E3 should be defined in this section and not in the results section.

*AR: Revised according to the reviewer's suggestion.*

11. Results. The number of Tables is quite high. I recommend adding some in the supplementary material: Tables 1, 2, 4, 5.

*AR: Tables, 1, 4, and 5 were moved to the Supplemental Material.*

12. Results. Besides salinity, are ancillary parameters available for this sampling (i.e., dissolved oxygen, nutrients, chlorophyll,...) that could help the interpretation of the DOM dynamics?

*AR: No oxygen data is available. Other ancillary data were collected by other groups and we cannot explicitly publish them. However, we have now performed a principal component analysis (PCA) that includes nutrients, chlorophyll a, suspended particulate matter, etc. to further help interpret the DOM dynamics. Please see response to comment 14 below.*

13. Results. I find there is a lack of use of statistical analyses. For example, ANOVA, t test, Mann Whithney test,... (depending on the normal distribution or not of samples) could be applied to determine statistical differences in the DOM concentrations between seasons, surface/bottom,....

*AR: ANOVA has now been conducted. The result indicates that 1) there were no significant bottom-surface differences in both DOC and $a_{330}$; 2) DOC presented small but significant seasonal variability, while $a_{330}$ lacked significant seasonal difference, which further strengthens our conclusion that the spatial and temporal variability of DOM in the saltier zone of the PRE is smaller than expected for a sizable estuary with a marked seasonality of river runoff. This result was added to the end of section 3.2 in the Results.*

14. Moreover, instead of separate a priori the samples by seasons and looking at differences between these seasons (that do not necessarily represent/reflect different hydrological or

meteorological events which have occurred during the sampling period), it could be also interesting to apply multi-way statistical methods (principal component analysis, hierarchical ascendant classification,...) on all samples regardless of their sampling period. This could lead to different clustering of samples and underline particular processes affecting DOM dynamics, such as the impact of the mixing between marine and river waters, the impact of precipitation/runoff/river flow rate (ex: discrimination between samples collected in dry period and samples collected wet period), which could be obviously independent from seasons.

*AR: Our results have clearly demonstrated that physical mixing (i.e. salinity) is the predominant factor controlling the variability of DOM in the PRE (Figs. 3 and 4). Here we performed a principal component analysis (PCA) on the all-season dataset that includes variables in addition to salinity, such as water temperature, chl-a, nutrients, suspended particulate matter, and freshwater discharge rate. The DOM dynamics is represented by CDOM absorption at 330 nm ($a_{330}$) and DOC concentration. The first two axes of the PCA explained >74% of the variability in the dataset. Using the first axis on the following graph, one can see that DOC and $a_{330}$, along with a bunch of other variables (e.g. nitrate, nitrite, silicate, chl-a), are strongly negatively correlated to salinity, which is a typical indication of a conservative mixing behavior. In contrast, DOC and $a_{330}$ are only weakly (negatively) linked to the freshwater discharge rate, again consistent with our result (line 604-606 & Fig. S9 in the original version).*

*As the PCA does not bring much new information on the DOM dynamics, we have added the plot to the Supplemental Material (instead of the main text) and briefly discussed it (i.e. reinforcing the conclusion already reached) in the revised manuscript.*

[Figure]

*Figure: PCA analysis based on the all-season dataset. SPM: suspended particulate matter; $PO_4^{3-}$: phosphate; $NO_2^-$: nitrite; DOC: dissolved organic carbon; $a_{CDOM}(330)$: CDOM absorption coefficient at 330 nm; $NO_3^-$: nitrate; Chla: chlorophyll a; $SiO_4^{4-}$: silicate; discharge: freshwater discharge rate.*

15. Discussion. Lines 600-614: "[DOC] and [CDOM] in the PRE are the lowest among the major world rivers..." This is indeed intriguing. Why DOC and CDOM contents are so low in the PRE. In this part, the authors should also include the assumption of a DOM loss by bacterial degradation and photochemistry.

**AR:** *We have demonstrated that bacterial uptake and photodegradation led to only minor losses of DOM in the saltier zone (usually at salinity >5) of the PRE due largely to the short residence time of freshwater in the estuary and the completion for light absorption by other optical constituents in the case photodegradation (line 492-509 in the original version). The manuscript proposed two main factors to explain the low DOM in the PRE: the poorly forested watershed and rapid bacterial DOM consumption in the upper reach of the estuary (salinity <5) (line 600-604).*

16. Discussion. Line 604: "The lack of correspondence between [DOC]* and a330* and the freshwater discharge rate (Fig. S9) suggests that [DOM] in the PRE be controlled by both soil leaching and pollution input". Here could be also added the hypothesis of in situ autochthonous

DOM production from phytoplankton activities, which are generally not negligible in rivers.

*AR: Good idea. This proposition was added.*

**References cited in this response:**

De Haan, H., 1983. Use of ultraviolet spectroscopy, gel filtration, pyrolysis/mass spectrometry and numbers of benzoate metabolizing bacteria in the study of humification and degradation of aquatic organic matter. In: Christman, R.F., Gjessing, E.T. (Eds.), Aquatic and Terrestrial Humic Materials. Ann Arbor Science, Michigan, pp. 165–182.

Lou, T., Xie, H., 2006. Photochemical alteration of the molecular weight of dissolved organic matter. Chemosphere 65, 2333–2342.

Peuravuori, J., Pihlaja, K., 1997. Molecular size distribution and spectroscopic properties of aquatic humic substances. Anal. Chim. Acta 337, 133–149.

Qi, L., Xie, H., Gagné, J.P., Chaillou, G., Massicotte, P. and Yang, G.P., 2018. Photoreactivities of two distinct dissolved organic matter pools in groundwater of a subarctic island. Marine Chemistry, 202, 97-120.

Xie, H., Bélanger, S., Demers, S., Vincent, W.F. and Papakyriakou, T.N., 2009. Photobiogeochemical cycling of carbon monoxide in the southeastern Beaufort Sea in spring and autumn. Limnology and Oceanography, 54(1), 234-249.

Zafiriou, O.C., Xie, H., Nelson, N.B., Najjar, R.G. and Wang, W., 2008. Diel carbon monoxide cycling in the upper Sargasso Sea near Bermuda at the onset of spring and in midsummer. Limnology and Oceanography, 53(2), 835-850.

---

## Author Response (AR1)

**Response to Editor's comments**

**Responses are *italicized*.**
*AR* stands for authors' response

1. Main conclusions of the article are difficult to follow since there is repetition of results throughout the text, and there are results that are not considered in the discussion, deviating attention to main points of the article. Examples: a) Water temperature is shown but there is no discussion of it, b) idem with results on water column mixing, c) in page 263, "Bottom water salinity at most stations was nearly identical to SWS in January, slightly greater in May, moderately elevated in November, and much higher in August (Fig. S2)". There is no discussion of it in the text. If there is a meaning for this, then it needs to be quantitatively explained, not as currently written (slightly, much, etc.).

*AR: We have re-organized the structure of the article to minimize the repetition of the results. a) water temperature has now been incorporated into the principal component analysis (PCA) for discussion (lines 453-461); b) & c) the effect of water column mixing/stratification on the vertical distribution of DOM has now been briefly discussed (lines 486-491); c) this sentence has been modified (lines 253-256).*

2. There is an excessive use of Supplementary tables and figures around relevant discussion and conclusions. Supplementary figures and tables are meant to back up tables and figures of the main text. A new version will require rethinking and reorganizing tables and figures accordingly.

*AR: We have substantially reduced the supplementary tables and figures in the new version.*

3. Qualitative assessments should be avoided. such as saltier, less salty (Reviewer 3 suggests using well-known and accepted terminology by the estuarine community).

*AR: The "head region" is now used to refer to the narrow low-salinity zone and "main estuary" to denote the saltier zone.*

4. Hypothesis. "… hypothesize that DOM in the PRE presents substantial seasonal variability in terms of both abundance and chemical composition and that the PRE is an important source of DOM to global oceans. "Chemical composition you are referring to is targeting a quantitatively minor fraction of DOC pool (in the order of 2%), therefore you cannot test that hypothesis for the entire pool using this approach.

*AR: The hypothesis has been modified to "Given the large volume and seasonality of the freshwater discharge of the Pearl River, we hypothesize that the quantity of DOM and the quality of CDOM in the*

*PRE present substantial seasonal variability and that the PRE is an important source of DOM to the global ocean".*

5. What are units of DOC and CDOM fluxes in Table 6. Nowhere is mentioned how you estimated fluxes from absorbance data.

*AR: The units are already there: grams for DOC and $m^2$ for CDOM. The first 4 rows are for each season and the last row for one year. The equation and procedure for estimating the CDOM flux are already given in the original version (first paragraph of section 4.4).*

6. Keep in mind Short Comment:
" Although the manuscript is well written and reads easily, the way that sections are structure makes the manuscript repetitive when presenting and discussing results. I think it would become more concise and interesting if the authors focus on making a rearrangement of sections (by merging/condensing some of them) and on making a review through the text to avoid such repetitions. Additionally, the introduction is a bit too long and could be shortened by providing only information needed for interpretation of results from this study…."

*AR: Following the reviewer's comments, we have restructured and shortened the Introduction and Results sections.*

7. Section on Pearl River estuary is definitely too long, so it is background on DOM. Please choose the most relevant aspects.

*AR: Theses two sections have been restructured and shortened.*

8. "… [DOM], [CDOM], and [FDOM] stand for the abundances of…". Square brackets are used in chemistry to denote concentration and [CDOM] and [FDOM] are not; they could be considered proxies of concentration. Different things.

*AR: Now ⟨CDOM⟩ and ⟨FDOM⟩ are used to denote the proxies of CDOM and FDOM abundances.*

9. Use of non-standard acronym such as SWS only makes reading more difficult (It is used only 7 times in the text, all in one page).

*AR: This acronym has now been spelled out throughput the text.*

10. P, 286, P 409, etc.. Correlation and regression are not the same. In correlation there is no independent variable and coefficient of correlation (r) ranges from -1 to +1. In regression, there is X and Y, and coefficient of determination (R2) ranges from 0 to 1 (0 to 100%). Please check and revise accordingly

*AR: This has been checked and revised.*

11. Method. "Hansell's low carbon ([DOC]: 1–2 μmol L−1) and deep Florida Strait ([DOC]: 41–44 μmol L−1) reference waters "
What was the quantitatively results of this calibration?

*AR: The calibration results have been added to the revised version (lines 153-154)*

12. About the analytical uncertainty mentioned by Reviewer 2. #8. " … aCDOM at 330 nm (a330) was 2.19 m−1 (range: 1.19–4.37 m−1)…" corresponds to the range of values of a330 measured in the river during the August cruise. Analytical uncertainty on the other hand, deals with dispersion of values associated to a measure and, therefore samples has to be as similar as possible.

*AR: Now the uncertainty of measurements on 6 replicates of the same sample is reported. (lines 160-163).*

13. Lines 375-376. Please explain what you want to say here

*AR: This sentence does not exist anymore in the revised version.*

14. Lines 235-236 should be in methods

*AR: Now moved to the Methods (lines 225-227).*

**Response to Reviewer 1**

**Responses are *italicized*.**
*AR* stands for authors' response

The paper entitled "Distribution, seasonality, optical characteristics, and fluxes of dis- solved organic matter (DOM) in the Pearl River (Zhujiang) estuary, China" investigated seasonal and spatial variations of CDOM and FDOM characterized by absorption and fluorescence spectroscopy. Since I am an organic geochemist focusing on the organic carbon and nitrogen cycling mechanism in estuarine coastal zones and the role of microbes during the organic matter cycling, I am very familiar with the topic of this manuscript. This manuscript identified the compositional characteristics and sources of DOM. The main conclusion is that (i) microbial inputs and anthropogenic inputs are important sources of DOM in the freshwater end; (ii) small seasonal variations with respect to DOC and CDOM; and (iii) PR exports the lowest quantality of DOC among 30 large world rivers, although the size of PR watershed ranked the thirteenth largest in the world by area. Considering the anthropogenic activities can influence the quality and quantity of DOM in aquatic ecosystems and urbanization trends continue in response to human population growth, anthropogenic influences on DOM composition will likely become more widespread. Such human effects on DOM quality could have strong impacts on carbon cycles and need to be better understood. Therefore, this study provides a typical case study to approach the scientific questions mentioned above. However, some points need to be addressed as follows. Nevertheless, this work did provide interesting findings, and the data is reasonably strong to make the conclusions, and there I suggest a moderate revision needs to perform before the acceptance of this manuscript.

General comments:

1. In terms of English, I suggest the writing should be improved further.

*AR: We did further language polishing.*

2. The description of "overview of DOM" is great. However, I realize that it is too general. I hope the authors could provide introduction related with their discussion or the questions that need to be solved (or knowledge gap). In addition, the transition from 1.1 to 1.2 seems not that smooth to me.

3. The chapter "1.2 The Pearl River estuary (PRE)" is too lengthy to describe the important focus and question, and some of descriptions can be moved to "Site description", otherwise part of the information seems duplicated. For instance, the authors spent 9 paragraphs to describe the PRE, and some of the information is not closely related with the results/discussions. This needs to be shortened and be questions oriented.

*AR: Re comments 2&3. The introduction has now been restructured and shortened.*

4. The authors mentioned precipitation is an important factor affecting soil flushing, which may affect both DOM equality and quantity. It would be great if the author could incorporate some monthly or seasonal precipitation data to support their claims. In particular, the article indicated the terrigenous DOM is the main source of investigated areas, but it did not describe the influences of land runoff and rainfall on seasonal variations of DOM.

*AR: The freshwater discharge to the PRE, which has already been described in the paper, is directly correlated to precipitation over its watershed and is a more direct indicator of the impact of precipitation (than precipitation itself) on the study area.*

*Note that the article does not conclude that terrigenous DOM is the main source of DOM in the PRE. Instead, it underscores the microbial nature of this DOM pool and a potentially important contribution from river-borne DOM (line 462-471 in the original version).*

5. In this manuscript the author suggested that the low DOC concentrations in PRE (especially the low salinity region) was affected by biological degradation (due to input of labile DOM) and low inputs due to the low forest cover. This is a good point! I suggest the author expand this description a little bit. For instance, (i) the addition of labile DOM may "prime" the degradation of terrestrial (relatively more recalcitrant) DOM; (ii) the author could specify the land use percentages of the PR watershed and compare it with the other large river-estuarine systems (such as the Amazon River). Some of the land use% data has been organized in Wagner et al. (2015), and I believe the land use% data is not that difficult to find for PR watershed; (iii) since the authors claim that the PRE is a super eutrophic system, it would be interesting at least present some nutrient data (from literatures) to further support their main findings.

*AR: (i) The "priming" concept is a good suggestion. Nonetheless, our results indicate that this effect, if any, was minor, at least in May, August, and January. In the low-salinity section, the [DOC] after the rapid removal of the labile constituents (Fig. 3), except November, was in the same range as that of the*

*background [DOC] reported for the Pearl River upstream of the Pearl River Delta (114-137 uM, line 122 and line 465-466 in the original version), demonstrating little "priming". Downstream of the upper reach, [DOC] either decreased (August and January) or remained roughly constant (May and November) with increasing salinity, again disproving a major DOC loss process caused by priming. We believe that the land-derived DOC in the Pearl River is either priming-resistant or the short residence times of freshwater in the PRE (a few days, line 496-498 in the original version) prevented a significant priming effect from occurring.*

*In the revised manuscript, we have briefly discussed the potential role of the priming effect, particularly for November when the [DOC] at the downstream side of the low-salinity section was substantially lower than the land-derived background [DOC].*

*(ii) Sorry, we exhausted our resources but could not find the land use% data for the Pearl River region. The landscape information reported by Luo et al. (2002), which we cited, though in a more general nature, provides a similar support for the relevant discussion.*

*(iii) We thoroughly checked the manuscript and found that **nowhere** does the article claim the PRE to be a super eutrophic system. The word "eutrophic" does not exist in this article.*

6. I really like the main findings in the manuscript, but these findings are not well reflected in the abstract. I suggest the author re-organize their abstracts and focusing on the main findings. Reporting numbers are great, but there seem to be too many. Keep the important ones would be good enough.

*AR: We reorganized the abstract by emphasizing the major findings and reducing numbers.*

7. Considering the author spent a huge effort collecting all these samples, it would be very interesting to perform some statistical analysis such as the principal component analysis (PCA) to further confirm the major controls to the DOM variability across the whole dataset.

*AR: Our results have clearly demonstrated that physical mixing (i.e. salinity) is the predominant factor controlling the variability of DOM in the PRE (Figs. 3 and 4). Here we performed a principal component analysis (PCA) on the all-season dataset that includes variables in addition to salinity, such as water temperature, chl-a, nutrients, suspended particulate matter, and freshwater discharge rate. The DOM dynamics is represented by CDOM absorption at 330 nm ($a_{330}$) and DOC concentration. The first two axes of the PCA explained ~74% of the variability in the dataset. Using the first axis on the following graph, one can see that DOC and $a_{330}$ (along with nitrate and silicate) are strongly*

*negatively correlated to salinity, which is a typical indication of a conservative mixing behavior. In contrast, DOC and a_330 are only weakly linked to the freshwater discharge rate, again consistent with our result (line 604-606 & Fig. S9 in the original version).*

*We have added the plot to the main text (Fig. 9) and briefly discussed it in the revised manuscript (lines 453-461).*

[Figure]

*Figure: PCA analysis based on the all-season dataset. SPM: suspended particulate matter; $PO_4^{3-}$: phosphate; $NO_2^-$: nitrite; DOC: dissolved organic carbon; $a_{CDOM}(330)$: CDOM absorption coefficient at 330 nm; $NO_3^-$: nitrate; Chla: chlorophyll a; $SiO_4^{4-}$: silicate; discharge: freshwater discharge rate.*

Specific comments:

1. There was no explanation about the inverse changes of BIX and HIX in Fig.7

**AR:** *This is self-evident according to the definitions of BIX and HIX (now in the Methods section): BIX denotes the relative contribution of fresh, microbial-derived FDOM, while HIX signifies the degree of humification, with old, humified FDOM having higher HIX values.*

*Now a statement as follows has been added in the second last paragraph of section 3.5:*

*"BIX displayed a distribution roughly inverse to that of HIX (Fig. 7d), as can be inferred their definitions (Sect. 2.3)."*

2. I suggest the author make it clear what is "the saltier zone" because this is a ambiguous description.

*AR: The saltier zone is indirectly defined between line 358 and 361 in the original version. It refers to the zone with salinity generally >5, where the reported DOM variables showed much slower changes with increasing salinity as compared to the rapid changes near the head of the estuary (i.e. the low-salinity zone). However, the salinity separating these two areas was at times slightly season- and/or variable-specific.*

*Following relevant comments from reviewer 3 and the associate editor, we have now termed the low-salinity zone as the head region of the estuary and the saltier zone as the main estuary.*

3. Considering there are way too many tables. I suggest move some of the tables (e.g., Table 1) to the supplementary information. The DOC (μmol L-1) needs to be moved to the second column.

*AR: Tables 1, 4, and 5 were moved to Supplemental Material. DOC was moved to the second column in Table 8.*

4. Would be wonderful if the author could point out the major metropolitan areas (or even land use patterns) in Figure 1 since it closely related with the major discussions in this manuscript.

*AR: As stated in our response to comment#5, we could not find the land use data for this region. The major cities are already labeled. The discussion does not require information on the metropolitan borderlines.*

5. When the authors describe each PARAFAC component, I suggest the author use DOM Open- fluor database to compare the components in this study with literature data. Murphy, K. R., Stedmon, C. A., Wenig, P., & Bro, R. (2014). OpenFluor–an online spectral library of auto-fluorescence by organic compounds in the environment. Analytical Methods, 6(3), 658-661.

*AR: This has now been done and added to the Methods section.*

6. R.U. should be defined in the abstract.

*AR: Thanks. Done.*

**Response to Reviewer 2**

**Responses are *italicized*.**
**AR** stands for authors' response

This paper deals with the seasonal variability, spatial distribution, transformation processes and fluxes of dissolved organic matter (DOM) in the Pearl River estuary (PRE) in China. DOM is investigated through dissolved organic carbon (DOC), chromophoric (CDOM) and fluorescent (FDOM) dissolved organic matter. Overall, this work provides relevant results and good quality data concerning the dynamics and fluxes of DOM in the PRE. The manuscript is well structured, quite well written, and is obviously within the scope of Biogeosciences. Therefore, I recommend the paper to be published in Biogeosciences after "moderate" revisions. Below my comments:

1. Title. The part "optical characteristics" could be removed from the title.

*AR: "optical characteristics" was removed.*

2. Although English is not bad, the manuscript could benefit from corrections of an English native speaker.

*AR: The language has been further polished.*

3. The abstract has to be substantially improved. It does not reflect at all the relevance of the study. For instance, the following part: "The seasonality of average DOM abun- dance varied as follows: DOC: May (156 µmol L−1) > January (114 µmol L−1) ~ August (112 µmol L−1) > November (86 µmol L−1); CDOM absorption at 330 nm: Au- gust (1.76 m−1) > November (1.39 m−1) ~ January (1.30 m−1); FDOM expressed as the sum of the maximum fluorescence intensities of all FDOM components: November (1.77 R.U.) > August (1.54 R.U.) ~ January (1.49 27 R.U.). Average DOM abundance in surface water was higher than in bottom water, their difference being marginal (0.1– 10%) for DOC in all seasons and for CDOM and FDOM in November and January, and moderate (16– 21%) for CDOM and FDOM in August" did not deserve to be included in the abstract.

*AR: We reorganized the abstract by emphasizing the major findings and reducing numbers.*

4. Introduction. Subtitles ("1.1 Overview of DOM", "1.2 The Pear River estuary", "1.3 Hypothesis and objectives") should be removed. Usually there is no subtitle in the introduction. The first part concerning DOM is OK but the second one (PRE) is too long and too detailed. Most of these details should go in the "2 Methods" part, in a "2.1 Study area" section, which currently does not exist by the way. Only information about PRE that is useful for highlighting the problematic and hypothesis is necessary in the Introduction.

*AR: The Introduction has been re-arranged and shortened. Details of the PRE are moved to a separate section (2.1. Site description) in the Methods.*

5. Introduction. The sentence: "The biogeochemical and optical significance of DOM depends on both its abundance and quality (i.e. chemical composition), with the latter strongly linked to its origin of formation" is not clear. Please re-phrase.

*AR: This sentence does not exist anymore in the revised Introduction.*

6. Sample collection. I guess the number of samples collected at each season for DOM analyses is not mentioned. This should be mentioned here.

*AR: Stating the number of samples does not provide extra essential information, since the numbers of sampling stations and depths are already reported.*

7. The subtitle "2.2 Sample analysis" should be replaced by "2.2. DOM "analysis"

*AR: Changed to "DOM analysis".*

8. DOM analyses. "The analytical uncertainty of aCDOM measurement was assessed by analyzing six pairs of duplicate samples collected from the August cruise. Average aCDOM at 330 nm (a330) was 2.19 m−1 (range: 1.19–4.37 m−1); the average difference in each pair was 0.07 ± 0.05 m−1, or 3.0% ± 1.4%." This method for assessing the analytical uncertainty (precision?) is not clear to me. Why using six pairs of duplicates? I would have used six replicates (of the same sample). The values "0.07 ± 0.05 m−1, or 3.0% ± 1.4%" is not pertinent.

*AR: Now the uncertainty of measurements on 6 replicates of the same sample is reported. (lines 160-163).*

9. DOM analyses. CDOM spectral slope in the range 300-500 nm (S300-500 in nm-1) is reported in the supplementary material (Table S1) but is not really discussed in the manuscript. Also, in addition to

S300-500 I would recommend the determination and examination of S275-295, proposed by Helms et al. (2008) and largely used yet. It could bring significant information about CDOM molecular weight and transformation processes.

*AR: The purpose of providing the $S_{300-500}$ in the Supplemental Material, as stated in the manuscript, is to facilitate the reader to compare results from different studies.*

*The spectral slope and slope ratio ($S_{275-295}$, $S_{350-400}$ and $S_R$) were also investigated and they showed similar patterns to those of $E_2/E_3$. $E_2/E_3$ was chosen, because 1) it exhibited larger variations than the spectral slopes and slope ratio; 2) it has been used as a valid proxy of molecular weight for a much longer history (De Haan, 1983; Peuravuori and Pihlaja, 1997) than the spectral slope and slope ratio, particularly for fresh and brackish waters (including estuarine waters); 3) it is very sensitive to and quantitatively responds to photobleaching (Lou and Xie, 2006; Qi et al., 2018); 4) a quantitative and validated relationship between $E_2/E_3$ and the molecular weight (MW) of CDOM is available (Lou and Xie, 2006; Qi et al., 2018), so that this relationship can be used to estimate the MW of CDOM for the present study (line 439-443 in the original manuscript). Note that such a broadly applicable relationship has not been established between $S_{275-295}$ and MW.*

*We have explicitly stated in the revised manuscript that $E_2/E_3$ serves similar functions to those of $S_{275-295}$ (lines 205-210).*

10. DOM analyses. HIX, BIX and E2/E3 should be defined in this section and not in the results section.

*AR: Revised according to the reviewer's suggestion.*

11. Results. The number of Tables is quite high. I recommend adding some in the supplementary material: Tables 1, 2, 4, 5.

*AR: Tables, 1, 4, and 5 were moved to the Supplemental Material.*

12. Results. Besides salinity, are ancillary parameters available for this sampling (i.e., dissolved oxygen, nutrients, chlorophyll,...) that could help the interpretation of the DOM dynamics?

*AR: No oxygen data is available. Other ancillary data were collected by other groups and we cannot explicitly publish them. However, we have now performed a principal component analysis (PCA) that includes nutrients, chlorophyll a, suspended particulate matter, etc. to further help interpret the DOM*

*dynamics. Please see response to comment 14 below.*

13. Results. I find there is a lack of use of statistical analyses. For example, ANOVA, t test, Mann Whithney test,... (depending on the normal distribution or not of samples) could be applied to determine statistical differences in the DOM concentrations between seasons, surface/bottom,....

*AR: ANOVA and t-test have been conducted. The results indicate that 1) there were no significant bottom-surface differences in both DOC and $a_{330}$; 2) DOC presented small but significant seasonal variability, while $a_{330}$ lacked significant seasonal difference, which further strengthens our conclusion that the spatial and temporal variability of DOM in the saltier zone of the PRE is smaller than expected for a sizable estuary with a marked seasonality of river runoff. The results of ANOVA and t-test are incorporated into the Results section.*

14. Moreover, instead of separate a priori the samples by seasons and looking at differences between these seasons (that do not necessarily represent/reflect different hydrological or meteorological events which have occurred during the sampling period), it could be also interesting to apply multi-way statistical methods (principal component analysis, hierarchical ascendant classification,...) on all samples regardless of their sampling period. This could lead to different clustering of samples and underline particular processes affecting DOM dynamics, such as the impact of the mixing between marine and river waters, the impact of precipitation/runoff/river flow rate (ex: discrimination between samples collected in dry period and samples collected wet period), which could be obviously independent from seasons.

*AR: Our results have clearly demonstrated that physical mixing (i.e. salinity) is the predominant factor controlling the variability of DOM in the PRE (Figs. 3 and 4). Here we performed a principal component analysis (PCA) on the all-season dataset that includes variables in addition to salinity, such as water temperature, chl-a, nutrients, suspended particulate matter, and freshwater discharge rate. The DOM dynamics is represented by CDOM absorption at 330 nm ($a_{330}$) and DOC concentration. The first two axes of the PCA explained ~74% of the variability in the dataset. Using the first axis on the following graph, one can see that DOC and $a_{330}$ (along with nitrate and silicate) are strongly negatively correlated to salinity, which is a typical indication of a conservative mixing behavior. In contrast, DOC and $a_{330}$ are only weakly linked to the freshwater discharge rate, again consistent with our result (line 604-606 & Fig. S9 in the original version).*

*We have added the plot to the main text (Fig. 9) and briefly discussed it in the revised manuscript (lines*

*453-461).*

[Figure]

*Figure: PCA analysis based on the all-season dataset. SPM: suspended particulate matter; $PO_4^{3-}$: phosphate; $NO_2^-$: nitrite; DOC: dissolved organic carbon; $a_{CDOM}(330)$: CDOM absorption coefficient at 330 nm; $NO_3^-$: nitrate; Chla: chlorophyll a; $SiO_4^{4-}$: silicate; discharge: freshwater discharge rate.*

15. Discussion. Lines 600-614: "[DOC] and [CDOM] in the PRE are the lowest among the major world rivers..." This is indeed intriguing. Why DOC and CDOM contents are so low in the PRE. In this part, the authors should also include the assumption of a DOM loss by bacterial degradation and photochemistry.

*AR: We have demonstrated that bacterial uptake and photodegradation led to only minor losses of DOM in the saltier zone (usually at salinity >5) of the PRE due largely to the short residence time of freshwater in the estuary and the completion for light absorption by other optical constituents in the case photodegradation (line 492-509 in the original version). The manuscript proposed two main factors to explain the low DOM in the PRE: the poorly forested watershed and rapid bacterial DOM consumption in the upper reach of the estuary (salinity <5) (line 600-604).*

16. Discussion. Line 604: "The lack of correspondence between [DOC]* and a330* and the freshwater discharge rate (Fig. S9) suggests that [DOM] in the PRE be controlled by both soil leaching and pollution input". Here could be also added the hypothesis of in situ autochthonous DOM production from phytoplankton activities, which are generally not negligible in rivers.

*AR: Good idea. A river-born component (from phytoplankton and/or bacterial activities) is added to this proposition (lines 568-570).*

**References cited in this response:**

De Haan, H., 1983. Use of ultraviolet spectroscopy, gel filtration, pyrolysis/mass spectrometry and numbers of benzoate metabolizing bacteria in the study of humification and degradation of aquatic organic matter. In: Christman, R.F., Gjessing, E.T. (Eds.), Aquatic and Terrestrial Humic Materials. Ann Arbor Science, Michigan, pp. 165–182.

Lou, T., Xie, H., 2006. Photochemical alteration of the molecular weight of dissolved organic matter. Chemosphere 65, 2333–2342.

Peuravuori, J., Pihlaja, K., 1997. Molecular size distribution and spectroscopic properties of aquatic humic substances. Anal. Chim. Acta 337, 133–149.

Qi, L., Xie, H., Gagné, J.P., Chaillou, G., Massicotte, P. and Yang, G.P., 2018. Photoreactivities of two distinct dissolved organic matter pools in groundwater of a subarctic island. Marine Chemistry, 202, 97-120.

Xie, H., Bélanger, S., Demers, S., Vincent, W.F. and Papakyriakou, T.N., 2009. Photobiogeochemical cycling of carbon monoxide in the southeastern Beaufort Sea in spring and autumn. Limnology and Oceanography, 54(1), 234-249.

Zafiriou, O.C., Xie, H., Nelson, N.B., Najjar, R.G. and Wang, W., 2008. Diel carbon monoxide cycling in the upper Sargasso Sea near Bermuda at the onset of spring and in midsummer. Limnology and Oceanography, 53(2), 835-850.

**Response to Reviewer 3**

**Responses are *italicized*.**
**AR** stands for authors' response

This work presents the seasonal distribution (May, Aug, Nov, and Jan 2015) of DOM (DOC concentrations, CDOM absorption and CDOM fluorescent components (from PARAFAC analysis) in Pearl River estuary (PRE), China. DOC concentrations and CDOM absorption and fluorescence properties (and their qualitative metrics) were examined in relation to salinity as well as to each other. In addition, fluxes of DOC and CDOM from the PRE to South China Sea were also estimated. Overall, results of this study provides new insights into the seasonal DOC and optical properties of CDOM in PRE. In comparison, most previous studies have mainly reported one or two field campaigns, while this study comprised a more seasonal study (four field campaigns).

However, the analysis of the data throughout involves simple correlation analysis and is descriptive with no rigorous analysis of field data (spatial analysis, precipitation, chlorophyll and turbidity measurements that were indicated in the text to have been measured). The additional analysis would support a better understanding of the sources and sinks related to the DOM in PRE.

*AR: All the discussion and conclusions are based on the quantitative analysis of the data. Our results have clearly demonstrated that physical mixing (i.e. salinity) is the predominant factor controlling the variability of DOM in the PRE (Figs. 3 and 4). We have now added a principal component analysis (PCA) on the all-season dataset to further strengthening the manuscript. The PCA includes variables in addition to salinity, such as water temperature, chl-a, nutrients, suspended particulate matter, and freshwater discharge rate. The DOM dynamics is represented by CDOM absorption at 330 nm ($a_{330}$) and DOC concentration. The first two axes of the PCA explained ~74% of the variability in the dataset (see graph below). Using the first axis on the following graph, one can see that DOC and $a_{330}$ (along with nitrate and silicate), are strongly negatively correlated to salinity, which is a typical indication of a conservative mixing behavior. In contrast, DOC and $a_{330}$ are only weakly linked to the freshwater discharge rate, again consistent with our result (line 604-606 & Fig. S9 in the original version).*

*We have added the plot to the main text (Fig. 9) and briefly discussed it in the revised manuscript (lines 453-461).*

[Figure]

*Figure: PCA analysis based on the all-season dataset. SPM: suspended particulate matter; $PO_4^{3-}$: phosphate; $NO_2^-$: nitrite; DOC: dissolved organic carbon; $a_{CDOM}(330)$: CDOM absorption coefficient at 330 nm; $NO_3^-$: nitrate; Chla: chlorophyll a; $SiO_4^{4-}$: silicate; discharge: freshwater discharge rate.*

I find that the manuscript needs further improvements and the authors should address some major concerns/suggestions before the paper can be accepted for publication.

Major comments/suggestions: 1) There are various major sources of freshwater to the PRE. Previous studies have also indicated spatial differences in the surface and bottom properties in CDOM optical properties (absorption coefficients and spectral slope; e.g., Lei et al. 2018). Furthermore, seasonal analysis of DOC (Ye et al. 2018) indicated strong seasonality in DOC with substantial removal of DOC in the salinity range 5-22. I think a more comprehensive analysis using all the available data (e.g., chlorophyll, turbidity, etc) including spatial distribution plots (surface and bottom) would greatly help in supporting the conclusions of this study.

*AR: Our conclusions are based on an analysis of not only quantitative variables ([DOC], $a_{CDOM}$, and 5 FDOM components) but also a large number of qualitative variables ($E_2/E_3$, BIX, HIX, and the percentages of FDOM components). The more comprehensive data analysis (including chlorophyll and SPM) using PCA shown above further strengthens the conclusions already reached in our article.*

*The difference between the studies the reviewer mentioned and ours may be caused by different spatiotemporal coverage of water sampling and potentially large interannual variability of the DOM dynamics in the PRE, as already suggested in the original manuscript (line 131-141; line 548-553 in the original version). In the revised manuscript, we reinforced this point by including the very recent reference suggested by the reviewer (i.e. Ye et al., 2018; the paper by Lei et al. (2018) was already cited). Note that the potential interannual variability further complicates the generalization of the DOM dynamics and biogeochemical cycling in the PRE.*

2) Throughout this study the authors describe the data collected in the main estuary as the saltier zone as opposed to fresh water zone. I think a more traditional separation of the zones (e.g., Cai et al. 2004; upstream region, estuary, outer estuary) would be more appropriate and could better support the results of this study.

*AR: The "head region" is now used to refer to the narrow low-salinity zone and "main estuary" to denote the saltier zone.*

3) The absorption coefficient at 330 nm used in this study has not generally been used and therefore not easily comparable to other studies. Although Table S1 includes some of these wavelengths, it would help if the authors replace the absorption at 330 nm with another commonly used wavelength. Also the spectral slope between 275-295 nm is now generally used to assess CDOM properties and should be included in the analysis.

*AR: There are several points to support the use of the wavelength of 330 nm for $a_{CDOM}$. First, the wavelength at or close to 330 nm is where the majority of aquatic CDOM photoreactions (including photobleaching) exhibits the maximum rates in surface waters under solar radiation (e.g. Vähätalo et al., 2000; Zhang et al., 2006; Osburn et al., 2009; Xie et al., 2009, 2012; White et al., 2010; Song et al., 2013; Hong et al., 2014; Qi et al., 2018). The wavelength of 330 nm is, therefore, is linked to an important process controlling the cycling of CDOM in natural waters. This point has now been explicitly stated in the revised manuscript. Second, $a_{CDOM}(330)$ has been used as an indicator of CDOM content by many labs including those well recognized labs (e.g. Brisco and Ziegler, 2004; White et al., 2008; Osburn et al., 2009; Xie et al., 2009; Gareis et al., 2010; Mann et al., 2012; Song et al., 2017; Qi et al., 2018). Third, there is no consensus on which wavelength is best to serves as a proxy of CDOM content. A limited review of the literature shows at least 13 wavelengths (254, 300, 320, 325, 330, 350, 355, 375, 380, 400, 412, 420, and 440 nm) have been adopted for this purpose.*

*Finally, in case the reader is interested in other wavelengths, we have provided absorption coefficients at 6 other wavelengths across the UV and visible regimes that are commonly seen as well in the literature (Table S1 in the Supplemental Material). Furthermore, we also published the spectral slope between 300 and 500 nm (again in Table S1), so that the reader can retrieve the absorption coefficient at any wavelength between the 300 and 500 nm interval. We believe we have done our best to accommodate the different needs of the scientific community.*

*The spectral slope and slope ratio ($S_{275-295}$, $S_{350-400}$ and $S_R$) were also investigated and they showed similar patterns to those of $E_2/E_3$. $E_2/E_3$ was chosen, because 1) it exhibited larger variations than the spectral slopes and slope ratio; 2) it has been used as a valid proxy of molecular weight for a much longer history (De Haan, 1983; Peuravuori and Pihlaja, 1997) than the spectral slope and slope ratio, particularly for fresh and brackish waters (including estuarine waters); 3) it is very sensitive to and quantitatively responds to photobleaching (Lou and Xie, 2006; Qi et al., 2018) and biogeochemical processing; 4) a quantitative and validated relationship between $E_2/E_3$ and the molecular weight (MW) of CDOM is available (Lou and Xie, 2006; Qi et al., 2018), so that this relationship can be used to estimate the MW of CDOM for the present study (line 439-443 in the original manuscript). Note that such a broadly applicable relationship has not been established between $S_{275-295}$ and MW.*

*We have explicitly stated in the revised manuscript that $E_2/E_3$ serves similar functions to those of $S_{275-295}$ (lines 205-210).*

4) CDOM generally is a good optical proxy for DOC, especially in estuaries. Also, CDOM undergoes rapid photobleaching in the estuaries or the coastal waters. It may not be useful include estimates of CDOM fluxes at 330 mn from the estuary to the SCS, especially since the wavelength used is so unique to this study.

*AR: For the wavelength issue, we think we have chosen an appropriate wavelength to represent CDOM content and photobleaching and (see our response to comment 3).*

*Even if CDOM degrades rapidly in estuaries and coastal waters (often that's not true, see below), it does not necessarily imply that the export of CDOM to the ocean is not important. If the remaining component of CDOM exported to the ocean, albeit small in amount, is bio- and photo-resistant, it can accumulate in open oceans. This is why the oceanographic community has put tremendous efforts in identifying and quantifying potential terrigenous DOM (the main part of it could be CDOM) in open oceans (Opsahl and Benner, 1997; Cauwet, 2002; Raymond et al., 2007; Bianchi and Allison, 2009;*

*Dai et al., 2012; Wang et al., 2012; Raymond and Spencer, 2015). This issue is fundamental for understanding the global carbon cycle. This is in part why (other aspects involve ocean optics) scientists have started making efforts to evaluate the land-to-ocean CDOM fluxes (e.g. Stedmon et al., 2011; Spencer et al., 2013; Aarnos et al., 2018).*

*Concerning the specific case of the PRE, our data clearly indicate that CDOM behaved essentially conservatively in the main estuary (i.e. ca. salinity >5), implying that photobleaching was insignificant. We also made a direct estimate of the amount of CDOM that could be removed by photobleaching in the PRE; it was at most 7% (line 487-507 in the original version), supporting the inference from the conservative CDOM vs. salinity plots. This not surprising, given that 1) the residence time of freshwater (and thus CDOM as well) in the PRE is very short (a few days, line 494-497 in the original version; 2) the competition of light absorption by particles (water in the PRE is turbid); and 3) self-shading due to high CDOM and particle abundances in the PRE.*

*In general, estuaries and strongly runoff-impacted coastal waters are not prone to having efficient CDOM photobleaching due to at least the three causes stated above. Efficient photobleaching usually takes place in waters on the outer shelf (e.g. shelf break) where CDOM has been sufficiently spread out and the majority of the particles have settled down to the seafloor (so that self-shading is diminished).*

5) It may be useful to look at meteorological data (e.g., wind field) to see if mixing played a role in reducing the variability in DOM surface and bottom properties.

**AR:** *It is the salinity and temperature structures (Figs. S1 and S2), not the meteorological information, that **directly** indicate the degree of water column mixing. We used the salinity and temperature data to discuss the surface and bottom variability on each relevant occasion.*

Minor comments: -No indication of how salinity was measured -Methods section could describe the study site rather than in the Introduction.

**AR:** *It is already there (see line 182-183 in the original version).*

References: X. Lei, J. pan, A. T. Devlin. 2018. Mixing behavior of chromophoric dissolved organic matter in the Pearl River Estuary in sprig. Continental Shelf Research, 154, 46-54.

F. Ye, W. Guo, G. Wei, and G. Jia. 2018. The sources and transformations of dissolved organic matter in the Pearl River Estuary, China, as revealed by stable isotopes. J. Geophys. Res.: Oceans, 123, 6893-6908.

*AR: Thanks for providing these two references. Lei et al (2018) was already cited in the original manuscript. Ye et al (2018) has now been added.*

**References cited in this response:**

Aarnos, H., Gélinas, Y., Kasurinen, V., Gu, Y., Puupponen, V.M. and Vähätalo, A.V., 2018. Photochemical mineralization of terrigenous DOC to dissolved inorganic carbon in ocean. Global Biogeochemical Cycles, 32(2), 250-266.

Bianchi, T.S. and Allison, M.A., 2009. Large-river delta-front estuaries as natural "recorders" of global environmental change. Proceedings of the National Academy of Sciences, 106(20), 8085-8092.

Brisco, S. and Ziegler, S., 2004. Effects of solar radiation on the utilization of dissolved organic matter (DOM) from two headwater streams. Aquatic microbial ecology, 37(2), 197-208.

Cauwet, G., 2002. DOM in the coastal zone. Biogeochemistry of marine dissolved organic matter.

Dai, M., Yin, Z., Meng, F., Liu, Q. and Cai, W.J., 2012. Spatial distribution of riverine DOC inputs to the ocean: an updated global synthesis. Current Opinion in Environmental Sustainability, 4(2), 170-178.

De Haan, H., 1983. Use of ultraviolet spectroscopy, gel filtration, pyrolysis/mass spectrometry and numbers of benzoate metabolizing bacteria in the study of humification and degradation of aquatic organic matter. In: Christman, R.F., Gjessing, E.T. (Eds.), Aquatic and Terrestrial Humic Materials. Ann Arbor Science, Michigan, pp. 165–182.

Gareis, J.A., Lesack, L.F. and Bothwell, M.L., 2010. Attenuation of in situ UV radiation in Mackenzie Delta lakes with varying dissolved organic matter compositions. Water Resources Research, 46(9).

Hong, J., Xie, H., Guo, L. and Song, G., 2014. Carbon monoxide photoproduction: implications for photoreactivity of arctic permafrost-derived soil dissolved organic matter. Environmental science & technology, 48(16), 9113-9121.

Lou, T., Xie, H., 2006. Photochemical alteration of the molecular weight of dissolved organic matter. Chemosphere 65, 2333–2342.

Mann, P.J., Davydova, A., Zimov, N., Spencer, R.G.M., Davydov, S., Bulygina, E., Zimov, S. and Holmes, R.M., 2012. Controls on the composition and lability of dissolved organic matter in Siberia's Kolyma River basin. Journal of Geophysical Research: Biogeosciences, 117(G1).

Opsahl, S. and Benner, R., 1998. Photochemical reactivity of dissolved lignin in river and ocean waters. Limnology and Oceanography, 43(6), pp.1297-1304.

Osburn, C.L., Retamal, L. and Vincent, W.F., 2009. Photoreactivity of chromophoric dissolved organic matter transported by the Mackenzie River to the Beaufort Sea. Marine Chemistry, 115(1-2), 10-20.

Peuravuori, J., Pihlaja, K., 1997. Molecular size distribution and spectroscopic properties of aquatic humic substances. Anal. Chim. Acta 337, 133–149.

Qi, L., Xie, H., Gagné, J.P., Chaillou, G., Massicotte, P. and Yang, G.P., 2018. Photoreactivities of two distinct dissolved organic matter pools in groundwater of a subarctic island. Marine Chemistry, 202, 97-120.

Raymond, P.A. and Spencer, R.G., 2015. Riverine DOM. In Biogeochemistry of marine dissolved organic matter (pp. 509-533). Academic Press.

Raymond, P.A., McClelland, J.W., Holmes, R.M., Zhulidov, A.V., Mull, K., Peterson, B.J., Striegl, R.G., Aiken, G.R. and Gurtovaya, T.Y., 2007. Flux and age of dissolved organic carbon exported to the Arctic Ocean: A carbon isotopic study of the five largest arctic rivers. Global Biogeochemical Cycles, 21(4).

Song, G., Li, Y., Hu, S., Li, G., Zhao, R., Sun, X. and Xie, H., 2017. Photobleaching of chromophoric dissolved organic matter (CDOM) in the Yangtze River estuary: kinetics and effects of temperature, pH, and salinity. Environmental Science: Processes & Impacts, 19(6), 861-873.

Song, G., Xie, H., Bélanger, S., Leymarie, E. and Babin, M., 2013. Spectrally resolved efficiencies of carbon monoxide (CO) photoproduction in the western Canadian Arctic: particles versus solutes. Biogeosciences, 10(6), 3731-3748.

Spencer, R. G. M., Aiken, G. R., Dornblaser, M. M., Butler, K. D., Holmes, R. M., Fiske, G., Mann, P. J., and Stubbins, A.: Chromophoric dissolved organic matter export from U.S. rivers, Geophys. Res. Lett., 40, 1575–1579, doi:10.1029/grl50357, 2013.

Stedmon, C. A., Amon, R. M. W., Rinehart, A. J., and Walker, S. A.: The supply and characteristics of colored dissolved organic matter (CDOM) in the Arctic Ocean: Pan Arcitc trends and differences, Mar. Chem., 124, 108–118, 2011.

Vähätalo, A.V., Salkinoja‑Salonen, M., Taalas, P. and Salonen, K., 2000. Spectrum of the quantum yield for photochemical mineralization of dissolved organic carbon in a humic lake. Limnology and Oceanography, 45(3), 664-676.

Wang, X., Ma, H., Li, R., Song, Z. and Wu, J., 2012. Seasonal fluxes and source variation of organic carbon transported by two major Chinese Rivers: The Yellow River and Changjiang (Yangtze) River. Global Biogeochemical Cycles, 26(2).

White, E.M., Kieber, D.J. and Mopper, K., 2008. Determination of photochemically produced carbon dioxide in seawater. Limnology and Oceanography: Methods, 6(9), 441-453.

White, E.M., Kieber, D.J., Sherrard, J., Miller, W.L. and Mopper, K., 2010. Carbon dioxide and carbon monoxide photoproduction quantum yields in the Delaware Estuary. Marine Chemistry, 118(1-2), 11-21.

Xie, H., Bélanger, S., Demers, S., Vincent, W.F. and Papakyriakou, T.N., 2009. Photobiogeochemical cycling of carbon monoxide in the southeastern Beaufort Sea in spring and autumn. Limnology and Oceanography, 54(1), 234-249.

Xie, H., Bélanger, S., Song, G., Benner, R., Taalba, A., Blais, M., Tremblay, J.É. and Babin, M., 2012. Photoproduction of ammonium in the southeastern Beaufort Sea and its biogeochemical implications. Biogeosciences, 9(8), 3047-3061.

Zhang, Y., Xie, H. and Chen, G., 2006. Factors affecting the efficiency of carbon monoxide photoproduction in the St. Lawrence estuarine system (Canada). Environmental science &

technology, 40(24), 7771-7777.

**Responses are *italicized*.**
*AR* stands for authors' response

**SC:** Dissolved organic matter is an important component of the carbon cycle in aquatic systems and it exerts direct impact on the overall biogeochemical process in the ocean. DOM spectroscopy has emerged as a cost-effective and easy-to-measure technique for quantifying and, more recently, qualify the DOM content in the environment. The manuscript by Li and colleagues brings results on DOM amount (expressed by means of DOC and spectroscopic measurements), characterization (through EEM- PARAFAC), fluxes and seasonal variability for the Pearl River Estuary, China. The data set is robust and the methods applied align with current literature. Although the sampling grid remains the same for the different seasons, the seasonal averages presented in the MS might be biased by the spatial variability presented within the water masses spatial distribution within the region. Therefore, I suggest the authors to have lead the MS through a more "oceanographic point of view", i.e., by investigating the seasonal changes within the water masses presented within the region.

*AR: We adopted the classical approaches for describing chemical variables in an estuary: property vs. distance and property vs. salinity. Salinity is an indication of mixing processes, while distance is more related to residence time and processing time. These two approaches are complementary. The seasonal averages presented in our MS are based on the "distance" approach, given that the coordinates of the sampling stations were the same for different seasons. These averages thus reflect the seasonality of the residence and processing times of the water masses in the estuary. On the other hand, the property vs. salinity plots provided information on how the mixing behavior of a variable of interest changed seasonally. As water masses in an estuary are primarily defined by salinity, the seasonal variability revealed by this approach is essentially water mass-based. A more complete picture of the seasonality of the variables is acquired by combining the results from the distance and salinity approaches. This is the rationale behind the scheme we employed to present our data.*

*As our sampling stations were principally distributed along the main longitudinal axis of the estuary with little lateral coverage (as is true for many other estuarine studies), the data thus collected is insufficient to characterize the spatial distribution of water masses in the region, making the "oceanographic point of view" approach suggested by the reviewer difficult to implement.*

**SC:** Although the manuscript is well written and reads easily, the way that sections are structure makes the manuscript repetitive when presenting and discussing results. I think it would become more concise and interesting if the authors focus on making a rearrangement of sections (by merging/condensing some of them) and on making a review through the text to avoid such repetitions. Additionally, the introduction is a bit too long and could be shortened by providing only information needed for interpretation of results from this study. Thus, to my judgment, the manuscript may be publishable after major reviews.

*AR: Following the reviewer's comments, we have restructured and shortened the Introduction and Results sections.*

GENERAL COMMENTS:

**SC:** The abstract does not clearly illustrate the main findings obtained in the study.

*AR: We have shortened and rewritten the abstract to focus on the main findings.*

**SC:** The hypothesis presented in section 1.3 seem weak and vague, and could be sharper. Seasonal variability in DOM flux is already expected from an estuary with marked seasonal variability in freshwater export, as documented by the authors.

*AR: DOM flux is only one of the many DOM variables (both quantitative and qualitative) reported in this study. In fact, most other variables showed smaller spatial and seasonal variations than expected from this sizable estuary with an important seasonal fluctuation of freshwater discharge (see the Conclusions section). The fluxes of DOC and CDOM are also the lowest compared to other major world rivers, contrasting with the hypothesis. Therefore, we feel that the current working hypothesis is appropriate and strong enough.*

**SC:** Sampling strategy: why was decided to collect the "deep water" sample near the bottom and not below the pycnocline? It can be affected by sediment resuspension, if there is any.

*AR: One of the purposes of this study was to determine if there was a significant sedimentary impact on DOM in the water column. The consistent property–salinity patterns (Figures 3 and 4) and lack of relationship with suspended particle concentration (Line 512 in the original version and now the PCA*

*analysis as well) suggest that this effect was minor. Note that the effect of sediment resuspension, if any, could reach the depths just below the pycnocline, given the overall shallow water depths of the PRE (mostly <10 m, Table 1 in the original version)*

**SC:** Have the authors looked at the CDOM absorption spectral slope and slope ratio? It could provide more insights into the photochemical reactions along the estuarine mixing.

*AR: The spectral slope and slope ratio ($S_{275-295}$, $S_{350-400}$ and $S_R$) were also investigated and they showed similar patterns to those of $E_2/E_3$. $E_2/E_3$ was chosen, because 1) it exhibited larger variations than the spectral slopes and slope ratio; 2) it has been used as a valid proxy of molecular weight for a much longer history (De Haan, 1983; Peuravuori and Pihlaja, 1997) than the spectral slope and slope ratio, particularly for fresh and brackish waters (including estuarine waters); 3) it is very sensitive to and quantitatively responds to photobleaching (Lou and Xie, 2006; Qi et al., 2018) and biogeochemical processing; 4) a quantitative and validated relationship between $E_2/E_3$ and the molecular weight (MW) of CDOM is available (Lou and Xie, 2006; Qi et al., 2018), so that this relationship can be used to estimate the MW of CDOM for the present study (line 439-443 in the original manuscript). Note that such a broadly applicable relationship has not been established between $S_{275-295}$ and MW.*

*We have explicitly stated in the revised manuscript that $E_2/E_3$ serves similar functions to those of $S_{275-295}$ (lines 205-210).*

**SC:** The authors could also try to use multivariate analysis (e.g., PCA) to analyze the variability between the campaigns (i.e., over time) and to elucidate what are the main drivers on DOM variability within the region.

*AR: Our results have clearly demonstrated that physical mixing (i.e. salinity) is the predominant factor controlling the variability of DOM in the PRE (Figs. 3 and 4). Here we performed a principal component analysis (PCA) on the all-season dataset that includes variables in addition to salinity, such as water temperature, chl-a, nutrients, suspended particulate matter, and freshwater discharge rate. The DOM dynamics is represented by CDOM absorption at 330 nm ($a_{330}$) and DOC concentration. The first two axes of the PCA explained ~74% of the variability in the dataset. Using the first axis on the following graph, one can see that DOC and $a_{330}$ (along with nitrate and silicate) are strongly negatively correlated to salinity, which is a typical indication of a conservative mixing behavior. In contrast, DOC and $a_{330}$ are only weakly linked to the freshwater discharge rate, again consistent with our result (line 604-606 & Fig. S9 in the original version).*

*We have added the plot to the main text (Fig. 9) and briefly discussed it in the revised manuscript (lines 453-461).*

[Figure]

*Figure: PCA analysis based on the all-season dataset. SPM: suspended particulate matter; $PO_4^{3-}$: phosphate; $NO_2^-$: nitrite; DOC: dissolved organic carbon; $a_{CDOM}(330)$: CDOM absorption coefficient at 330 nm; $NO_3^-$: nitrate; Chla: chlorophyll a; $SiO_4^{4-}$: silicate; discharge: freshwater discharge rate.*

**SC:** I suggest the authors to compare their PARAFAC-derived components spectra with the OpenFluor database (https://openfluor.lablicate.com/). This would benefit the comparison established with other studies along the MS.

*AR: This has now been done and added to the Methods section.*

**SC:** With respect to the sources of DOM to region, especially the pollution-derived DOM, they could be more stressed along the MS. It is not totally clear how the findings of this study support that.

*AR: Pollution-derived DOM is a dominant source of DOM in the upper reach of the PRE, generally upstream of Humen. Note that this is **not** our finding, rather a conclusion of previous studies (as clearly stated in the Introduction, line 120-130 in the original version). Some previous studies (e.g. Lin et al., 2007; He et al., 2010) conducted sampling much farther upstream into the Guangzhou Channel, where the capital of the Guangdong Province is located. The concentration of DOC in that channel*

*could reach as high as 500 uM, which is ~4 times the background DOC (119 uM) in the Pearl River upstream of the Pearl River Delta (He, 2010). This observation, combined with the enormous amount of industrial and domestic waste discharged into the PRE ($5.8*10^9$ tons/year) across its deltaic region, led these authors to concluding that the highly enriched DOC in the upper reach of the estuary mostly originates form sewage effluents. The pollution-derived DOC is, however, very labile and much of it is consumed by bacteria in the low-salinity zone of the estuary (He, 2010, He et al., 2010). Our data provided two lines of evidence to support the pollution argument for our sampling seasons: 1) a rapid drawdown of DOC and CDOM in the upper reach, which is consistent with the labile character of pollution-derived DOM as elucidated in the previous studies; 2) the protein-rich character of this DOM pool as revealed by the fluorescence-based metrics (BIX and %(C1+C5)). These two points are elaborated in the relevant context (section 4.1).*

**SC:** Section 4.5 establishes comparisons among global DOM studies but I expected the discussion to bring some conclusions on the reason for such differences rather than just comparing them.

*AR: We are a bit confused by this comment. Section 4.5 clearly indicates that two factors mainly contribute to the lowest DOM abundance and flux in the PRE: 1) the deficiency of organic matter in soil of the Pearl River's watershed having almost no forest; 2) the rapid microbial consumption of pollution-derived DOM in the upper estuary. These two factors are once again emphasized in the Conclusions section. Moreover, the main portion of section 4.5 is discussion instead of "just comparison".*

SPECIFIC COMMENTS:

**SC:** L75-79: authors could give more background on anthropogenic/pollution-derived DOM, given that it is a DOM source for the region, as pointed out in this study.

*AR: This point is actually brought up on two other occasions in the Introduction about the PRE (line 122-125; line 145-148 in the original version). We believe the background information for this point is sufficient, particularly considering that the Introduction is already long and needs to be shortened.*

**SC:** L115-119: Please present values (ranges) for the variables. How much does the phytoplankton biomass vary within the seasons?

*AR: The Introduction is greatly shortened and this kind of non-essential information is not provided in the revised version in part because different papers reported widely different values and in part because we conducted a PCA that includes the chl-a values from our cruises.*

**SC:** L124-125: Are there only those two studies supporting this affirmation? No study published in English?

*AR: After re-searching the literature, we found one more paper (He et al., 2010, published in English) for supporting this argument. This reference has now been added.*

**SC:** L306-307: what do the authors mean by "freshwater input from this river appeared to have little influence on [DOC]" ?

*AR: Sta. M01, 02 and 03 were distributed along a transect across the three outlets of the East River (i.e. upper, middle, and lower outlets, Fig. 1). However, the [DOC]s at these three stations in May were nearly constant, suggesting that the freshwater input from the East River did not significantly affect the [DOC]. This further implies that [DOC] in the East River in May was roughly equal to that in the North River, which is the larger freshwater source of the upper reach of the PRE (~2 times that of the East River, line 95-98 in the Introduction).*

*The revised manuscript does not contain this content anymore in order to restructure and condense the Results section.*

**SC:** L500-503: Missing references.

*AR: Thanks. The missing reference (He, 2010) was added.*

**SC:** L522-526: I found the explanation for different mixing behavior weak and should be discussed more in deep.

*AR: The observation needs to be explained: In the saltier zone, [DOC] remained rather constant while [CDOM] (in terms of $a_{330}$) decreased linearly with increasing salinity in November; in August and January, [CDOM] decreased much faster than [DOC] with increasing salinity.*

*Our explanation: 1) CDOM was only a minor component of the entire DOM pool (so that the change in [CDOM] had little impact on [DOC]); 2) the marine endmember was less colored (i.e. lower $a_{CDOM}$) than the freshwater endmember (so that [CDOM] decreased with increasing salinity); 3) the difference*

*between the marine and freshwater DOC endmembers was much smaller than that for CDOM (so that the salinity-based gradient for [DOC] was much smaller than that for [CDOM]). A combination of points 2 and 3 leads to a smaller [DOC]-normalized $a_{CDOM}$ for the marine endmember than that for the freshwater endmember (which is what we presented in the manuscript).*

*We believe that our explanation is sound. These points are made clearer in the revised version.*

**SC:** L527-535: this paragraph/discussion could be deepened in the sense to explain the reasons for such variations.

*AR: This paragraph is actually a summary of section 4.2. The deeper discussion is presented in the preceding paragraphs. Moreover, the lack of sampling within the main freshwater outlets (e.g. Hengmen, Jiaomen, Hongqimen) downstream of Humen prevents us from further discussing the potential impact of different freshwater masses.*

**SC:** L538-547: Why does it only have good correlations for summer and winter? What happens with the correlations during the other seasons? Additionally, was the DOC- aCDOM correlation significant and strong? I ask that, because that correlation does not hold true for several environments.

*AR: In spring and fall, [DOC] in the saltier zone was relatively constant and consequently not correlated with salinity as opposed to the case in summer and winter. $a_{CDOM}$, however, showed negative correlations with salinity in all three sampling seasons (summer, fall, and winter). This distribution pattern is already described in section 3.4 and discussed in section 4.2, and thus not repeated in section 4.3. Instead, we referred the reader to Fig. 3 for understanding the relevant context.*

*Yes, the DOC-$a_{CDOM}$ is significant and strong (p<0.0001, now added to the text). Although this kind of correlation may not hold universally, many marine environments, include estuaries and coastal waters, do exhibit such correlations, e.g. the Middle Atlantic Bight (Del Vecchio and Blough, 2004), Yukon River (Spencer et al., 2009), Yangtze River estuary (Guo et al., 2014), and the Baltic coastal sea (Harvey et al., 2015).*

**SC:** L556-580: authors could deepen the discussion regarding the fluxes.

*AR: More discussion about the fluxes is provided in section 4.5.*

**SC:** L615-623: what could the authors point out as the reason for such differences?

*AR: This is because the [DOC] and [CDOM] in the PRE are the lowest among the world major rivers. Line 600-6004 in the original version has already speculated on two factors causing this phenomenon: the poorly forested watershed of the Pearl River and the rapid degradation of sewage-derived DOM.*

**SC:** Figure 1: It would be interesting to have two panel composing this figure: one with the sampling sites and another with the city names and also the main circulation patterns.

*AR: As the circulation pattern changes with season, which needs four panels to do it. Moreover, the distributional pattern of the sampling stations (an along-estuary transect without much cross-estuary coverage) does not allow us to adequately characterize the circulation patterns during our sampling periods. Hence, adding a circulation pattern panel may not significantly improve the presentation and interpretation of the data.*

**SC:** Figs 3, 4, 5 and 8: please present the curve fits and stats.

*AR: Lines in Figure 5 denote the conservative mixing lines, not the data fits. The curve fits and statistics are already presented in Table 4 for Figures 3 and 4 and in Table 5 for Figure 8 in the original manuscript.*

**References cited in this response:**

De Haan, H., 1983. Use of ultraviolet spectroscopy, gel filtration, pyrolysis/mass spectrometry and numbers of benzoate metabolizing bacteria in the study of humification and degradation of aquatic organic matter. In: Christman, R.F., Gjessing, E.T. (Eds.), Aquatic and Terrestrial Humic Materials. Ann Arbor Science, Michigan, pp. 165–182.

Del Vecchio, R. and Blough, N.V., 2004. Spatial and seasonal distribution of chromophoric dissolved organic matter and dissolved organic carbon in the Middle Atlantic Bight. Marine Chemistry, 89(1-4), 169-187.

Guo, W., Yang, L., Zhai, W., Chen, W., Osburn, C.L., Huang, X. and Li, Y., 2014. Runoff‐mediated seasonal oscillation in the dynamics of dissolved organic matter in different branches of a large bifurcated estuary−The Changjiang Estuary. Journal of Geophysical Research: Biogeosciences, 119(5), 776-793.

Harvey, E.T., Kratzer, S. and Andersson, A., 2015. Relationships between colored dissolved organic matter and dissolved organic carbon in different coastal gradients of the Baltic Sea. Ambio, 44(3), 392-401.

He, B., Dai, M., Zhai, W., Wang, L., Wang, K., Chen, J., Lin, J., Hua, A., and Xu, Y.: Distribution, degradation and dynamics of dissolved organic carbon and its major compound classes in the pearl river estuary, China, Mar. Chem., 119, 52–64, 2010.

He, B.: Organic Matter in the Pearl River Estuary: its Composition, Source, Distribution, Bioactivity and their Linkage to Oxygen Depletion (Ph.D. Dissertation), Xiamen university, 2010 (In Chinese).

Lin, J.: On the behavior and flux of Dissolved Organic Carbon in two large Chinese estuaries-Changjiang and Zhujiang (Master Dissertation), Xiamen university, 2007 (In Chinese).

Lou, T., Xie, H., 2006. Photochemical alteration of the molecular weight of dissolved organic matter. Chemosphere 65, 2333–2342.

Peuravuori, J., Pihlaja, K., 1997. Molecular size distribution and spectroscopic properties of aquatic humic substances. Anal. Chim. Acta 337, 133–149.

Qi, L., Xie, H., Gagné, J.P., Chaillou, G., Massicotte, P. and Yang, G.P., 2018. Photoreactivities of two distinct dissolved organic matter pools in groundwater of a subarctic island. Marine Chemistry, 202, 97-120.

Spencer, R.G., Aiken, G.R., Butler, K.D., Dornblaser, M.M., Striegl, R.G. and Hernes, P.J., 2009. Utilizing chromophoric dissolved organic matter measurements to derive export and reactivity of dissolved organic carbon exported to the Arctic Ocean: A case study of the Yukon River, Alaska. Geophysical Research Letters, 36(6).

[revised manuscript text omitted]

**Response to Editor's comments**

**Responses are *italicized*.**
*AR* stands for authors' response

1. Main conclusions of the article are difficult to follow since there is repetition of results throughout the text, and there are results that are not considered in the discussion, deviating attention to main points of the article. Examples: a) Water temperature is shown but there is no discussion of it, b) idem with results on water column mixing, c) in page 263, "Bottom water salinity at most stations was nearly identical to SWS in January, slightly greater in May, moderately elevated in November, and much higher in August (Fig. S2)". There is no discussion of it in the text. If there is a meaning for this, then it needs to be quantitatively explained, not as currently written (slightly, much, etc.).

*AR: We have re-organized the structure of the article to minimize the repetition of the results. a) water temperature has now been incorporated into the principal component analysis (PCA) for discussion (lines 453-461); b) & c) the effect of water column mixing/stratification on the vertical distribution of DOM has now been briefly discussed (lines 486-491); c) this sentence has been modified (lines 253-256).*

2. There is an excessive use of Supplementary tables and figures around relevant discussion and conclusions. Supplementary figures and tables are meant to back up tables and figures of the main text. A new version will require rethinking and reorganizing tables and figures accordingly.

*AR: We have substantially reduced the supplementary tables and figures in the new version.*

3. Qualitative assessments should be avoided. such as saltier, less salty (Reviewer 3 suggests using well-known and accepted terminology by the estuarine community).

*AR: The "head region" is now used to refer to the narrow low-salinity zone and "main estuary" to denote the saltier zone.*

4. Hypothesis. "… hypothesize that DOM in the PRE presents substantial seasonal variability in terms of both abundance and chemical composition and that the PRE is an important source of DOM to global oceans. "Chemical composition you are referring to is targeting a quantitatively minor fraction of DOC pool (in the order of 2%), therefore you cannot test that hypothesis for the entire pool using this approach.

*AR: The hypothesis has been modified to "Given the large volume and seasonality of the freshwater discharge of the Pearl River, we hypothesize that the quantity of DOM and the quality of CDOM in the*

*PRE present substantial seasonal variability and that the PRE is an important source of DOM to the global ocean".*

5. What are units of DOC and CDOM fluxes in Table 6. Nowhere is mentioned how you estimated fluxes from absorbance data.

*AR: The units are already there: grams for DOC and $m^2$ for CDOM. The first 4 rows are for each season and the last row for one year. The equation and procedure for estimating the CDOM flux are already given in the original version (first paragraph of section 4.4).*

6. Keep in mind Short Comment:
" Although the manuscript is well written and reads easily, the way that sections are structure makes the manuscript repetitive when presenting and discussing results. I think it would become more concise and interesting if the authors focus on making a rearrangement of sections (by merging/condensing some of them) and on making a review through the text to avoid such repetitions. Additionally, the introduction is a bit too long and could be shortened by providing only information needed for interpretation of results from this study…."

*AR: Following the reviewer's comments, we have restructured and shortened the Introduction and Results sections.*

7. Section on Pearl River estuary is definitely too long, so it is background on DOM. Please choose the most relevant aspects.

*AR: Theses two sections have been restructured and shortened.*

8. "… [DOM], [CDOM], and [FDOM] stand for the abundances of…". Square brackets are used in chemistry to denote concentration and [CDOM] and [FDOM] are not; they could be considered proxies of concentration. Different things.

*AR: Now ⟨CDOM⟩ and ⟨FDOM⟩ are used to denote the proxies of CDOM and FDOM abundances.*

9. Use of non-standard acronym such as SWS only makes reading more difficult (It is used only 7 times in the text, all in one page).

*AR: This acronym has now been spelled out throughput the text.*

10. P, 286, P 409, etc.. Correlation and regression are not the same. In correlation there is no independent variable and coefficient of correlation (r) ranges from -1 to +1. In regression, there is X and Y, and coefficient of determination (R2) ranges from 0 to 1 (0 to 100%). Please check and revise accordingly

*AR: This has been checked and revised.*

11. Method. "Hansell's low carbon ([DOC]: 1–2 µmol L−1) and deep Florida Strait
([DOC]: 41–44 µmol L−1) reference waters "
What was the quantitatively results of this calibration?

*AR: The calibration results have been added to the revised version (lines 153-154)*

12. About the analytical uncertainty mentioned by Reviewer 2. #8. " … aCDOM at
nm (a330) was 2.19 m−1 (range: 1.19–4.37 m−1)…" corresponds to the range of values of a330
measured in the river during the August cruise. Analytical uncertainty on the other hand, deals with
dispersion of values associated to a measure and, therefore samples has to be as similar as possible.

*AR: Now the uncertainty of measurements on 6 replicates of the same sample is reported. (lines 160-163).*

13. Lines 375-376. Please explain what you want to say here

*AR: This sentence does not exist anymore in the revised version.*

14. Lines 235-236 should be in methods

*AR: Now moved to the Methods (lines 225-227).*

**Responses are *italicized*.**
*AR* stands for authors' response

The paper entitled "Distribution, seasonality, optical characteristics, and fluxes of dis- solved organic matter (DOM) in the Pearl River (Zhujiang) estuary, China" investigated seasonal and spatial variations of CDOM and FDOM characterized by absorption and fluorescence spectroscopy. Since I am an organic geochemist focusing on the organic carbon and nitrogen cycling mechanism in estuarine coastal zones and the role of microbes during the organic matter cycling, I am very familiar with the topic of this manuscript. This manuscript identified the compositional characteristics and sources of DOM. The main conclusion is that (i) microbial inputs and anthropogenic inputs are important sources of DOM in the freshwater end; (ii) small seasonal variations with respect to DOC and CDOM; and (iii) PR exports the lowest quantality of DOC among 30 large world rivers, although the size of PR watershed ranked the thirteenth largest in the world by area. Considering the anthropogenic activities can influence the quality and quantity of DOM in aquatic ecosystems and urbanization trends continue in response to human population growth, anthropogenic influences on DOM composition will likely become more widespread. Such human effects on DOM quality could have strong impacts on carbon cycles and need to be better understood. Therefore, this study provides a typical case study to approach the scientific questions mentioned above. However, some points need to be addressed as follows. Nevertheless, this work did provide interesting findings, and the data is reasonably strong to make the conclusions, and there I suggest a moderate revision needs to perform before the acceptance of this manuscript.

General comments:

1. In terms of English, I suggest the writing should be improved further.

*AR: We did further language polishing.*

2. The description of "overview of DOM" is great. However, I realize that it is too general. I hope the authors could provide introduction related with their discussion or the questions that need to be solved (or knowledge gap). In addition, the transition from 1.1 to 1.2 seems not that smooth to me.

3. The chapter "1.2 The Pearl River estuary (PRE)" is too lengthy to describe the important focus and question, and some of descriptions can be moved to "Site description", otherwise part of the information seems duplicated. For instance, the authors spent 9 paragraphs to describe the PRE, and some of the information is not closely related with the results/discussions. This needs to be shortened and be questions oriented.

*AR: Re comments 2&3. The introduction has now been restructured and shortened.*

4. The authors mentioned precipitation is an important factor affecting soil flushing, which may affect both DOM equality and quantity. It would be great if the author could incorporate some monthly or seasonal precipitation data to support their claims. In particular, the article indicated the terrigenous DOM is the main source of investigated areas, but it did not describe the influences of land runoff and rainfall on seasonal variations of DOM.

*AR: The freshwater discharge to the PRE, which has already been described in the paper, is directly correlated to precipitation over its watershed and is a more direct indicator of the impact of precipitation (than precipitation itself) on the study area.*

*Note that the article does not conclude that terrigenous DOM is the main source of DOM in the PRE. Instead, it underscores the microbial nature of this DOM pool and a potentially important contribution from river-borne DOM (line 462-471 in the original version).*

5. In this manuscript the author suggested that the low DOC concentrations in PRE (especially the low salinity region) was affected by biological degradation (due to input of labile DOM) and low inputs due to the low forest cover. This is a good point! I suggest the author expand this description a little bit. For instance, (i) the addition of labile DOM may "prime" the degradation of terrestrial (relatively more recalcitrant) DOM; (ii) the author could specify the land use percentages of the PR watershed and compare it with the other large river-estuarine systems (such as the Amazon River). Some of the land use% data has been organized in Wagner et al. (2015), and I believe the land use% data is not that difficult to find for PR watershed; (iii) since the authors claim that the PRE is a super eutrophic system, it would be interesting at least present some nutrient data (from literatures) to further support their main findings.

*AR: (i) The "priming" concept is a good suggestion. Nonetheless, our results indicate that this effect, if any, was minor, at least in May, August, and January. In the low-salinity section, the [DOC] after the rapid removal of the labile constituents (Fig. 3), except November, was in the same range as that of the*

*background [DOC] reported for the Pearl River upstream of the Pearl River Delta (114-137 uM, line 122 and line 465-466 in the original version), demonstrating little "priming". Downstream of the upper reach, [DOC] either decreased (August and January) or remained roughly constant (May and November) with increasing salinity, again disproving a major DOC loss process caused by priming. We believe that the land-derived DOC in the Pearl River is either priming-resistant or the short residence times of freshwater in the PRE (a few days, line 496-498 in the original version) prevented a significant priming effect from occurring.*

*In the revised manuscript, we have briefly discussed the potential role of the priming effect, particularly for November when the [DOC] at the downstream side of the low-salinity section was substantially lower than the land-derived background [DOC].*

*(ii) Sorry, we exhausted our resources but could not find the land use% data for the Pearl River region. The landscape information reported by Luo et al. (2002), which we cited, though in a more general nature, provides a similar support for the relevant discussion.*

*(iii) We thoroughly checked the manuscript and found that **nowhere** does the article claim the PRE to be a super eutrophic system. The word "eutrophic" does not exist in this article.*

6. I really like the main findings in the manuscript, but these findings are not well reflected in the abstract. I suggest the author re-organize their abstracts and focusing on the main findings. Reporting numbers are great, but there seem to be too many. Keep the important ones would be good enough.

**AR:** *We reorganized the abstract by emphasizing the major findings and reducing numbers.*

7. Considering the author spent a huge effort collecting all these samples, it would be very interesting to perform some statistical analysis such as the principal component analysis (PCA) to further confirm the major controls to the DOM variability across the whole dataset.

**AR:** *Our results have clearly demonstrated that physical mixing (i.e. salinity) is the predominant factor controlling the variability of DOM in the PRE (Figs. 3 and 4). Here we performed a principal component analysis (PCA) on the all-season dataset that includes variables in addition to salinity, such as water temperature, chl-a, nutrients, suspended particulate matter, and freshwater discharge rate. The DOM dynamics is represented by CDOM absorption at 330 nm ($a_{330}$) and DOC concentration. The first two axes of the PCA explained ~74% of the variability in the dataset. Using the first axis on the following graph, one can see that DOC and $a_{330}$ (along with nitrate and silicate) are strongly*

*negatively correlated to salinity, which is a typical indication of a conservative mixing behavior. In contrast, DOC and a$_{330}$ are only weakly linked to the freshwater discharge rate, again consistent with our result (line 604-606 & Fig. S9 in the original version).*

*We have added the plot to the main text (Fig. 9) and briefly discussed it in the revised manuscript (lines 453-461).*

[Figure]

*Figure: PCA analysis based on the all-season dataset. SPM: suspended particulate matter; PO$_4^{3-}$: phosphate; NO$_2^-$: nitrite; DOC: dissolved organic carbon; a$_{CDOM}$(330): CDOM absorption coefficient at 330 nm; NO$_3^-$: nitrate; Chla: chlorophyll a; SiO$_4^{4-}$: silicate; discharge: freshwater discharge rate.*

Specific comments:

1. There was no explanation about the inverse changes of BIX and HIX in Fig.7

**AR:** *This is self-evident according to the definitions of BIX and HIX (now in the Methods section): BIX denotes the relative contribution of fresh, microbial-derived FDOM, while HIX signifies the degree of humification, with old, humified FDOM having higher HIX values.*

*Now a statement as follows has been added in the second last paragraph of section 3.5:*

*"BIX displayed a distribution roughly inverse to that of HIX (Fig. 7d), as can be inferred their definitions (Sect. 2.3)."*

2. I suggest the author make it clear what is "the saltier zone" because this is a ambiguous description.

*AR: The saltier zone is indirectly defined between line 358 and 361 in the original version. It refers to the zone with salinity generally >5, where the reported DOM variables showed much slower changes with increasing salinity as compared to the rapid changes near the head of the estuary (i.e. the low-salinity zone). However, the salinity separating these two areas was at times slightly season- and/or variable-specific.*

*Following relevant comments from reviewer 3 and the associate editor, we have now termed the low-salinity zone as the head region of the estuary and the saltier zone as the main estuary.*

3. Considering there are way too many tables. I suggest move some of the tables (e.g., Table 1) to the supplementary information. The DOC (μmol L-1) needs to be moved to the second column.

*AR: Tables 1, 4, and 5 were moved to Supplemental Material. DOC was moved to the second column in Table 8.*

4. Would be wonderful if the author could point out the major metropolitan areas (or even land use patterns) in Figure 1 since it closely related with the major discussions in this manuscript.

*AR: As stated in our response to comment#5, we could not find the land use data for this region. The major cities are already labeled. The discussion does not require information on the metropolitan borderlines.*

5. When the authors describe each PARAFAC component, I suggest the author use DOM Open- fluor database to compare the components in this study with literature data. Murphy, K. R., Stedmon, C. A., Wenig, P., & Bro, R. (2014). OpenFluor–an online spectral library of auto-fluorescence by organic compounds in the environment. Analytical Methods, 6(3), 658-661.

*AR: This has now been done and added to the Methods section.*

6. R.U. should be defined in the abstract.

*AR: Thanks. Done.*

**Response to Reviewer 2**

**Responses are** *italicized.*
**AR** stands for authors' response

This paper deals with the seasonal variability, spatial distribution, transformation processes and fluxes of dissolved organic matter (DOM) in the Pearl River estuary (PRE) in China. DOM is investigated through dissolved organic carbon (DOC), chromophoric (CDOM) and fluorescent (FDOM) dissolved organic matter. Overall, this work provides relevant results and good quality data concerning the dynamics and fluxes of DOM in the PRE. The manuscript is well structured, quite well written, and is obviously within the scope of Biogeosciences. Therefore, I recommend the paper to be published in Biogeosciences after "moderate" revisions. Below my comments:

1. Title. The part "optical characteristics" could be removed from the title.

*AR: "optical characteristics" was removed.*

2. Although English is not bad, the manuscript could benefit from corrections of an English native speaker.

*AR: The language has been further polished.*

3. The abstract has to be substantially improved. It does not reflect at all the relevance of the study. For instance, the following part: "The seasonality of average DOM abun- dance varied as follows: DOC: May (156 µmol L−1) > January (114 µmol L−1)    August (112 µmol L−1) > November (86 µmol L−1); CDOM absorption at 330 nm: Au- gust (1.76 m−1) > November (1.39 m−1)    January (1.30 m−1); FDOM expressed as the sum of the maximum fluorescence intensities of all FDOM components: November (1.77 R.U.) > August (1.54 R.U.)    January (1.49 27 R.U.). Average DOM abundance in surface water was higher than in bottom water, their difference being marginal (0.1– 10%) for DOC in all seasons and for CDOM and FDOM in November and January, and moderate (16– 21%) for CDOM and FDOM in August" did not deserve to be included in the abstract.

*AR: We reorganized the abstract by emphasizing the major findings and reducing numbers.*

4. Introduction. Subtitles ("1.1 Overview of DOM", "1.2 The Pear River estuary", "1.3 Hypothesis and objectives") should be removed. Usually there is no subtitle in the introduction. The first part concerning DOM is OK but the second one (PRE) is too long and too detailed. Most of these details should go in the "2 Methods" part, in a "2.1 Study area" section, which currently does not exist by the way. Only information about PRE that is useful for highlighting the problematic and hypothesis is necessary in the Introduction.

*AR: The Introduction has been re-arranged and shortened. Details of the PRE are moved to a separate section (2.1. Site description) in the Methods.*

5. Introduction. The sentence: "The biogeochemical and optical significance of DOM depends on both its abundance and quality (i.e. chemical composition), with the latter strongly linked to its origin of formation" is not clear. Please re-phrase.

*AR: This sentence does not exist anymore in the revised Introduction.*

6. Sample collection. I guess the number of samples collected at each season for DOM analyses is not mentioned. This should be mentioned here.

*AR: Stating the number of samples does not provide extra essential information, since the numbers of sampling stations and depths are already reported.*

7. The subtitle "2.2 Sample analysis" should be replaced by "2.2. DOM "analysis"

*AR: Changed to "DOM analysis".*

8. DOM analyses. "The analytical uncertainty of aCDOM measurement was assessed by analyzing six pairs of duplicate samples collected from the August cruise. Average aCDOM at 330 nm (a330) was 2.19 m−1 (range: 1.19–4.37 m−1); the average difference in each pair was 0.07 ± 0.05 m−1, or 3.0% ± 1.4%." This method for assessing the analytical uncertainty (precision?) is not clear to me. Why using six pairs of duplicates? I would have used six replicates (of the same sample). The values "0.07 ± 0.05 m−1, or 3.0% ± 1.4%" is not pertinent.

*AR: Now the uncertainty of measurements on 6 replicates of the same sample is reported. (lines 160-163).*

9. DOM analyses. CDOM spectral slope in the range 300-500 nm (S300-500 in nm-1) is reported in the supplementary material (Table S1) but is not really discussed in the manuscript. Also, in addition to

S300-500 I would recommend the determination and examination of S275-295, proposed by Helms et al. (2008) and largely used yet. It could bring significant information about CDOM molecular weight and transformation processes.

*AR: The purpose of providing the $S_{300-500}$ in the Supplemental Material, as stated in the manuscript, is to facilitate the reader to compare results from different studies.*

*The spectral slope and slope ratio ($S_{275-295}$, $S_{350-400}$ and $S_R$) were also investigated and they showed similar patterns to those of $E_2/E_3$. $E_2/E_3$ was chosen, because 1) it exhibited larger variations than the spectral slopes and slope ratio; 2) it has been used as a valid proxy of molecular weight for a much longer history (De Haan, 1983; Peuravuori and Pihlaja, 1997) than the spectral slope and slope ratio, particularly for fresh and brackish waters (including estuarine waters); 3) it is very sensitive to and quantitatively responds to photobleaching (Lou and Xie, 2006; Qi et al., 2018); 4) a quantitative and validated relationship between $E_2/E_3$ and the molecular weight (MW) of CDOM is available (Lou and Xie, 2006; Qi et al., 2018), so that this relationship can be used to estimate the MW of CDOM for the present study (line 439-443 in the original manuscript). Note that such a broadly applicable relationship has not been established between $S_{275-295}$ and MW.*

*We have explicitly stated in the revised manuscript that $E_2/E_3$ serves similar functions to those of $S_{275-295}$ (lines 205-210).*

10. DOM analyses. HIX, BIX and E2/E3 should be defined in this section and not in the results section.

*AR: Revised according to the reviewer's suggestion.*

11. Results. The number of Tables is quite high. I recommend adding some in the supplementary material: Tables 1, 2, 4, 5.

*AR: Tables, 1, 4, and 5 were moved to the Supplemental Material.*

12. Results. Besides salinity, are ancillary parameters available for this sampling (i.e., dissolved oxygen, nutrients, chlorophyll,...) that could help the interpretation of the DOM dynamics?

*AR: No oxygen data is available. Other ancillary data were collected by other groups and we cannot explicitly publish them. However, we have now performed a principal component analysis (PCA) that includes nutrients, chlorophyll a, suspended particulate matter, etc. to further help interpret the DOM*

*dynamics. Please see response to comment 14 below.*

13. Results. I find there is a lack of use of statistical analyses. For example, ANOVA, t test, Mann Whithney test,... (depending on the normal distribution or not of samples) could be applied to determine statistical differences in the DOM concentrations between seasons, surface/bottom,....

*AR: ANOVA and t-test have been conducted. The results indicate that 1) there were no significant bottom-surface differences in both DOC and $a_{330}$; 2) DOC presented small but significant seasonal variability, while $a_{330}$ lacked significant seasonal difference, which further strengthens our conclusion that the spatial and temporal variability of DOM in the saltier zone of the PRE is smaller than expected for a sizable estuary with a marked seasonality of river runoff. The results of ANOVA and t-test are incorporated into the Results section.*

14. Moreover, instead of separate a priori the samples by seasons and looking at differences between these seasons (that do not necessarily represent/reflect different hydrological or meteorological events which have occurred during the sampling period), it could be also interesting to apply multi-way statistical methods (principal component analysis, hierarchical ascendant classification,...) on all samples regardless of their sampling period. This could lead to different clustering of samples and underline particular processes affecting DOM dynamics, such as the impact of the mixing between marine and river waters, the impact of precipitation/runoff/river flow rate (ex: discrimination between samples collected in dry period and samples collected wet period), which could be obviously independent from seasons.

*AR: Our results have clearly demonstrated that physical mixing (i.e. salinity) is the predominant factor controlling the variability of DOM in the PRE (Figs. 3 and 4). Here we performed a principal component analysis (PCA) on the all-season dataset that includes variables in addition to salinity, such as water temperature, chl-a, nutrients, suspended particulate matter, and freshwater discharge rate. The DOM dynamics is represented by CDOM absorption at 330 nm ($a_{330}$) and DOC concentration. The first two axes of the PCA explained ~74% of the variability in the dataset. Using the first axis on the following graph, one can see that DOC and $a_{330}$ (along with nitrate and silicate) are strongly negatively correlated to salinity, which is a typical indication of a conservative mixing behavior. In contrast, DOC and $a_{330}$ are only weakly linked to the freshwater discharge rate, again consistent with our result (line 604-606 & Fig. S9 in the original version).*

*We have added the plot to the main text (Fig. 9) and briefly discussed it in the revised manuscript (lines*

*453-461).*

[Figure]

*Figure: PCA analysis based on the all-season dataset. SPM: suspended particulate matter; $PO_4^{3-}$: phosphate; $NO_2^-$: nitrite; DOC: dissolved organic carbon; $a_{CDOM}(330)$: CDOM absorption coefficient at 330 nm; $NO_3^-$: nitrate; Chla: chlorophyll a; $SiO_4^{4-}$: silicate; discharge: freshwater discharge rate.*

15. Discussion. Lines 600-614: "[DOC] and [CDOM] in the PRE are the lowest among the major world rivers..." This is indeed intriguing. Why DOC and CDOM contents are so low in the PRE. In this part, the authors should also include the assumption of a DOM loss by bacterial degradation and photochemistry.

*AR: We have demonstrated that bacterial uptake and photodegradation led to only minor losses of DOM in the saltier zone (usually at salinity >5) of the PRE due largely to the short residence time of freshwater in the estuary and the completion for light absorption by other optical constituents in the case photodegradation (line 492-509 in the original version). The manuscript proposed two main factors to explain the low DOM in the PRE: the poorly forested watershed and rapid bacterial DOM consumption in the upper reach of the estuary (salinity <5) (line 600-604).*

16. Discussion. Line 604: "The lack of correspondence between [DOC]* and a330* and the freshwater discharge rate (Fig. S9) suggests that [DOM] in the PRE be controlled by both soil leaching and pollution input". Here could be also added the hypothesis of in situ autochthonous DOM production from phytoplankton activities, which are generally not negligible in rivers.

*AR: Good idea. A river-born component (from phytoplankton and/or bacterial activities) is added to this proposition (lines 568-570).*

**References cited in this response:**

De Haan, H., 1983. Use of ultraviolet spectroscopy, gel filtration, pyrolysis/mass spectrometry and numbers of benzoate metabolizing bacteria in the study of humification and degradation of aquatic organic matter. In: Christman, R.F., Gjessing, E.T. (Eds.), Aquatic and Terrestrial Humic Materials. Ann Arbor Science, Michigan, pp. 165–182.

Lou, T., Xie, H., 2006. Photochemical alteration of the molecular weight of dissolved organic matter. Chemosphere 65, 2333–2342.

Peuravuori, J., Pihlaja, K., 1997. Molecular size distribution and spectroscopic properties of aquatic humic substances. Anal. Chim. Acta 337, 133–149.

Qi, L., Xie, H., Gagné, J.P., Chaillou, G., Massicotte, P. and Yang, G.P., 2018. Photoreactivities of two distinct dissolved organic matter pools in groundwater of a subarctic island. Marine Chemistry, 202, 97-120.

Xie, H., Bélanger, S., Demers, S., Vincent, W.F. and Papakyriakou, T.N., 2009. Photobiogeochemical cycling of carbon monoxide in the southeastern Beaufort Sea in spring and autumn. Limnology and Oceanography, 54(1), 234-249.

Zafiriou, O.C., Xie, H., Nelson, N.B., Najjar, R.G. and Wang, W., 2008. Diel carbon monoxide cycling in the upper Sargasso Sea near Bermuda at the onset of spring and in midsummer. Limnology and Oceanography, 53(2), 835-850.

**Responses are *italicized*.**
**AR** stands for authors' response

This work presents the seasonal distribution (May, Aug, Nov, and Jan 2015) of DOM (DOC concentrations, CDOM absorption and CDOM fluorescent components (from PARAFAC analysis) in Pearl River estuary (PRE), China. DOC concentrations and CDOM absorption and fluorescence properties (and their qualitative metrics) were examined in relation to salinity as well as to each other. In addition, fluxes of DOC and CDOM from the PRE to South China Sea were also estimated. Overall, results of this study provides new insights into the seasonal DOC and optical properties of CDOM in PRE. In comparison, most previous studies have mainly reported one or two field campaigns, while this study comprised a more seasonal study (four field campaigns).

However, the analysis of the data throughout involves simple correlation analysis and is descriptive with no rigorous analysis of field data (spatial analysis, precipitation, chlorophyll and turbidity measurements that were indicated in the text to have been measured). The additional analysis would support a better understanding of the sources and sinks related to the DOM in PRE.

*AR: All the discussion and conclusions are based on the quantitative analysis of the data. Our results have clearly demonstrated that physical mixing (i.e. salinity) is the predominant factor controlling the variability of DOM in the PRE (Figs. 3 and 4). We have now added a principal component analysis (PCA) on the all-season dataset to further strengthening the manuscript. The PCA includes variables in addition to salinity, such as water temperature, chl-a, nutrients, suspended particulate matter, and freshwater discharge rate. The DOM dynamics is represented by CDOM absorption at 330 nm ($a_{330}$) and DOC concentration. The first two axes of the PCA explained ~74% of the variability in the dataset (see graph below). Using the first axis on the following graph, one can see that DOC and $a_{330}$ (along with nitrate and silicate), are strongly negatively correlated to salinity, which is a typical indication of a conservative mixing behavior. In contrast, DOC and $a_{330}$ are only weakly linked to the freshwater discharge rate, again consistent with our result (line 604-606 & Fig. S9 in the original version).*

*We have added the plot to the main text (Fig. 9) and briefly discussed it in the revised manuscript (lines 453-461).*

[Figure]

*Figure: PCA analysis based on the all-season dataset. SPM: suspended particulate matter; $PO_4^{3-}$: phosphate; $NO_2^-$: nitrite; DOC: dissolved organic carbon; $a_{CDOM}(330)$: CDOM absorption coefficient at 330 nm; $NO_3^-$: nitrate; Chla: chlorophyll a; $SiO_4^{4-}$: silicate; discharge: freshwater discharge rate.*

I find that the manuscript needs further improvements and the authors should address some major concerns/suggestions before the paper can be accepted for publication.

Major comments/suggestions: 1) There are various major sources of freshwater to the PRE. Previous studies have also indicated spatial differences in the surface and bottom properties in CDOM optical properties (absorption coefficients and spectral slope; e.g., Lei et al. 2018). Furthermore, seasonal analysis of DOC (Ye et al. 2018) indicated strong seasonality in DOC with substantial removal of DOC in the salinity range 5-22. I think a more comprehensive analysis using all the available data (e.g., chlorophyll, turbidity, etc) including spatial distribution plots (surface and bottom) would greatly help in supporting the conclusions of this study.

*AR: Our conclusions are based on an analysis of not only quantitative variables ([DOC], $a_{CDOM}$, and 5 FDOM components) but also a large number of qualitative variables ($E_2/E_3$, BIX, HIX, and the percentages of FDOM components). The more comprehensive data analysis (including chlorophyll and SPM) using PCA shown above further strengthens the conclusions already reached in our article.*

*The difference between the studies the reviewer mentioned and ours may be caused by different spatiotemporal coverage of water sampling and potentially large interannual variability of the DOM dynamics in the PRE, as already suggested in the original manuscript (line 131-141; line 548-553 in the original version). In the revised manuscript, we reinforced this point by including the very recent reference suggested by the reviewer (i.e. Ye et al., 2018; the paper by Lei et al. (2018) was already cited). Note that the potential interannual variability further complicates the generalization of the DOM dynamics and biogeochemical cycling in the PRE.*

2) Throughout this study the authors describe the data collected in the main estuary as the saltier zone as opposed to fresh water zone. I think a more traditional separation of the zones (e.g., Cai et al. 2004; upstream region, estuary, outer estuary) would be more appropriate and could better support the results of this study.

*AR: The "head region" is now used to refer to the narrow low-salinity zone and "main estuary" to denote the saltier zone.*

3) The absorption coefficient at 330 nm used in this study has not generally been used and therefore not easily comparable to other studies. Although Table S1 includes some of these wavelengths, it would help if the authors replace the absorption at 330 nm with another commonly used wavelength. Also the spectral slope between 275-295 nm is now generally used to assess CDOM properties and should be included in the analysis.

*AR: There are several points to support the use of the wavelength of 330 nm for $a_{CDOM}$. First, the wavelength at or close to 330 nm is where the majority of aquatic CDOM photoreactions (including photobleaching) exhibits the maximum rates in surface waters under solar radiation (e.g. Vähätalo et al., 2000; Zhang et al., 2006; Osburn et al., 2009; Xie et al., 2009, 2012; White et al., 2010; Song et al., 2013; Hong et al., 2014; Qi et al., 2018). The wavelength of 330 nm is, therefore, is linked to an important process controlling the cycling of CDOM in natural waters. This point has now been explicitly stated in the revised manuscript. Second, $a_{CDOM}(330)$ has been used as an indicator of CDOM content by many labs including those well recognized labs (e.g. Brisco and Ziegler, 2004; White et al., 2008; Osburn et al., 2009; Xie et al., 2009; Gareis et al., 2010; Mann et al., 2012; Song et al., 2017; Qi et al., 2018). Third, there is no consensus on which wavelength is best to serves as a proxy of CDOM content. A limited review of the literature shows at least 13 wavelengths (254, 300, 320, 325, 330, 350, 355, 375, 380, 400, 412, 420, and 440 nm) have been adopted for this purpose.*

*Finally, in case the reader is interested in other wavelengths, we have provided absorption coefficients at 6 other wavelengths across the UV and visible regimes that are commonly seen as well in the literature (Table S1 in the Supplemental Material). Furthermore, we also published the spectral slope between 300 and 500 nm (again in Table S1), so that the reader can retrieve the absorption coefficient at any wavelength between the 300 and 500 nm interval. We believe we have done our best to accommodate the different needs of the scientific community.*

*The spectral slope and slope ratio ($S_{275-295}$, $S_{350-400}$ and $S_R$) were also investigated and they showed similar patterns to those of $E_2/E_3$. $E_2/E_3$ was chosen, because 1) it exhibited larger variations than the spectral slopes and slope ratio; 2) it has been used as a valid proxy of molecular weight for a much longer history (De Haan, 1983; Peuravuori and Pihlaja, 1997) than the spectral slope and slope ratio, particularly for fresh and brackish waters (including estuarine waters); 3) it is very sensitive to and quantitatively responds to photobleaching (Lou and Xie, 2006; Qi et al., 2018) and biogeochemical processing; 4) a quantitative and validated relationship between $E_2/E_3$ and the molecular weight (MW) of CDOM is available (Lou and Xie, 2006; Qi et al., 2018), so that this relationship can be used to estimate the MW of CDOM for the present study (line 439-443 in the original manuscript). Note that such a broadly applicable relationship has not been established between $S_{275-295}$ and MW.*

*We have explicitly stated in the revised manuscript that $E_2/E_3$ serves similar functions to those of $S_{275-295}$ (lines 205-210).*

4) CDOM generally is a good optical proxy for DOC, especially in estuaries. Also, CDOM undergoes rapid photobleaching in the estuaries or the coastal waters. It may not be useful include estimates of CDOM fluxes at 330 mn from the estuary to the SCS, especially since the wavelength used is so unique to this study.

*AR: For the wavelength issue, we think we have chosen an appropriate wavelength to represent CDOM content and photobleaching and (see our response to comment 3).*

*Even if CDOM degrades rapidly in estuaries and coastal waters (often that's not true, see below), it does not necessarily imply that the export of CDOM to the ocean is not important. If the remaining component of CDOM exported to the ocean, albeit small in amount, is bio- and photo-resistant, it can accumulate in open oceans. This is why the oceanographic community has put tremendous efforts in identifying and quantifying potential terrigenous DOM (the main part of it could be CDOM) in open oceans (Opsahl and Benner, 1997; Cauwet, 2002; Raymond et al., 2007; Bianchi and Allison, 2009;*

*Dai et al., 2012; Wang et al., 2012; Raymond and Spencer, 2015). This issue is fundamental for understanding the global carbon cycle. This is in part why (other aspects involve ocean optics) scientists have started making efforts to evaluate the land-to-ocean CDOM fluxes (e.g. Stedmon et al., 2011; Spencer et al., 2013; Aarnos et al., 2018).*

*Concerning the specific case of the PRE, our data clearly indicate that CDOM behaved essentially conservatively in the main estuary (i.e. ca. salinity >5), implying that photobleaching was insignificant. We also made a direct estimate of the amount of CDOM that could be removed by photobleaching in the PRE; it was at most 7% (line 487-507 in the original version), supporting the inference from the conservative CDOM vs. salinity plots. This not surprising, given that 1) the residence time of freshwater (and thus CDOM as well) in the PRE is very short (a few days, line 494-497 in the original version; 2) the competition of light absorption by particles (water in the PRE is turbid); and 3) self-shading due to high CDOM and particle abundances in the PRE.*

*In general, estuaries and strongly runoff-impacted coastal waters are not prone to having efficient CDOM photobleaching due to at least the three causes stated above. Efficient photobleaching usually takes place in waters on the outer shelf (e.g. shelf break) where CDOM has been sufficiently spread out and the majority of the particles have settled down to the seafloor (so that self-shading is diminished).*

5) It may be useful to look at meteorological data (e.g., wind field) to see if mixing played a role in reducing the variability in DOM surface and bottom properties.

**AR:** *It is the salinity and temperature structures (Figs. S1 and S2), not the meteorological information, that **directly** indicate the degree of water column mixing. We used the salinity and temperature data to discuss the surface and bottom variability on each relevant occasion.*

Minor comments: -No indication of how salinity was measured -Methods section could describe the study site rather than in the Introduction.

**AR:** *It is already there (see line 182-183 in the original version).*

References: X. Lei, J. pan, A. T. Devlin. 2018. Mixing behavior of chromophoric dissolved organic matter in the Pearl River Estuary in sprig. Continental Shelf Research, 154, 46-54.

F. Ye, W. Guo, G. Wei, and G. Jia. 2018. The sources and transformations of dissolved organic matter in the Pearl River Estuary, China, as revealed by stable isotopes. J. Geophys. Res.: Oceans, 123, 6893-6908.

*AR: Thanks for providing these two references. Lei et al (2018) was already cited in the original manuscript. Ye et al (2018) has now been added.*

**References cited in this response:**

Aarnos, H., Gélinas, Y., Kasurinen, V., Gu, Y., Puupponen, V.M. and Vähätalo, A.V., 2018. Photochemical mineralization of terrigenous DOC to dissolved inorganic carbon in ocean. Global Biogeochemical Cycles, 32(2), 250-266.

Bianchi, T.S. and Allison, M.A., 2009. Large-river delta-front estuaries as natural "recorders" of global environmental change. Proceedings of the National Academy of Sciences, 106(20), 8085-8092.

Brisco, S. and Ziegler, S., 2004. Effects of solar radiation on the utilization of dissolved organic matter (DOM) from two headwater streams. Aquatic microbial ecology, 37(2), 197-208.

Cauwet, G., 2002. DOM in the coastal zone. Biogeochemistry of marine dissolved organic matter.

Dai, M., Yin, Z., Meng, F., Liu, Q. and Cai, W.J., 2012. Spatial distribution of riverine DOC inputs to the ocean: an updated global synthesis. Current Opinion in Environmental Sustainability, 4(2), 170-178.

De Haan, H., 1983. Use of ultraviolet spectroscopy, gel filtration, pyrolysis/mass spectrometry and numbers of benzoate metabolizing bacteria in the study of humification and degradation of aquatic organic matter. In: Christman, R.F., Gjessing, E.T. (Eds.), Aquatic and Terrestrial Humic Materials. Ann Arbor Science, Michigan, pp. 165–182.

Gareis, J.A., Lesack, L.F. and Bothwell, M.L., 2010. Attenuation of in situ UV radiation in Mackenzie Delta lakes with varying dissolved organic matter compositions. Water Resources Research, 46(9).

Hong, J., Xie, H., Guo, L. and Song, G., 2014. Carbon monoxide photoproduction: implications for photoreactivity of arctic permafrost-derived soil dissolved organic matter. Environmental science & technology, 48(16), 9113-9121.

Lou, T., Xie, H., 2006. Photochemical alteration of the molecular weight of dissolved organic matter. Chemosphere 65, 2333–2342.

Mann, P.J., Davydova, A., Zimov, N., Spencer, R.G.M., Davydov, S., Bulygina, E., Zimov, S. and Holmes, R.M., 2012. Controls on the composition and lability of dissolved organic matter in Siberia's Kolyma River basin. Journal of Geophysical Research: Biogeosciences, 117(G1).

Opsahl, S. and Benner, R., 1998. Photochemical reactivity of dissolved lignin in river and ocean waters. Limnology and Oceanography, 43(6), pp.1297-1304.

Osburn, C.L., Retamal, L. and Vincent, W.F., 2009. Photoreactivity of chromophoric dissolved organic matter transported by the Mackenzie River to the Beaufort Sea. Marine Chemistry, 115(1-2), 10-20.

Peuravuori, J., Pihlaja, K., 1997. Molecular size distribution and spectroscopic properties of aquatic humic substances. Anal. Chim. Acta 337, 133–149.

Qi, L., Xie, H., Gagné, J.P., Chaillou, G., Massicotte, P. and Yang, G.P., 2018. Photoreactivities of two distinct dissolved organic matter pools in groundwater of a subarctic island. Marine Chemistry, 202, 97-120.

Raymond, P.A. and Spencer, R.G., 2015. Riverine DOM. In Biogeochemistry of marine dissolved organic matter (pp. 509-533). Academic Press.

Raymond, P.A., McClelland, J.W., Holmes, R.M., Zhulidov, A.V., Mull, K., Peterson, B.J., Striegl, R.G., Aiken, G.R. and Gurtovaya, T.Y., 2007. Flux and age of dissolved organic carbon exported to the Arctic Ocean: A carbon isotopic study of the five largest arctic rivers. Global Biogeochemical Cycles, 21(4).

Song, G., Li, Y., Hu, S., Li, G., Zhao, R., Sun, X. and Xie, H., 2017. Photobleaching of chromophoric dissolved organic matter (CDOM) in the Yangtze River estuary: kinetics and effects of temperature, pH, and salinity. Environmental Science: Processes & Impacts, 19(6), 861-873.

Song, G., Xie, H., Bélanger, S., Leymarie, E. and Babin, M., 2013. Spectrally resolved efficiencies of carbon monoxide (CO) photoproduction in the western Canadian Arctic: particles versus solutes. Biogeosciences, 10(6), 3731-3748.

Spencer, R. G. M., Aiken, G. R., Dornblaser, M. M., Butler, K. D., Holmes, R. M., Fiske, G., Mann, P. J., and Stubbins, A.: Chromophoric dissolved organic matter export from U.S. rivers, Geophys. Res. Lett., 40, 1575–1579, doi:10.1029/grl50357, 2013.

Stedmon, C. A., Amon, R. M. W., Rinehart, A. J., and Walker, S. A.: The supply and characteristics of colored dissolved organic matter (CDOM) in the Arctic Ocean: Pan Arcitc trends and differences, Mar. Chem., 124, 108–118, 2011.

Vähätalo, A.V., Salkinoja‐Salonen, M., Taalas, P. and Salonen, K., 2000. Spectrum of the quantum yield for photochemical mineralization of dissolved organic carbon in a humic lake. Limnology and Oceanography, 45(3), 664-676.

Wang, X., Ma, H., Li, R., Song, Z. and Wu, J., 2012. Seasonal fluxes and source variation of organic carbon transported by two major Chinese Rivers: The Yellow River and Changjiang (Yangtze) River. Global Biogeochemical Cycles, 26(2).

White, E.M., Kieber, D.J. and Mopper, K., 2008. Determination of photochemically produced carbon dioxide in seawater. Limnology and Oceanography: Methods, 6(9), 441-453.

White, E.M., Kieber, D.J., Sherrard, J., Miller, W.L. and Mopper, K., 2010. Carbon dioxide and carbon monoxide photoproduction quantum yields in the Delaware Estuary. Marine Chemistry, 118(1-2), 11-21.

Xie, H., Bélanger, S., Demers, S., Vincent, W.F. and Papakyriakou, T.N., 2009. Photobiogeochemical cycling of carbon monoxide in the southeastern Beaufort Sea in spring and autumn. Limnology and Oceanography, 54(1), 234-249.

Xie, H., Bélanger, S., Song, G., Benner, R., Taalba, A., Blais, M., Tremblay, J.É. and Babin, M., 2012. Photoproduction of ammonium in the southeastern Beaufort Sea and its biogeochemical implications. Biogeosciences, 9(8), 3047-3061.

Zhang, Y., Xie, H. and Chen, G., 2006. Factors affecting the efficiency of carbon monoxide photoproduction in the St. Lawrence estuarine system (Canada). Environmental science &

technology, 40(24), 7771-7777.

**Response to Public Short Comment**

**Responses are *italicized*.**
*AR* stands for authors' response

**SC:** Dissolved organic matter is an important component of the carbon cycle in aquatic systems and it exerts direct impact on the overall biogeochemical process in the ocean. DOM spectroscopy has emerged as a cost-effective and easy-to-measure technique for quantifying and, more recently, qualify the DOM content in the environment. The manuscript by Li and colleagues brings results on DOM amount (expressed by means of DOC and spectroscopic measurements), characterization (through EEM- PARAFAC), fluxes and seasonal variability for the Pearl River Estuary, China. The data set is robust and the methods applied align with current literature. Although the sampling grid remains the same for the different seasons, the seasonal averages presented in the MS might be biased by the spatial variability presented within the water masses spatial distribution within the region. Therefore, I suggest the authors to have lead the MS through a more "oceanographic point of view", i.e., by investigating the seasonal changes within the water masses presented within the region.

*AR: We adopted the classical approaches for describing chemical variables in an estuary: property vs. distance and property vs. salinity. Salinity is an indication of mixing processes, while distance is more related to residence time and processing time. These two approaches are complementary. The seasonal averages presented in our MS are based on the "distance" approach, given that the coordinates of the sampling stations were the same for different seasons. These averages thus reflect the seasonality of the residence and processing times of the water masses in the estuary. On the other hand, the property vs. salinity plots provided information on how the mixing behavior of a variable of interest changed seasonally. As water masses in an estuary are primarily defined by salinity, the seasonal variability revealed by this approach is essentially water mass-based. A more complete picture of the seasonality of the variables is acquired by combining the results from the distance and salinity approaches. This is the rationale behind the scheme we employed to present our data.*

*As our sampling stations were principally distributed along the main longitudinal axis of the estuary with little lateral coverage (as is true for many other estuarine studies), the data thus collected is insufficient to characterize the spatial distribution of water masses in the region, making the "oceanographic point of view" approach suggested by the reviewer difficult to implement.*

**SC:** Although the manuscript is well written and reads easily, the way that sections are structure makes the manuscript repetitive when presenting and discussing results. I think it would become more concise and interesting if the authors focus on making a rearrangement of sections (by merging/condensing some of them) and on making a review through the text to avoid such repetitions. Additionally, the introduction is a bit too long and could be shortened by providing only information needed for interpretation of results from this study. Thus, to my judgment, the manuscript may be publishable after major reviews.

*AR: Following the reviewer's comments, we have restructured and shortened the Introduction and Results sections.*

GENERAL COMMENTS:

**SC:** The abstract does not clearly illustrate the main findings obtained in the study.

*AR: We have shortened and rewritten the abstract to focus on the main findings.*

**SC:** The hypothesis presented in section 1.3 seem weak and vague, and could be sharper. Seasonal variability in DOM flux is already expected from an estuary with marked seasonal variability in freshwater export, as documented by the authors.

*AR: DOM flux is only one of the many DOM variables (both quantitative and qualitative) reported in this study. In fact, most other variables showed smaller spatial and seasonal variations than expected from this sizable estuary with an important seasonal fluctuation of freshwater discharge (see the Conclusions section). The fluxes of DOC and CDOM are also the lowest compared to other major world rivers, contrasting with the hypothesis. Therefore, we feel that the current working hypothesis is appropriate and strong enough.*

**SC:** Sampling strategy: why was decided to collect the "deep water" sample near the bottom and not below the pycnocline? It can be affected by sediment resuspension, if there is any.

*AR: One of the purposes of this study was to determine if there was a significant sedimentary impact on DOM in the water column. The consistent property–salinity patterns (Figures 3 and 4) and lack of relationship with suspended particle concentration (Line 512 in the original version and now the PCA*

*analysis as well) suggest that this effect was minor. Note that the effect of sediment resuspension, if any, could reach the depths just below the pycnocline, given the overall shallow water depths of the PRE (mostly <10 m, Table 1 in the original version)*

**SC:** Have the authors looked at the CDOM absorption spectral slope and slope ratio? It could provide more insights into the photochemical reactions along the estuarine mixing.

*AR: The spectral slope and slope ratio ($S_{275-295}$, $S_{350-400}$ and $S_R$) were also investigated and they showed similar patterns to those of $E_2/E_3$. $E_2/E_3$ was chosen, because 1) it exhibited larger variations than the spectral slopes and slope ratio; 2) it has been used as a valid proxy of molecular weight for a much longer history (De Haan, 1983; Peuravuori and Pihlaja, 1997) than the spectral slope and slope ratio, particularly for fresh and brackish waters (including estuarine waters); 3) it is very sensitive to and quantitatively responds to photobleaching (Lou and Xie, 2006; Qi et al., 2018) and biogeochemical processing; 4) a quantitative and validated relationship between $E_2/E_3$ and the molecular weight (MW) of CDOM is available (Lou and Xie, 2006; Qi et al., 2018), so that this relationship can be used to estimate the MW of CDOM for the present study (line 439-443 in the original manuscript). Note that such a broadly applicable relationship has not been established between $S_{275-295}$ and MW.*

*We have explicitly stated in the revised manuscript that $E_2/E_3$ serves similar functions to those of $S_{275-295}$ (lines 205-210).*

**SC:** The authors could also try to use multivariate analysis (e.g., PCA) to analyze the variability between the campaigns (i.e., over time) and to elucidate what are the main drivers on DOM variability within the region.

*AR: Our results have clearly demonstrated that physical mixing (i.e. salinity) is the predominant factor controlling the variability of DOM in the PRE (Figs. 3 and 4). Here we performed a principal component analysis (PCA) on the all-season dataset that includes variables in addition to salinity, such as water temperature, chl-a, nutrients, suspended particulate matter, and freshwater discharge rate. The DOM dynamics is represented by CDOM absorption at 330 nm ($a_{330}$) and DOC concentration. The first two axes of the PCA explained ~74% of the variability in the dataset. Using the first axis on the following graph, one can see that DOC and $a_{330}$ (along with nitrate and silicate) are strongly negatively correlated to salinity, which is a typical indication of a conservative mixing behavior. In contrast, DOC and $a_{330}$ are only weakly linked to the freshwater discharge rate, again consistent with our result (line 604-606 & Fig. S9 in the original version).*

*We have added the plot to the main text (Fig. 9) and briefly discussed it in the revised manuscript (lines 453-461).*

[Figure]

*Figure: PCA analysis based on the all-season dataset. SPM: suspended particulate matter; $PO_4^{3-}$: phosphate; $NO_2^-$: nitrite; DOC: dissolved organic carbon; $a_{CDOM}$(330): CDOM absorption coefficient at 330 nm; $NO_3^-$: nitrate; Chla: chlorophyll a; $SiO_4^{4-}$: silicate; discharge: freshwater discharge rate.*

**SC:** I suggest the authors to compare their PARAFAC-derived components spectra with the OpenFluor database (https://openfluor.lablicate.com/). This would benefit the comparison established with other studies along the MS.

*AR: This has now been done and added to the Methods section.*

**SC:** With respect to the sources of DOM to region, especially the pollution-derived DOM, they could be more stressed along the MS. It is not totally clear how the findings of this study support that.

*AR: Pollution-derived DOM is a dominant source of DOM in the upper reach of the PRE, generally upstream of Humen. Note that this is **not** our finding, rather a conclusion of previous studies (as clearly stated in the Introduction, line 120-130 in the original version). Some previous studies (e.g. Lin et al., 2007; He et al., 2010) conducted sampling much farther upstream into the Guangzhou Channel, where the capital of the Guangdong Province is located. The concentration of DOC in that channel*

*could reach as high as 500 uM, which is ~4 times the background DOC (119 uM) in the Pearl River upstream of the Pearl River Delta (He, 2010). This observation, combined with the enormous amount of industrial and domestic waste discharged into the PRE ($5.8*10^9$ tons/year) across its deltaic region, led these authors to concluding that the highly enriched DOC in the upper reach of the estuary mostly originates form sewage effluents. The pollution-derived DOC is, however, very labile and much of it is consumed by bacteria in the low-salinity zone of the estuary (He, 2010, He et al., 2010). Our data provided two lines of evidence to support the pollution argument for our sampling seasons: 1) a rapid drawdown of DOC and CDOM in the upper reach, which is consistent with the labile character of pollution-derived DOM as elucidated in the previous studies; 2) the protein-rich character of this DOM pool as revealed by the fluorescence-based metrics (BIX and %(C1+C5)). These two points are elaborated in the relevant context (section 4.1).*

**SC:** Section 4.5 establishes comparisons among global DOM studies but I expected the discussion to bring some conclusions on the reason for such differences rather than just comparing them.

*AR: We are a bit confused by this comment. Section 4.5 clearly indicates that two factors mainly contribute to the lowest DOM abundance and flux in the PRE: 1) the deficiency of organic matter in soil of the Pearl River's watershed having almost no forest; 2) the rapid microbial consumption of pollution-derived DOM in the upper estuary. These two factors are once again emphasized in the Conclusions section. Moreover, the main portion of section 4.5 is discussion instead of "just comparison".*

SPECIFIC COMMENTS:

**SC:** L75-79: authors could give more background on anthropogenic/pollution-derived DOM, given that it is a DOM source for the region, as pointed out in this study.

*AR: This point is actually brought up on two other occasions in the Introduction about the PRE (line 122-125; line 145-148 in the original version). We believe the background information for this point is sufficient, particularly considering that the Introduction is already long and needs to be shortened.*

**SC:** L115-119: Please present values (ranges) for the variables. How much does the phytoplankton biomass vary within the seasons?

*AR: The Introduction is greatly shortened and this kind of non-essential information is not provided in the revised version in part because different papers reported widely different values and in part because we conducted a PCA that includes the chl-a values from our cruises.*

**SC:** L124-125: Are there only those two studies supporting this affirmation? No study published in English?

*AR: After re-searching the literature, we found one more paper (He et al., 2010, published in English) for supporting this argument. This reference has now been added.*

**SC:** L306-307: what do the authors mean by "freshwater input from this river appeared to have little influence on [DOC]" ?

*AR: Sta. M01, 02 and 03 were distributed along a transect across the three outlets of the East River (i.e. upper, middle, and lower outlets, Fig. 1). However, the [DOC]s at these three stations in May were nearly constant, suggesting that the freshwater input from the East River did not significantly affect the [DOC]. This further implies that [DOC] in the East River in May was roughly equal to that in the North River, which is the larger freshwater source of the upper reach of the PRE (~2 times that of the East River, line 95-98 in the Introduction).*

*The revised manuscript does not contain this content anymore in order to restructure and condense the Results section.*

**SC:** L500-503: Missing references.

*AR: Thanks. The missing reference (He, 2010) was added.*

**SC:** L522-526: I found the explanation for different mixing behavior weak and should be discussed more in deep.

*AR: The observation needs to be explained: In the saltier zone, [DOC] remained rather constant while [CDOM] (in terms of $a_{330}$) decreased linearly with increasing salinity in November; in August and January, [CDOM] decreased much faster than [DOC] with increasing salinity.*

*Our explanation: 1) CDOM was only a minor component of the entire DOM pool (so that the change in [CDOM] had little impact on [DOC]); 2) the marine endmember was less colored (i.e. lower $a_{CDOM}$) than the freshwater endmember (so that [CDOM] decreased with increasing salinity); 3) the difference*

*between the marine and freshwater DOC endmembers was much smaller than that for CDOM (so that the salinity-based gradient for [DOC] was much smaller than that for [CDOM]). A combination of points 2 and 3 leads to a smaller [DOC]-normalized $a_{CDOM}$ for the marine endmember than that for the freshwater endmember (which is what we presented in the manuscript).*

*We believe that our explanation is sound. These points are made clearer in the revised version.*

**SC:** L527-535: this paragraph/discussion could be deepened in the sense to explain the reasons for such variations.

*AR: This paragraph is actually a summary of section 4.2. The deeper discussion is presented in the preceding paragraphs. Moreover, the lack of sampling within the main freshwater outlets (e.g. Hengmen, Jiaomen, Hongqimen) downstream of Humen prevents us from further discussing the potential impact of different freshwater masses.*

**SC:** L538-547: Why does it only have good correlations for summer and winter? What happens with the correlations during the other seasons? Additionally, was the DOC- aCDOM correlation significant and strong? I ask that, because that correlation does not hold true for several environments.

*AR: In spring and fall, [DOC] in the saltier zone was relatively constant and consequently not correlated with salinity as opposed to the case in summer and winter. $a_{CDOM}$, however, showed negative correlations with salinity in all three sampling seasons (summer, fall, and winter). This distribution pattern is already described in section 3.4 and discussed in section 4.2, and thus not repeated in section 4.3. Instead, we referred the reader to Fig. 3 for understanding the relevant context.*

*Yes, the DOC-$a_{CDOM}$ is significant and strong (p<0.0001, now added to the text). Although this kind of correlation may not hold universally, many marine environments, include estuaries and coastal waters, do exhibit such correlations, e.g. the Middle Atlantic Bight (Del Vecchio and Blough, 2004), Yukon River (Spencer et al., 2009), Yangtze River estuary (Guo et al., 2014), and the Baltic coastal sea (Harvey et al., 2015).*

**SC:** L556-580: authors could deepen the discussion regarding the fluxes.

*AR: More discussion about the fluxes is provided in section 4.5.*

**SC:** L615-623: what could the authors point out as the reason for such differences?

*AR: This is because the [DOC] and [CDOM] in the PRE are the lowest among the world major rivers. Line 600-6004 in the original version has already speculated on two factors causing this phenomenon: the poorly forested watershed of the Pearl River and the rapid degradation of sewage-derived DOM.*

**SC:** Figure 1: It would be interesting to have two panel composing this figure: one with the sampling sites and another with the city names and also the main circulation patterns.

*AR: As the circulation pattern changes with season, which needs four panels to do it. Moreover, the distributional pattern of the sampling stations (an along-estuary transect without much cross-estuary coverage) does not allow us to adequately characterize the circulation patterns during our sampling periods. Hence, adding a circulation pattern panel may not significantly improve the presentation and interpretation of the data.*

**SC:** Figs 3, 4, 5 and 8: please present the curve fits and stats.

*AR: Lines in Figure 5 denote the conservative mixing lines, not the data fits. The curve fits and statistics are already presented in Table 4 for Figures 3 and 4 and in Table 5 for Figure 8 in the original manuscript.*

**References cited in this response:**

De Haan, H., 1983. Use of ultraviolet spectroscopy, gel filtration, pyrolysis/mass spectrometry and numbers of benzoate metabolizing bacteria in the study of humification and degradation of aquatic organic matter. In: Christman, R.F., Gjessing, E.T. (Eds.), Aquatic and Terrestrial Humic Materials. Ann Arbor Science, Michigan, pp. 165–182.

Del Vecchio, R. and Blough, N.V., 2004. Spatial and seasonal distribution of chromophoric dissolved organic matter and dissolved organic carbon in the Middle Atlantic Bight. Marine Chemistry, 89(1-4), 169-187.

Guo, W., Yang, L., Zhai, W., Chen, W., Osburn, C.L., Huang, X. and Li, Y., 2014. Runoff‐mediated seasonal oscillation in the dynamics of dissolved organic matter in different branches of a large bifurcated estuary−The Changjiang Estuary. Journal of Geophysical Research: Biogeosciences, 119(5), 776-793.

Harvey, E.T., Kratzer, S. and Andersson, A., 2015. Relationships between colored dissolved organic matter and dissolved organic carbon in different coastal gradients of the Baltic Sea. Ambio, 44(3), 392-401.

He, B., Dai, M., Zhai, W., Wang, L., Wang, K., Chen, J., Lin, J., Hua, A., and Xu, Y.: Distribution, degradation and dynamics of dissolved organic carbon and its major compound classes in the pearl river estuary, China, Mar. Chem., 119, 52–64, 2010.

He, B.: Organic Matter in the Pearl River Estuary: its Composition, Source, Distribution, Bioactivity and their Linkage to Oxygen Depletion (Ph.D. Dissertation), Xiamen university, 2010 (In Chinese).

Lin, J.: On the behavior and flux of Dissolved Organic Carbon in two large Chinese estuaries-Changjiang and Zhujiang (Master Dissertation), Xiamen university, 2007 (In Chinese).

Lou, T., Xie, H., 2006. Photochemical alteration of the molecular weight of dissolved organic matter. Chemosphere 65, 2333–2342.

Peuravuori, J., Pihlaja, K., 1997. Molecular size distribution and spectroscopic properties of aquatic humic substances. Anal. Chim. Acta 337, 133–149.

Qi, L., Xie, H., Gagné, J.P., Chaillou, G., Massicotte, P. and Yang, G.P., 2018. Photoreactivities of two distinct dissolved organic matter pools in groundwater of a subarctic island. Marine Chemistry, 202, 97-120.

Spencer, R.G., Aiken, G.R., Butler, K.D., Dornblaser, M.M., Striegl, R.G. and Hernes, P.J., 2009. Utilizing chromophoric dissolved organic matter measurements to derive export and reactivity of dissolved organic carbon exported to the Arctic Ocean: A case study of the Yukon River, Alaska. Geophysical Research Letters, 36(6).

---

## Editor Decision (ED1)

15-Mar-19

Dear Dr. Xie,

Thanks for providing responses to 3 reviewers and to a member of the biogeoscience community who commented on your article bg-2018-403.  Most of your responses are appropriate and their suggestions will result in a better piece of scientific work if they are considered in a new version.  Based on those comments and responses and my own reading, I consider that your data and results merit publication in Biogeosciences, but the article cannot be considered in the journal as it is and needs to undergo a major revision that would involve sending it out for review.  Please consider the following in particular:

1. Main conclusions of the article are difficult to follow since there is repetition of results throughout the text, and there are results that are not considered in the discussion, deviating attention to main points of the article.  Examples:  a) Water temperature is shown but there is no discussion of it, b) idem with results on water column mixing, c) in page 263,  "Bottom water salinity at most stations was nearly identical to SWS in January, slightly greater in May, moderately elevated in November, and much higher in August  (Fig. S2)".  There is no discussion of it in the text.  If there is a meaning for this, then it needs to be quantitatively explained, not as currently written (slightly, much, etc.).

2. There is an excessive use of Supplementary tables and figures **around relevant** discussion and conclusions.  Supplementary figures and tables are meant to back up tables and figures of the main text.  A new version will require rethinking and reorganizing tables and figures accordingly.

3.  Qualitative assessments should be avoided. such as saltier, less salty (Reviewer 3 suggests using well-known and accepted terminology by the estuarine community).

4. Hypothesis.  "… hypothesize that DOM in the PRE presents substantial seasonal variability in terms of both abundance and **chemical composition** and that the PRE is an important source of DOM to global oceans."

   Chemical composition you are referring to is targeting a quantitatively minor fraction of DOC pool (in the order of 2%), therefore you cannot test that hypothesis for the entire pool using this approach.

5. What are units of DOC and CDOM fluxes in Table 6.  Nowhere is mentioned how you estimated fluxes from absorbance data.

6. Keep in mind Short Comment:
   *" Although the manuscript is well written and reads easily, the way that sections are structure makes the manuscript repetitive when presenting and discussing results. I think it would become more concise and interesting if the authors focus on*

*making a rearrangement of sections (by merging/condensing some of them) and on making a review through the text to avoid such repetitions. Additionally, the introduction is a bit too long and could be shortened by providing only information needed for interpretation of results from this study…."*

7. Section on Pearl River estuary is definitely too long, so it is background on DOM. Please choose the most relevant aspects.

8. "… [DOM], [CDOM], and [FDOM] stand for the abundances of…".  Square brackets are used in chemistry to denote concentration and [CDOM] and [FDOM] are not; they could be considered proxies of concentration.  Different things.

9. Use of non-standard acronym such as SWS only makes reading more difficult (It is used only 7 times in the text, all in one page).

10. P, 286, P 409, etc..  Correlation and regression are not the same.  In correlation there is no independent variable and coefficient of correlation (r) ranges from -1 to +1.  In regression, there is X and Y, and coefficient of determination ($R^2$) ranges from 0 to 1 (0 to 100%).  Please check and revise accordingly

11. Method.  "Hansell's low carbon ([DOC]: 1–2 µmol L−1) and deep Florida Strait ([DOC]:  41–44 µmol L−1) reference waters "

    What was the quantitatively results of this calibration?

12. About the analytical uncertainty mentioned by Reviewer 2.  #8.  " … *aCDOM at 330 nm (a330) was 2.19 m−1 (range: 1.19–4.37 m−1)…*" corresponds to the range of values of a330 measured in the river during the August cruise.  Analytical uncertainty on the other hand, deals with dispersion of values associated to a measurand, therefore samples has to be as similar as possible.

13. Lines 375-376.  Please explain what you want to say here

14. Lines 235-236 should be in methods

Sincerely yours

Silvio Pantoja
Associate Editor